# RNA-mediated double-strand break repair by end-joining mechanisms

Youngkyu Jeon [1,4], Yilin Lu [1,10], Margherita Maria Ferrari [2,5,10], Tejasvi Channagiri[2,10], Penghao Xu [1,10], Chance Meers [1,6], Yiqi Zhang[1,7,8], Sathya Balachander[1,9], Vivian S. Park[3], Stefania Marsili[1], Zachary F. Pursell [3], Nataša Jonoska [2] ✉ & Francesca Storici [1] ✉

Double-strand breaks (DSBs) in DNA are challenging to repair. Cells employ at least three DSB-repair mechanisms, with a preference for non-homologous end joining (NHEJ) over homologous recombination (HR) and microhomology-mediated end joining (MMEJ). While most eukaryotic DNA is transcribed into RNA, providing complementary genetic information, much remains unknown about the direct impact of RNA on DSB-repair outcomes and its role in DSB-repair via end joining. Here, we show that both sense and antisense-transcript RNAs impact DSB repair in a sequence-specific manner in wild-type human and yeast cells. Depending on its sequence complementarity with the broken DNA ends, a transcript RNA can promote repair of a DSB or a double-strand gap in its DNA gene via NHEJ or MMEJ, independently from DNA synthesis. The results demonstrate a role of transcript RNA in directing the way DSBs are repaired in DNA, suggesting that RNA may directly modulate genome stability and evolution.

Double-strand breaks (DSBs) in DNA are among the most difficult lesions to repair. Cells use three main DSB-repair mechanisms: non-homologous end joining (NHEJ), homologous recombination (HR), and microhomology-mediated end joining (MMEJ), with a preference for NHEJ[1,2]. In contrast to HR, NHEJ and MMEJ do not utilize a DNA template molecule to recover damaged and/or lost nucleotides[2]. NHEJ directly ligates broken DNA ends, while MMEJ exploits the alignment of short microhomologies on the DSB sides and is associated with deletions of the sequence between the microhomologies[3,4].

RNA, transcribed from DNA as a complementary single strand copy, is a multifunctional nucleic acid that has the primary role of messenger RNA (mRNA) to convert the information stored in DNA into proteins via translation. Despite a large fraction of the eukaryotic genome is transcribed into RNA, e.g., about 70–90% of the human[5] and 85–90% of the yeast genome[6], much remains unknown about RNA functions in cells[7]. Moreover, it is still unclear whether and how RNA plays a direct role in DNA repair. However, recent studies over the last decade have provided emerging evidence for RNA's more direct involvement in DSB repair.

In budding yeast, an endogenous RNA transcript can be used as a direct homologous template for accurate DSB repair of its DNA gene in *cis* via RNA-templated DSB repair (R-TDR), or in *trans* after reverse transcription of the RNA into a DNA copy (cDNA) via c-TDR[8,9]. R-TDR is blocked by ribonucleases (RNases) H1 and H2, which cleave RNA in a

[1]School of Biological Sciences, Georgia Institute of Technology, Atlanta, GA, USA. [2]Department of Mathematics and Statistics, University of South Florida, Tampa, FL, USA. [3]Department of Biochemistry and Molecular Biology, Tulane Cancer Center, Tulane University of Medicine, New Orleans, LA, USA. [4]Present address: Molecular Targets Program, Center for Cancer Research (CCR), National Cancer Institute (NCI), National Institutes of Health (NIH), Fredrick, MD, USA. [5]Present address: Department of Mathematics, University of Manitoba, Winnipeg, MB, Canada. [6]Present address: Columbia University Irving Medical Center, New York, NY, USA. [7]Present address: Program for Lung and Vascular Biology, Section for Injury Repair and Regeneration Research, Stanley Manne Children's Research Institute, Ann & Robert H. Lurie Children's Hospital of Chicago, Chicago, IL, USA. [8]Present address: Department of Pediatrics, Division of Critical Care, Northwestern University Feinberg School of Medicine, Chicago, IL, USA. [9]Present address: Emory University, Atlanta, GA, USA. [10]These authors contributed equally: Yilin Lu, Margherita Maria Ferrari, Tejasvi Channagiri, Penghao Xu. ✉e-mail: jonoska@usf.edu; storici@gatech.edu

hybrid with DNA[8,9]. Nevertheless, such RNase H inhibition of R-TDR highlights the marked capacity of RNA to form RNA-DNA hybrids at a DSB site. Several studies have identified different, non-templating roles of transcript RNA in promoting DSB repair of transcribed DNA by HR in mammalian cells[10–12]. Beyond HR, NHEJ-related proteins have been found to form a multiprotein complex with RNA polymerase II and to be associated with transcribed genes after inducing a DSB in these DNA loci, suggesting that RNA may help error-free NHEJ in human cells[13,14]. The chromatin context was shown to have an impact on DNA DSB repair induced by the clustered regularly interspaced short palindromic repeats (CRISPR) associated protein 9 (Cas9). In fact, it was recently found that NHEJ is broadly biased toward euchromatin in human cells, while MMEJ has a higher contribution in the heterochromatin context[15]. Furthermore, there is mounting evidence that RNA-DNA hybrids form at DNA DSBs from pre-transcribed RNA in transcriptionally active loci[16]. However, it remains uncharted whether RNA has the capacity to directly participate in the DSB-repair mechanisms via end joining and affect the frequency of the DSB-repair products in a sequence-dependent manner. Here, we find that both sense and antisense-transcript RNAs affect DSB repair in a sequence-specific manner in wild-type human and yeast cells.

## Results

To determine whether a transcript RNA plays a direct role in DSB repair via end joining, we developed an assay in human cells employing various RNA transcripts that differ by sequence and transcription level and are generated from a constitutively transcribed gene, in which we induced one or two DSBs by the Cas9 endonuclease. The constructs were put on plasmid DNA to facilitate the engineering and maintain a controlled (less affected by genome mutations and/or rearrangements) and exportable (allowing to introduce and study the constructs in cells of different genotypes) system. We engineered a plasmid to contain the *DsRed* gene with two exons (Exon1 and Exon2) and an artificial intron (Intron) in between. The *DsRed* gene was transcribed from the strong constitutive CMV promoter and carried the SV40 origin for DNA replication in the human embryonic kidney cells expressing the T antigen (HEK-293T). This construct is called Sense (Fig. 1a and Supplementary Fig. 1). From the Sense, we made a second construct, called BranchΔ, in which we deleted the branch region of the intron to prevent splicing of the intron, and a third construct, called pCMVΔ, in which we removed the CMV promoter to minimize transcription of the *DsRed* gene while still allowing intron splicing (Fig. 1a and Supplementary Figs. 1 and 2a, b). In all three constructs, we induced a DSB in the *DsRed* gene on either side of the intron, or on both sides using two single guide RNAs (sgRNAs) with Cas9 endonuclease using the same sgRNAs cutting in the same sites on these three constructs. In all constructs, the sgRNA A binds near the junction of the 5′ exon, Exon1, and the intron, and the sgRNA B binds near the junction of the intron and the 3′ exon, Exon2 (Fig. 1a). We verified that Cas9 with sgRNA A, sgRNA B, or both sgRNAs A and B generate DSBs at predesigned sites in the constructs carrying the intact intron (Sense and pCMVΔ) as well as in the construct with branch deletion (BranchΔ) with similar efficiency. This was evaluated by performing in vitro cleavage of two PCR products by Cas9 and sgRNA A and/or sgRNA B, with one containing the branch region of the intron like in the Sense and pCMVΔ, and one lacking it like in the BranchΔ. The results showed that Cas9 with sgRNA A and/or sgRNA B cleaved the two different DNA substrates equally well (Supplementary Fig. 3a). The Sense, BranchΔ, and pCMVΔ plasmid constructs were assayed together in the same experiment employing the same experimental procedures and conditions. Specifically, individually, the Sense, BranchΔ, and pCMVΔ constructs were transfected four independent times each into cells of the same culture of HEK-293T wild-type cells, as well as HEK-293T knock-out cells having mutations in the catalytic subunit of RNase H2 (RNase H2A KO) (Supplementary Fig. 4). Together with each *DsRed*

construct, we transfected the same plasmid expressing Cas9 and the same plasmid producing sgRNA A or sgRNA B to generate 1 DSB, or both plasmids for sgRNA A and B to generate a double-strand gap (2 DSBs) (Fig. 1a, b). By the time Cas9 has been transcribed, translated, and imported into the nucleus to cut the target site(s) together with the sgRNA(s), it is reasonable to expect that there is already transcript from the *DsRed* gene on the different constructs. As a No-DSB control, each plasmid was also co-transfected in the wild-type or RNase H2A KO cells with sgRNA A and B, but without Cas9. In addition, as discussed below, a sequence 30 bp distant from the DSB site was also used for no-DSB control. After a few days, the plasmid DNAs of the three constructs were extracted from the cells at the same time and prepared for next generation sequencing (NGS) to study the sequence around the DSB site or the double-strand gap (see "Methods", and Supplementary Data 1 and 2). The sequences of the reads from each sample library were then analyzed for specific signatures to categorize them as either DSB repair by NHEJ (small in/dels), or MMEJ (deletions between microhomologies) (Fig. 1b). Then the frequency of each of these two repair mechanisms was calculated (see "Methods").

### The sequence of a transcript RNA guides DSB repair in its DNA by NHEJ

We first examined the sequencing data obtained from the induction of one DSB by sgRNA A or B in the Sense, BranchΔ, and pCMVΔ constructs transfected in the RNase H2 wild-type and RNase H2A KO HEK-293T cells. While it was not possible to distinguish the sequence of error-free repair by NHEJ from that of the uncut constructs and from constructs with error free recombination between a cut and an uncut plasmid, we searched for in/dels near the DSB site as the signature for NHEJ (see "Methods"). The in/del signature was practically absent in the No-DSB controls, as well as in the control sequences 30 bp downstream from the DSB sites (Supplementary Fig. 5a, b). NHEJ was prevalent compared to MMEJ in all constructs (Supplementary Fig. 6a, b). However, NHEJ was particularly dominant over MMEJ in the construct without splicing, BranchΔ, compared to the constructs with splicing, Sense and pCMVΔ (Supplementary Fig. 6a, b). The analysis of the sequencing data revealed that the construct without splicing, BranchΔ, had higher frequency of NHEJ in/dels than the constructs with splicing (Sense and pCMVΔ) in wild-type and more evidently in the RNase H2A KO cells (Fig. 2a and Supplementary Data 3). We then performed an analysis of the NHEJ in/del variation observed in the samples. Using the NGS data, we aligned the sequencing reads to an error-free end-joining reference sequence, obtained the 20 nucleotides of the alignment around the DSB site (called the *DSB-sequence window*), and determined the type and number of variations of each read sequence compared to the reference sequence to generate *variation-distance graphs* for each of the constructs (see Methods). The results showed that the three constructs displayed a very similar pattern of in/dels (Supplementary Figs. 7 and 8a, c and Fig. 2b), suggesting a similar mechanism of DSB repair. Notably, when we compared Sense vs. BranchΔ, the comparison graphs revealed that the most prevalent in/dels were insertions in sgRNA A-treated samples and deletions in sgRNA B-treated samples, and these in/dels usually had higher frequency for the BranchΔ construct (Fig. 2c). Moreover, in RNase H2A KO cells, the majority of in/dels had higher frequency for the BranchΔ compared to the Sense construct (Fig. 2c and Supplementary Fig. 8b, d). These results provide evidence that the presence or absence of the intron region in the transcript RNA, which affects the transcript's complementarity to the DSB ends, can directly influence the frequency of the repaired products by NHEJ.

### The sequence of a transcript RNA guides DSB repair in its DNA by MMEJ

Seeking MMEJ-signature sequences and their frequencies, we analyzed the sequencing data obtained from DSB induction by Cas9 and sgRNA

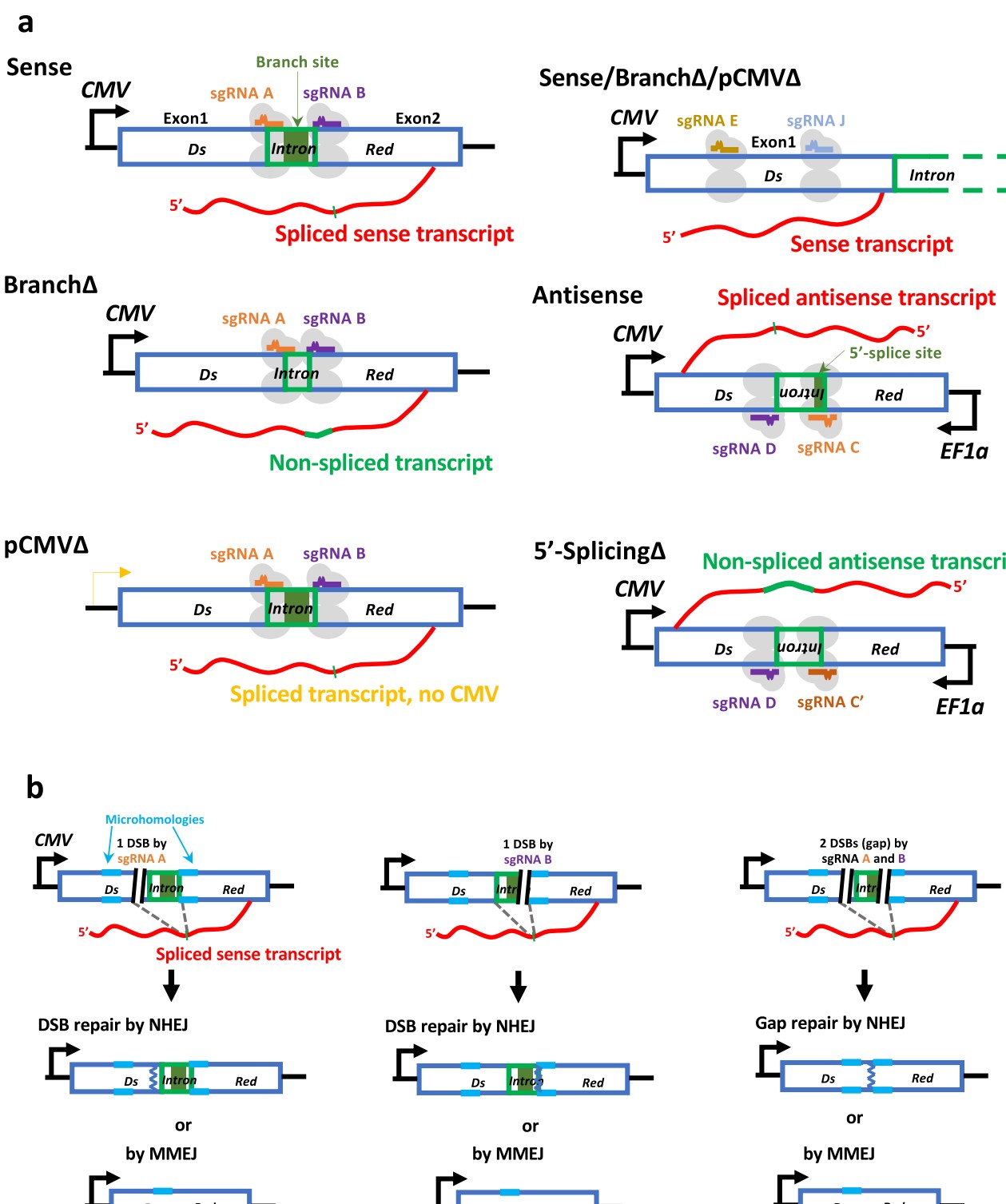

**Fig. 1 | Assay to study RNA-mediated DSB repair in human cells. a** Schemes of sense and antisense-genetic constructs expressing RNA transcripts that differ by sequence and transcription level. (left) Sense constructs with sgRNA A and B: Sense, BranchΔ, and pCMVΔ (right) Sense constructs with sgRNA E and J: Sense/ BranchΔ/ pCMVΔ, Antisense constructs with sgRNA C(C') and D: Antisense and 5'-SplicingΔ. All these constructs contain the *DsRed* gene (blue-framed box) with an intron (green-framed box) in the sense or antisense orientation, respectively. The sense and antisense transcript RNAs (in red) are depicted after intron splicing (thin green mark) or carrying the intron (thick green line) for the splicing-mutant constructs. The single-guide RNA (sgRNA) A (orange line), sgRNA B (purple line), sgRNA E (brown line) and sgRNA J (blue line) with Cas9 endonuclease (light gray ovals) bind to the complementary DNA of the sense constructs to generate a DSB. The sgRNA C (orange line) or C' (dark orange line) and D (purple line) with Cas9 endonuclease bind to the complementary DNA of the antisense constructs to generate a DSB. Black arrows: *CMV*, cytomegalovirus promoter, or *EF1α*, human eukaryotic translation elongation factor 1 alpha promoter. Transcription activity by a cryptic promoter is indicated with a yellow arrow. Dark green box: branch site of the intron or 5'-splice site for the sense and antisense constructs, respectively. **b** Scheme of DNA products for the Sense construct obtained following DSB repair by different DSB repair mechanisms. NHEJ non-homologous end joining, MMEJ microhomology-mediated end joining, DSB (black parallel lines); an example of a microhomology pair (light blue lines); DNA repaired by NHEJ (blue zigzag line).

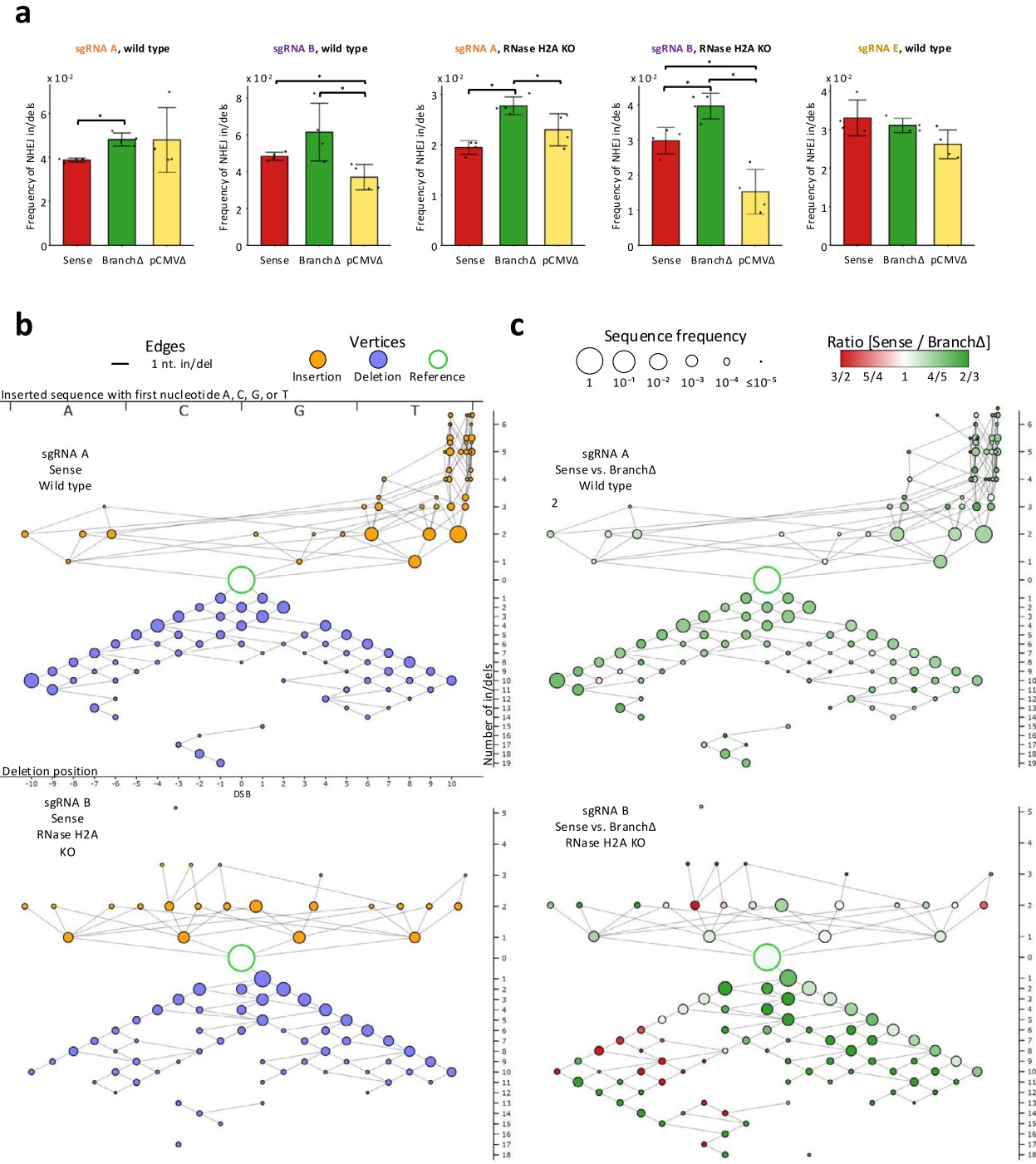

**Fig. 2 | Transcript RNA promotes DSB repair by NHEJ in a sequence-dependent manner. a** Frequencies of NHEJ in/dels observed following a DSB by the sgRNA A, B or E in the Sense (red), BranchΔ (green), and pCMVΔ (yellow) constructs of wild-type and RNase H2A KO cells. Plotted data are the mean ± s.d. of the 4 biological replicates with the individual values shown as dots; $N = 4$. *$p$ value = 0.029 (two-tailed Mann−Whitney $U$ test). Source data are provided as a Source Data file. **b** Individual variation-distance graphs illustrating sequence variations within DSB-sequence windows obtained after DSB induction by sgRNA A (top) or sgRNA B (bottom) in the Sense construct of wild-type cells. Each vertex represents a single DSB-sequence window. An edge between two vertices indicates that the two corresponding DSB-sequence windows differ by a single nucleotide in/del. Insertion vertices (in orange) are placed above the reference vertex (in white with a green outline), while deletion vertices (in blue) are placed below it. The vertex size shows the log of the mean frequency of the corresponding DSB-sequence window in the four repeats of the experiment. For insertions, the alphabetical order of the

inserted sequences, from A on the left to T on the right, are indicated by the $x$-coordinate. Insertions of size 3 or more have vertices on multiple lines, staggered vertically to reduce overlap. The $x$-coordinate of deletions indicates the position of the first deleted nucleotide, from the most upstream (left-most) to the most downstream (right-most). The $y$-coordinate indicates the number of variations in the DSB-sequence windows, with higher variations placed further from the reference. See Supplementary Fig. 7 for the variation-distance graph key. **c** Comparison variation-distance graphs of the DSB-sequence windows obtained after DSB induction by sgRNA A (top) or sgRNA B (bottom) for the Sense vs. the BranchΔ construct of wild-type cells. The vertices represent the same DSB-sequence windows as for the individual graphs while the vertex colors specify the relative frequency in the Sense (red) vs. the BranchΔ (green) construct; the vertex sizes show the log of the higher of the two mean frequencies of the corresponding DSB-sequence windows in the two compared constructs.

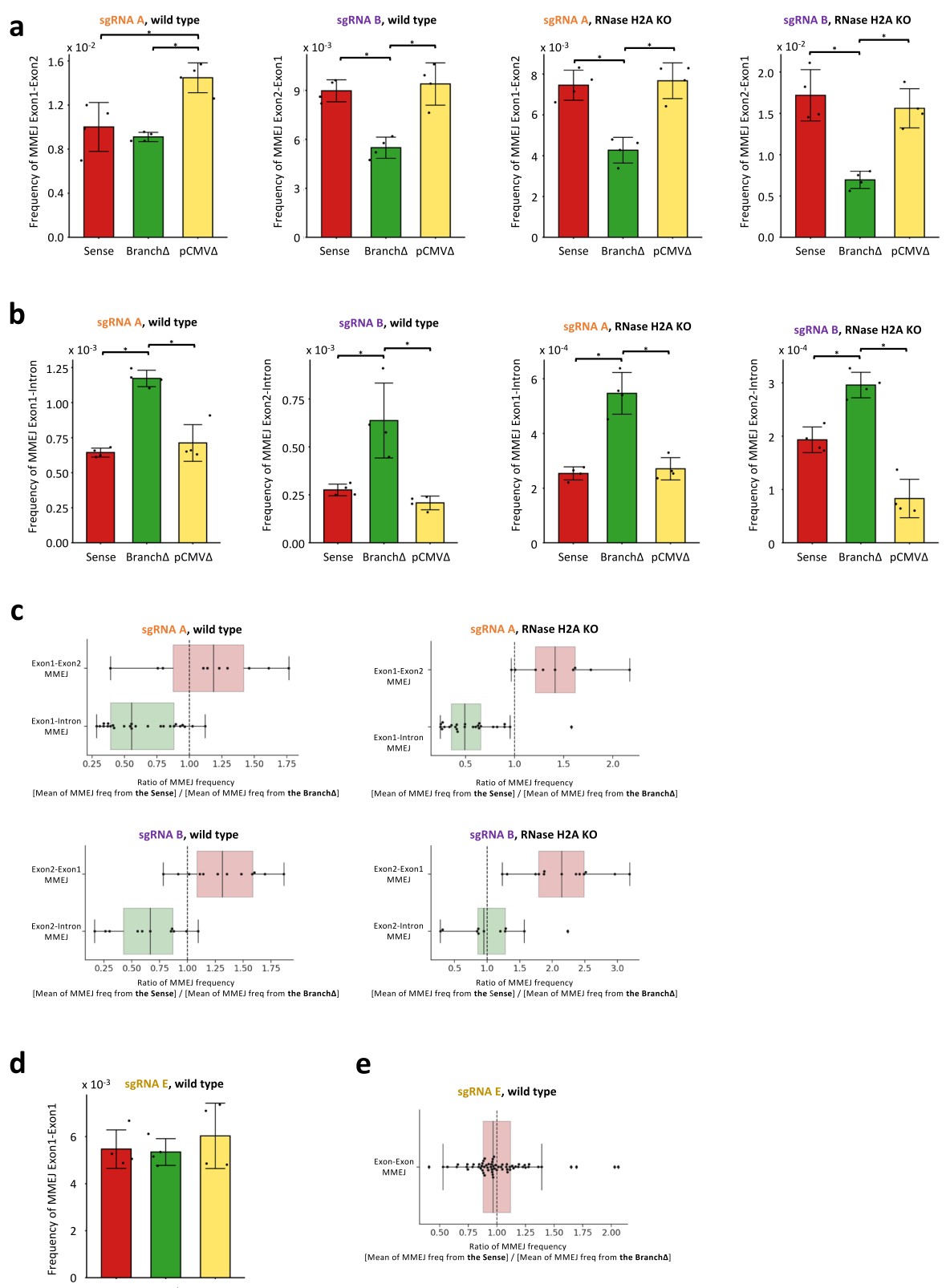

A or B in the Sense, BranchΔ, and pCMVΔ constructs transfected in the wild-type as well as in the RNase H2A KO HEK-293T cells. We identified all possible microhomology pairs of three base pairs or longer in the *DsRed* exon and intron sequences in common between the splicing and non-splicing constructs (Supplementary Fig. 9a–c). We then determined the deletion signature for each microhomology pair and

identified the signatures that were detected in all the NGS libraries of the Sense, BranchΔ, and pCMVΔ constructs for the repair of the DSB generated by sgRNA A or B in the wild-type or RNase H2A KO HEK-293T cells (see "Methods"). The results of DSB repair by MMEJ for the exon-exon microhomology pairs (those between the two exons: Exon1-Exon2 for cleavage by sgRNA A and Exon2-Exon1 for cleavage by

**Fig. 3 | Transcript RNA promotes DSB repair by MMEJ in a sequence-dependent manner. a** Sum of MMEJ frequencies from all microhomology pairs of exon-exon or (**b**) exon-intron MMEJ following a DSB by the sgRNA A (see Supplementary Fig. 9b, e or b, d) or B (see Supplementary Fig. 9c, e or c, d) in the Sense (red), BranchΔ (green), and pCMVΔ (yellow) constructs of wild-type (left) and RNase H2A KO (right) cells. Plotted data are the mean ± s.d. of the 4 biological replicates with the individual values shown as dots; *N* = 4. *$p$ value = 0.029 (two-tailed Mann–Whitney *U* test). **c** Boxplot showing the ratios of MMEJ frequencies from exon-exon (red box) and exon-intron (green box) microhomologies between the Sense and the BranchΔ constructs of wild-type (left) and RNase H2A KO (right) cells after DSB by sgRNA A (top) or B (bottom). The numerator and denominator of each ratio was calculated by using an average of 4 repeats of MMEJ frequencies from each construct. Ten Exon1-Exon2 and 25 Exon1-Intron microhomologies (black dots) are shown for MMEJ following a DSB by the sgRNA A (top). Thirteen Exon2-Exon1 and 11 Exon2-

Intron microhomologies are shown for MMEJ following a DSB by the sgRNA B (bottom). The median of the points is shown as the middle line of the box. The first and third quartiles are indicated by the box frames and the whiskers represent the largest point not more than 1.5 interquartile range (IQR) beyond the box frame. All data points outside the whiskers are classified as outliers and shown as diamond points. **d** Sum of MMEJ frequencies from all microhomology pairs of exon-exon MMEJ following a DSB by the sgRNA E (see Supplementary Fig. 9d, f). Plotted data are the mean ± s.d. of the 4 biological replicates with the individual values shown as dots; *N* = 4. **e** Boxplot showing the ratios of MMEJ frequencies from exon-exon microhomologies between the Sense and the BranchΔ constructs of wild-type cells after DSB by sgRNA E. Sixty-four Exon1-Exon1 microhomologies (black dots) are shown for MMEJ following a DSB by the sgRNA E. Details as in (**c**). Source data are provided as a Source Data file.

sgRNA B) were opposite to those of exon-intron microhomology pairs (those between an exon and intron: Exon1-Intron for cleavage by sgRNA A, and Exon2-Intron for cleavage by sgRNA B). The exon-exon MMEJ had higher frequency of DSB repair for the constructs with splicing (Sense and pCMVΔ) than for the construct without splicing (BranchΔ) (Fig. 3a, and results with individual-microhomology pairs in Supplementary Fig. 9e), while the exon-intron MMEJ had higher frequency of DSB repair in the construct without splicing than for those with splicing (Fig. 3b, and results with individual-microhomology pairs in Supplementary Fig. 9e). This higher exon-exon MMEJ frequency in the splicing constructs partially compensates for the higher frequency of NHEJ in the non-splicing construct (Fig. 3a and Supplementary Fig. 6).

Next, for each microhomology pair, we calculated the ratio of the mean frequency in the Sense construct to the mean frequency in the BranchΔ construct. As shown in Fig. 3c, the Sense/BranchΔ ratios for the exon-exon microhomologies were mostly higher than 1, while the Sense/BranchΔ ratios for the exon-intron microhomology pairs were mostly lower than 1. Also, the ratio of the total exon-exon frequencies to the total exon-intron frequencies for MMEJ within the Sense and pCMVΔ libraries was significantly higher than that obtained for the BranchΔ libraries thus suggesting more efficient MMEJ between exon sequences for the constructs with splicing compared to the construct without splicing (Supplementary Fig. 9g). These results support the role of spliced RNA in promoting MMEJ between exon-exon microhomologies, and the role of the non-spliced RNA in promoting MMEJ between exon-intron microhomologies. Following DSB by sgRNA A or B, the distance between the two exons is shorter in the BranchΔ construct compared to the Sense or the pCMVΔ construct because the intron lacks the branch site. This shorter distance should facilitate MMEJ between exon-exon microhomologies in the BranchΔ construct, but we consistently observe the opposite result. Moreover, for the DSB by sgRNA A the distance between the Exon1-Intron microhomologies is the same for the Sense, pCMVΔ, and BranchΔ constructs (see Supplementary Fig. 9b), yet Exon1-Intron MMEJ is more efficient for the BranchΔ construct. These results argue against biased Cas9 cleavage of the different constructs and support the role of transcript RNA in guiding MMEJ. Notably, the results for DSB repair by NHEJ (Fig. 2a) parallel those of DSB repair by exon-intron microhomologies (higher frequency in the construct with non-spliced RNA than in those with spliced RNA) but are opposite to those of DSB repair by exon-exon microhomologies (Fig. 3a, b). Also, the results obtained from the RNase H2A KO cells were stronger than those obtained from the wild-type HEK-293T cells, especially for repair of the DSB caused by sgRNA B (Fig. 3 and Supplementary Data 3), which suggests that the interactions of the transcript RNA with the broken DNA ends have greater stability when RNase H2 is not functional. Altogether, these findings support the capacity of RNA to promote DSB repair via MMEJ in a sequence-specific manner.

## A DSB in the exon sequence is repaired with similar end-joining frequency in the splicing and non-splicing constructs

If the spliced RNA promotes MMEJ between exon-exon microhomologies, and the non-spliced RNA promotes NHEJ, as well as MMEJ between exon-intron microhomologies when a DSB is generated close to an exon-intron junction (by sgRNA A or sgRNA B), we expect that when a DSB is generated within the exon sequence both the spliced or non-spliced RNAs should impact NHEJ and MMEJ in a similar manner. Indeed, in stark contrast to results obtained generating a DSB at an intron exon junction by sgRNA A or B, a DSB generated within Exon1, using sgRNA E (Fig. 1a and Supplementary Figs. 1 and 9a, d) was repaired via NHEJ or MMEJ with similar frequencies among the Sense, BranchΔ, and pCMVΔ constructs (Figs. 2a and 3d and Supplementary Figs. 6c, d and 9f). Both the splicing (Sense and pCMVΔ) and the non-splicing (BranchΔ) constructs repaired the DSB with similar NHEJ and MMEJ frequencies. For each microhomology pair, we then calculated the ratio of the mean frequency in the Sense construct to the mean frequency in the BranchΔ construct. Differently from results for repair of a DSB by sgRNA A or B shown in Fig. 3c, the Sense/BranchΔ ratios for the exon-exon microhomologies within Exon1 were very close to 1 (Fig. 3e).

## The sequence of a transcript RNA guides double-strand gap repair in its DNA by NHEJ and MMEJ

To investigate whether RNA mediates the repair of a double-strand gap in DNA, we generated a DSB on each side of the intron in the *DsRed* gene by using both sgRNA A and B to cut the Sense, BranchΔ, or pCMVΔ constructs (Fig. 1b) in both wild-type and RNase H2A KO HEK-293T cells. The NGS data from the Sense, BranchΔ, and pCMVΔ constructs were analyzed to determine the frequency of NHEJ signatures with intron pop-out and MMEJ signatures with intron pop-out by recombination between exon-exon microhomologies (see "Methods"). We then compared these NHEJ and MMEJ frequencies of intron pop-out with those obtained for the signatures of intron retention in the Sense, BranchΔ or pCMVΔ constructs. We found that the constructs with splicing (Sense and pCMVΔ) had a higher frequency of intron pop-out than the BranchΔ construct (percentages in bold in Fig. 4a). The individual frequencies of intron pop-out by NHEJ were also higher for the splicing constructs compared to the BranchΔ construct (Fig. 4a). The results were stronger in the RNase H2A KO cells, suggesting direct RNA-DNA interaction (Fig. 4a and Supplementary Data 3). Though the aim here was to study double-strand gap repair, it is possible to have a repair of one DSB before the other DSB occurs by using both sgRNA A and B. However, when we generated a single DSB by either sgRNA A or B, only a small fraction of the reads had lost the intron, and these were mainly exon-exon MMEJ sequences (Supplementary Fig. 6). Thus, such products of intron pop-out after repair of one DSB before the other occurs were too low to influence the results of double-strand gap repair and would anyway be additional evidence that the spliced transcript promotes intron pop-out. Moreover, once the double-

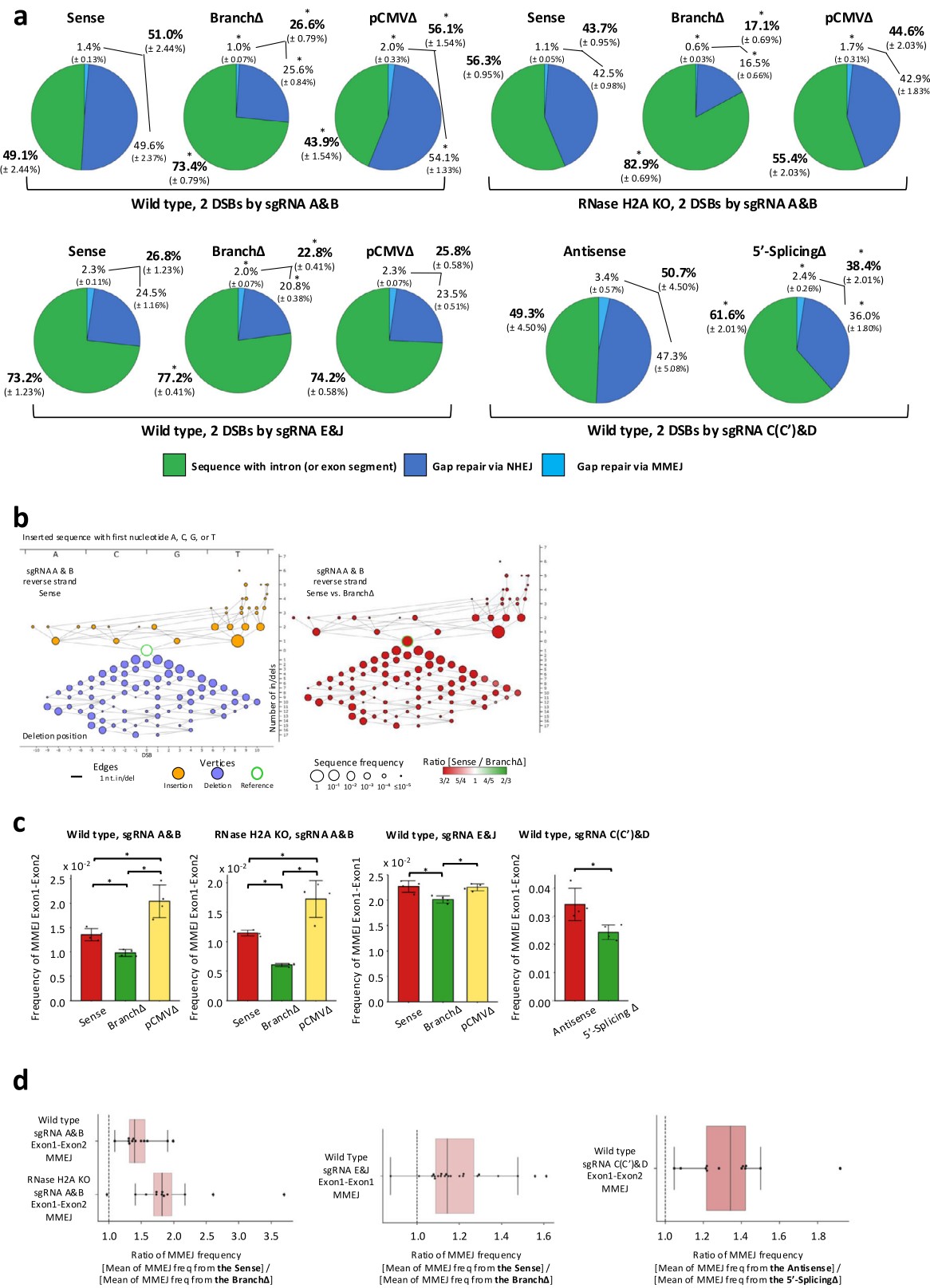

**a** — Pie charts showing repair outcomes. Legend: green = Sequence with intron (or exon segment); dark blue = Gap repair via NHEJ; light blue = Gap repair via MMEJ.

Top left group — Wild type, 2 DSBs by sgRNA A&B: Sense (1.4% ±0.13%, 51.0% ±2.44%, 49.6% ±2.37%, 49.1% ±2.44%); BranchΔ (1.0% ±0.07%, 26.6% ±0.79%, 25.6% ±0.84%, 73.4% ±0.79%); pCMVΔ (2.0% ±0.33%, 56.1% ±1.54%, 54.1% ±1.33%, 43.9% ±1.54%).

Top right group — RNase H2A KO, 2 DSBs by sgRNA A&B: Sense (1.1% ±0.05%, 43.7% ±0.95%, 42.5% ±0.98%, 56.3% ±0.95%); BranchΔ (0.6% ±0.03%, 17.1% ±0.69%, 16.5% ±0.66%, 82.9% ±0.69%); pCMVΔ (1.7% ±0.31%, 44.6% ±2.03%, 42.9% ±1.83%, 55.4% ±2.03%).

Bottom left group — Wild type, 2 DSBs by sgRNA E&J: Sense (2.3% ±0.11%, 26.8% ±1.23%, 24.5% ±1.16%, 73.2% ±1.23%); BranchΔ (2.0% ±0.07%, 22.8% ±0.41%, 20.8% ±0.38%, 77.2% ±0.41%); pCMVΔ (2.3% ±0.07%, 25.8% ±0.58%, 23.5% ±0.51%, 74.2% ±0.58%).

Bottom right group — Wild type, 2 DSBs by sgRNA C(C')&D: Antisense (3.4% ±0.57%, 50.7% ±4.50%, 47.3% ±5.08%, 49.3% ±4.50%); 5'-SplicingΔ (2.4% ±0.26%, 38.4% ±2.01%, 36.0% ±1.80%, 61.6% ±2.01%).

**b** — Variation-distance graphs. Inserted sequence with first nucleotide A, C, G, or T. Edges: 1 nt in/del. Vertices: Insertion (orange), Deletion (blue), Reference (green). Sequence frequency: 1, 10⁻¹, 10⁻², 10⁻³, 10⁻⁴, ≤10⁻⁵. Ratio [Sense / BranchΔ]: 3/2, 5/4, 1, 4/5, 2/3.

**c** — Bar graphs of Frequency of MMEJ Exon1-Exon2 (and Exon1-Exon1): Wild type sgRNA A&B; RNase H2A KO sgRNA A&B; Wild type sgRNA E&J; Wild type sgRNA C(C')&D.

**d** — Box plots of Ratio of MMEJ frequency.

strand gap has been repaired by the DSB repair pathways there cannot be more cleavage by Cas9 because the sgRNA binding sites are no longer present (Supplementary Fig. 1).

We then performed an analysis of the in/del variation observed in the sequencing reads that had double-strand gap repair by NHEJ. Using the NGS data, we aligned the sequencing reads to an error-free end-joining reference sequence with the gap removed, obtained the 20 nucleotides of the alignment around the double-strand gap site, and determined the type and number of variations of each read sequence compared to the reference sequence to generate variation-distance graphs (see Methods). The results showed that the Sense, BranchΔ, and pCMVΔ constructs displayed a similar pattern of NHEJ in/dels (Supplementary Fig. 10a and Fig. 4b), suggesting a similar mechanism of DSB repair. However, the comparison graphs, showing the relative

**Fig. 4 | Transcript RNA promotes double-strand gap repair in a sequence-dependent manner via NHEJ and MMEJ. a** Pie charts showing frequencies of sequencing reads displaying intron retention or pop-out following a double-strand gap by the sgRNAs A and B or sgRNAs E and J in the Sense, BranchΔ, and pCMVΔ constructs of wild-type and RNase H2A KO cells, or a double-strand gap by the sgRNAs C/C' and D in the Antisense and 5'-SplicingΔ constructs of wild-type cells. Percentages represent an average of 4 repeats with standard deviation in parenthesis; *N* = 4. The percentages of sequences with and without intron are bolded. *p* = 0.029 comparing frequencies of the BranchΔ or the pCMVΔ with those of the Sense construct, or comparing frequencies of the Antisense with those of the 5'-SplicingΔ construct via the two-tailed Mann–Whitney *U* test. **b** Individual (left) and comparison (right) variation-distance graphs illustrating sequence variations for the indicated samples. Refer to Fig. 2 and Supplementary Fig. 7 for details. The sequences are reverse-complemented prior to computing *xy*-coordinates so that they correspond to the forward-strand sequence coordinates. **c** Sum of MMEJ

frequencies from all microhomology pairs of exon-exon MMEJ (sense constructs, see Supplementary Fig. 11b, c, e, f, antisense constructs, see Supplementary Fig. 11d, g), following a double-strand gap by the sgRNAs A and B or sgRNAs E and J in the Sense (red), BranchΔ (green), and pCMVΔ (yellow) constructs of wild-type and RNase H2A KO cells, and the sgRNAs C/C' and D in the Antisense (red) and the 5'-SplicingΔ (green) constructs of wild-type cells. Plotted data are the mean ± s.d. of the 4 biological replicates with the individual values shown as dots; *N* = 4. *p* = 0.029 (two-tailed Mann–Whitney *U* test). **d** Boxplot showing specific ratios of MMEJ frequencies. Twelve Exon1-Exon2 microhomologies (black dots) are shown for MMEJ following the double-strand gap in the sense constructs generated by the sgRNAs A and B. Twenty-two Exon1-Exon1 microhomologies (black dots) are shown for MMEJ following the double-strand gap in the sense constructs generated by the sgRNAs E and J. Ten Exon1-Exon2 microhomologies (black dots) are shown for MMEJ following the double-strand gap in the antisense constructs generated by the sgRNA C/C' and D. For details see Fig. 3c legend. Source data are provided as a Source Data file.

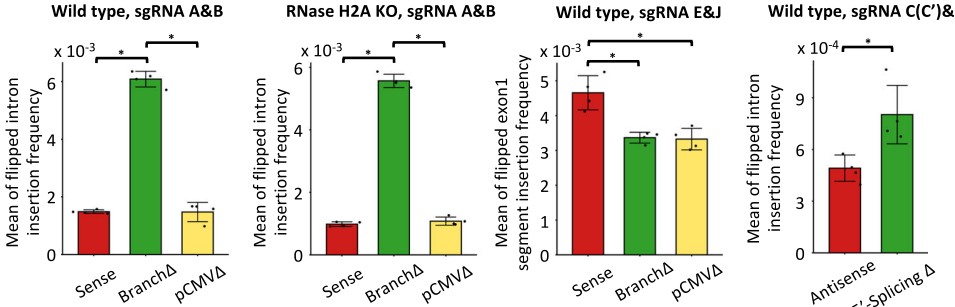

**Fig. 5 | Transcript RNA promotes intron flipping in double-strand gap repair.** Frequency of intron flipping caused by re-capture of the intron (or exon segment for sgRNAs E&J) via NHEJ following a double-strand gap by the sgRNAs A and B or sgRNAs E and J in the Sense (red), BranchΔ (green), and pCMVΔ (yellow) constructs of wild-type and RNase H2A KO cells, and the sgRNAs C/C' and D in the Antisense

(red) and the 5'-SplicingΔ (green) constructs of wild-type cells. Plotted data are the mean ± s.d. of the 4 biological replicates with the individual values shown as dots; *N* = 4. *p* = 0.029 (two-tailed Mann–Whitney *U* test). Source data are provided as a Source Data file.

frequencies of in/dels in the Sense vs. the BranchΔ construct, revealed a higher frequency of both insertions and deletions by NHEJ in the Sense construct (Fig. 4b and Supplementary Fig. 10b). The comparison graphs of the Sense vs. the pCMVΔ construct showed higher frequencies of insertions and deletions for the pCMVΔ construct (Supplementary Fig. 10b), as also revealed by results shown in Fig. 4a. Results in RNase H2A KO cells corroborated those obtained in the RNase H2 wild-type cells (Supplementary Fig. 10c, d). The pop-out of the intron from the *DsRed* gene following induction of the two DSBs near the intron-exon junctions was more frequent in the splicing constructs than in the non-splicing construct also for double-strand gap repair via MMEJ for exon-exon microhomologies (Fig. 4a, c, d and Supplementary Fig. 11a, b, e). The results were somewhat stronger in the RNase H2A KO compared to the wild-type cells (Fig. 4a, c, d, Supplementary Fig. 11e and Supplementary Data 3).

Opposite results were obtained for the insertion of the intron in the flipped orientation via NHEJ. The frequency of the flipped-intron insertion was a factor of three higher for the non-splicing than for the splicing constructs both in wild-type and RNase H2A KO cells (Fig. 5). These results, for the repair of the double-strand gap generated by two DSBs near the intron-exon junctions, point towards a role of the spliced RNA in promoting intron exclusion and thus, in opposing not only intron retention, but also intron flipping. Vice versa, the results highlight a role of the non-spliced RNA in promoting intron inclusion by maintaining the intron in its original locus or in facilitating its flipping. Interestingly, the frequency of double-strand gap repair with intron deletion both by NHEJ and MMEJ or with intron flipping for the pCMVΔ construct was similar to that of the Sense construct, in both wild-type and RNase H2A KO cells (Figs. 4a, c and 5), suggesting that the level of transcription is not a limiting factor for transcript RNA to mediate DNA double-strand gap repair via NHEJ or MMEJ.

## A double-strand gap within an exon sequence is repaired with similar end-joining frequency in the spliced and non-spliced constructs

We reasoned that if transcript RNA has a direct role in guiding end-joining repair, generation of a double-strand gap within the exon sequence should result in similar repair frequency of end joining between the splicing and non-splicing constructs because the exon sequence is retained both in the spliced and non-spliced RNAs. We generated two DSBs within the Exon1 in the *DsRed* gene by using both sgRNA E and J (Fig. 1a and Supplementary Fig. 11a, c) to cut the Sense, BranchΔ, or pCMVΔ constructs in wild-type HEK-293T cells. The NGS data from the Sense, BranchΔ, and pCMVΔ constructs were analyzed to determine the frequency of NHEJ signatures with segment pop-out and MMEJ signatures with segment pop-out by recombination between microhomologies flanking the gap within Exon1 (see "Methods"). We then compared these NHEJ and MMEJ frequencies of segment pop-out with those obtained for the signatures of segment retention in the Sense, BranchΔ, or pCMVΔ constructs. We found that differently from results of intron pop-out following a double-strand gap generated by sgRNA A and B, with a gap generated in Exon1, all constructs showed predominant capacity of segment retention vs. segment pop-out and had a more similar frequency of segment pop-out among each other (Fig. 4a, and see No-DSB controls shown in Supplementary Fig. 8a). The individual frequencies of segment pop-out by NHEJ and MMEJ were also much more similar between the splicing constructs and the BranchΔ construct (Fig. 4a, c, d and Supplementary Fig. 11f). Nonetheless, the segment pop-out frequencies were slightly higher for the splicing constructs compared to the non-splicing construct. We believe that this is because in the non-splicing construct the transcript RNA still retains longer complementarity with the broken

DNA than the spliced-transcript RNAs; thus, still in part facilitating segment retention. Different results from those obtained for the insertion of the flipped intron were also obtained for the insertion of the Exon1 segment in the flipped orientation via NHEJ. There was no lower frequency of flipped Exon1 segment for the splicing constructs construct compared to the BranchΔ construct. All constructs had a frequency higher than $3 \times 10^{-3}$, with the Sense construct displaying a slightly higher frequency than the other constructs (Fig. 5). Overall, these results for the repair of the double-strand gap generated by two DSBs within Exon1 support a direct role of transcript RNA in DSB repair and a role of RNA in retaining the integrity of its complementary DNA sequence upon DNA breakage.

## Overexpression of RNase H1 reduces the efficiency of RNA-mediated DSB repair

To highlight a role of transcript RNA in promoting double-strand gap repair by NHEJ and MMEJ for the Sense construct over the BranchΔ construct, we performed the double-strand gap repair assay in the presence of overexpressed RNase H1, which targets RNA-DNA hybrids independently from the cell cycle[17,18]. We co-transfected the wild-type HEK-293T cells with the Sense or BranchΔ construct together with the Cas9, sgRNA A, and sgRNA B plasmids, and with a plasmid expressing human RNase H1 or a mutant form of it as negative control (Fig. 6a), both under the CMV promoter. NGS libraries were prepared as described above and in the "Methods". The sequencing data from the Sense and BranchΔ constructs were analyzed to determine the

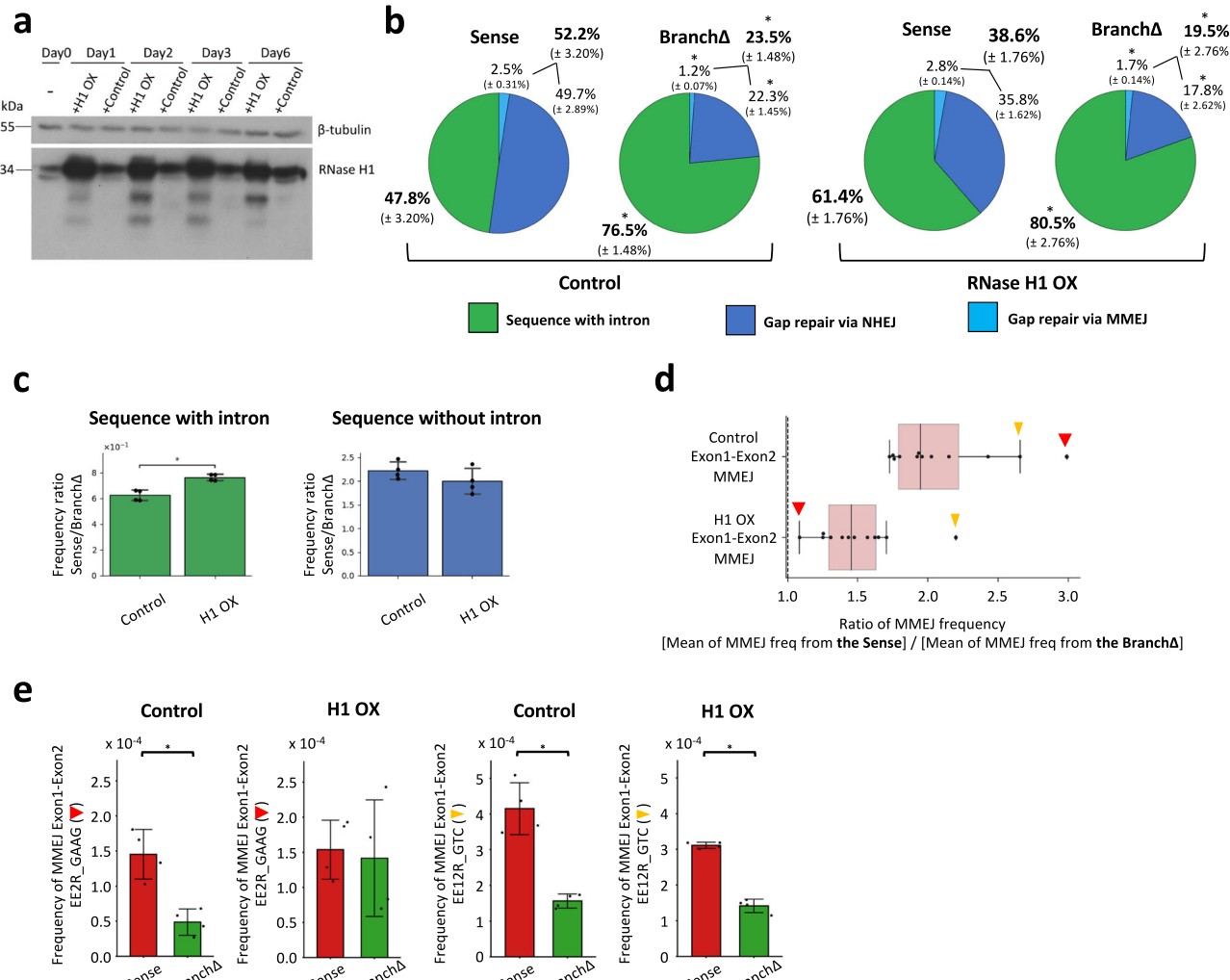

**Fig. 6 | Overexpression of RNase H1 represses RNA-mediated DNA double-strand gap repair. a** Result of Western blot for human RNase H1 overexpression (OX) in HEK-293T cells, $N = 2$. Control, overexpression of a non-functional mutant of RNase H1. **b** Pie charts showing frequencies of sequencing reads displaying intron retention or pop-out following a gap by the sgRNAs A and B in the Sense and BranchΔ constructs with Control (left) and with HI OX (right). Percentages represent an average of 4 repeats with standard deviation in parenthesis; $N = 4$. The percentages of sequences with and without intron are bolded. $*p = 0.029$ comparing frequencies of the BranchΔ with those of the Sense construct via the two-tailed Mann–Whitney $U$ test. **c** Sense/BranchΔ frequency ratios for two types of repaired sequences, with intron (green) and without intron (blue), detected in the sequencing libraries following a double-strand gap by the sgRNAs A and B in Control and H1 OX. **d** Boxplot showing the ratios of MMEJ frequencies between the Sense and the BranchΔ constructs of Control and H1 OX from Exon1-Exon2

microhomologies located upstream and downstream of the DNA gap. Twelve Exon1-Exon2 microhomologies (black dots) are shown for MMEJ following the DNA gap in the sense constructs generated by the sgRNA A and B. The thick red arrow indicates a ratio from the EE2R_GAAG microhomology (Supplementary Fig. 11b). The thin yellow arrow indicates a ratio from the EE12R_GTC microhomology (Supplementary Fig. 11b). For details see legend of Fig. 3c. **e** MMEJ frequencies from the outlier-data points presented in (**d**) derived from the exon-exon EE2R_GAAG microhomologies (thick, red arrows) that are the most distant from each other, and the EE12R_GTC microhomologies (thin, yellow arrows) that are the closest to each other (see Supplementary Figs. 11b and 12a), following a double-strand gap by the sgRNAs A and B in the Sense (red) and BranchΔ (green) constructs of Control and H1 OX. For (**c**) and (**e**), plotted data are the mean ± s.d. of the 4 biological replicates with the individual values shown as dots; $N = 4$. $*p = 0.029$ (two-tailed Mann–Whitney $U$ test). Source data are provided as a Source Data file.

frequency of NHEJ signatures with intron pop-out, and MMEJ signatures with intron pop-out by recombination between exon-exon microhomologies. We found that the construct with splicing (Sense) had a reduced frequency of intron pop-out when RNase H1 was overexpressed (percentages in bold in Fig. 6b, c and Supplementary Data 4), while the ratio of the intron retention frequencies of the Sense vs. the BranchΔ was closer to 1 when RNase H1 was overexpressed (Fig. 6c and Supplementary Data 4). In particular, the difference between the Sense and BranchΔ frequency of intron deletion via MMEJ was less prominent when RNase H1 was overexpressed (Fig. 6b), with the median Sense/BranchΔ ratio of individual MMEJ frequencies dropping from 1.95 to 1.45 (Fig. 6d). Interestingly, the outlier-data points in Fig. 6d, which represent ratios obtained from the EE2R_GAAG microhomologies (thick, red arrows) that are the most distant from each other, and from the EE12R_GTC microhomologies (thin, yellow arrows) that are the closest to each other (Supplementary Fig. 11b), highlight a major effect of RNase H1 on microhomologies that are distant from each other vs. those that are close to each other (Fig. 6d, e and Supplementary Fig. 12a). The frequency of the flipped intron insertion was in part reduced both for the Sense and the BranchΔ constructs when RNase H1 was overexpressed (Supplementary Fig. 12b). Overall, these results support a direct role for transcript RNA in DSB repair via end joining in human cells.

## An antisense transcript RNA guides double-strand gap repair in its DNA by NHEJ and MMEJ

To understand whether the capacity of RNA to mediate DSB repair in human cells could occur also with a non-coding antisense RNA, we engineered a plasmid construct, called Antisense, in which the *DsRed* gene was also transcribed in the antisense orientation from the constitutive EF1a promoter, and the intron sequence was reversed to allow splicing from the antisense transcript of the *DsRed* gene (Fig. 1a). We then engineered another plasmid, 5′-SplicingΔ, in which the 5′-splice site of the intron in the Antisense construct was deleted to block splicing (Fig. 1a and Supplementary Fig. 1). To investigate the capacity of the antisense RNA to mediate the repair of a double-strand gap in DNA, we generated a DSB on each side of the intron by using both sgRNA C and D to cut the Antisense construct, and C′ and D to cut the 5′-SplicingΔ construct (Fig. 1a and Supplementary Fig. 1). The respective sgRNAs cut the Antisense and 5′-SplicingΔ constructs similarly in vitro using Cas9 (Supplementary Fig. 3c). The Antisense and the 5′-SplicingΔ constructs were transfected into the wild-type HEK-293T cells together with the plasmid expressing Cas9 and the plasmids for the respective sgRNAs. After a few days, the plasmid DNAs were extracted from the cells and prepared for NGS to study the sequence of the *DsRed* gene around the DNA double-strand gap. The NGS data from the Antisense and the 5′-SplicingΔ constructs were analyzed to determine the frequency of signatures of NHEJ with intron deletion, and MMEJ with intron deletion by recombination between exon-exon microhomologies (see "Methods"). We then compared the frequencies of intron deletion obtained for the Antisense construct with those obtained for the 5′-SplicingΔ construct. Like the results for the Sense construct, we found that the Antisense construct with splicing had a higher frequency of intron loss than the 5′-SplicingΔ construct without splicing (Fig. 4a). Both the frequencies of intron deletion by NHEJ and those by MMEJ for the Antisense construct were higher than those obtained for the 5′-SplicingΔ construct (Fig. 4a, c, d and Supplementary Fig. 11a, d, g).

We then performed an analysis of the in/del variation observed in the sequencing reads that had double-strand gap repair by NHEJ. Using the NGS data, we aligned the sequencing reads to an error-free end-joining reference sequence with the gap removed, obtained the 20 nucleotides of the alignment around the double-strand gap site, and determined the type and number of variations of each read sequence compared to the reference sequence to generate variation-distance

graphs (see Methods). The results showed that the Antisense and the 5′-SplicingΔ constructs displayed a similar pattern of NHEJ in/dels (Supplementary Fig. 10e), suggesting a similar mechanism of DSB repair. However, the comparison graphs, showing the relative frequencies of in/dels in the Antisense vs. the 5′-SplicingΔ construct, revealed a higher frequency of both insertions and deletions by NHEJ in the Antisense construct (Supplementary Fig. 10f). The results were not as dominant for the Antisense construct as they were for the Sense construct (Supplementary Fig. 10d, left graphs), likely due to the impact of the sense RNA transcribed from the CMV promoter of the Antisense and the 5′-SplicingΔ constructs.

Next, we examined the frequency of the flipped-intron insertion after 2-DSB induction in the Antisense and the 5′-SplicingΔ constructs in the wild-type HEK-293T cells. In line with results obtained for the BranchΔ and the Sense constructs, we found that the flipped-intron insertion was more frequent in the 5′-SplicingΔ construct than in the Antisense construct (Fig. 5). Overall, the results obtained using the antisense constructs suggest that not just sense, but also antisense RNA has the capacity to promote double-strand gap repair by NHEJ or MMEJ in a sequence-dependent manner.

## The sequence of a transcript RNA guides DSB and double-strand gap repair in the corresponding DNA by NHEJ and MMEJ independently from DNA synthesis

The splicing and non-splicing constructs carry the SV40 origin of replication that is activated once transfected into the HEK293T cells because the cells express the T antigen[19]. To determine whether transcript RNA can directly affect the DSB repair outcomes independently from DNA synthesis in human cells, the Sense and the BranchΔ constructs were transfected into HEK293 cells, which do not express the T antigen, thus preventing plasmid replication. In the same experiment, the Sense and the BranchΔ constructs were transfected also into HEK293T cells, as reference. Splicing occurred from the Sense but not the BranchΔ construct in the HEK293 cells as expected (Supplementary Fig. 2c). By co-transfecting the Cas9 and the sgRNA A or both sgRNA A and B plasmids in the HEK293 and HEK293T cells, we generated one DSB or a double-strand gap, respectively, in the Sense and BranchΔ constructs (Fig. 1a and Supplementary Fig. 1). Because the plasmids do not replicate in the HEK293 cells, to prevent significant dilution of the Sense and the BranchΔ constructs, the plasmid DNA of the Sense and the BranchΔ constructs were extracted after three instead of six days following transfection (see "Methods"). The extracted DNAs were prepared for NGS to study the sequence of the *DsRed* gene around the DSB and the DNA double-strand gap and determine the of NHEJ and MMEJ. No-DSB controls, without the Cas9 plasmid, had more than 99% of sequences retaining the original intron both for the Sense and the BranchΔ constructs in the presence of sgRNA A or both sgRNA A and B plasmids (Supplementary Fig. 6e, g). The results in the presence of Cas9 showed strong similarity between repair outcomes obtained in the HEK293 and in the HEK293T cells (Figs. 7 and 8).

For the one-DSB experiment, the frequency of constructs with the original sequence (uncut or perfectly repaired by NHEJ) was significantly higher for the BranchΔ than the Sense construct (Fig. 7a). The frequency of NHEJ in/dels was slightly higher in the BranchΔ construct compared to the Sense construct (Fig. 7a, b) especially for the HEK293T cells, as previously observed (Fig. 2a). Markedly, the results of DSB repair by MMEJ for the exon-exon microhomology pairs (those between the two exons: Exon1-Exon2 for cleavage by sgRNA A) were opposite to those of exon-intron microhomology pairs (those between an exon and intron: Exon1-Intron for cleavage by sgRNA A) both for HEK293 and HEK293T constructs. The exon-exon MMEJ had higher frequency of DSB repair for the construct with splicing (Sense) than for the construct without splicing (BranchΔ) (Fig. 7a, c, and results with individual-microhomology pairs in Supplementary Fig. 13), while

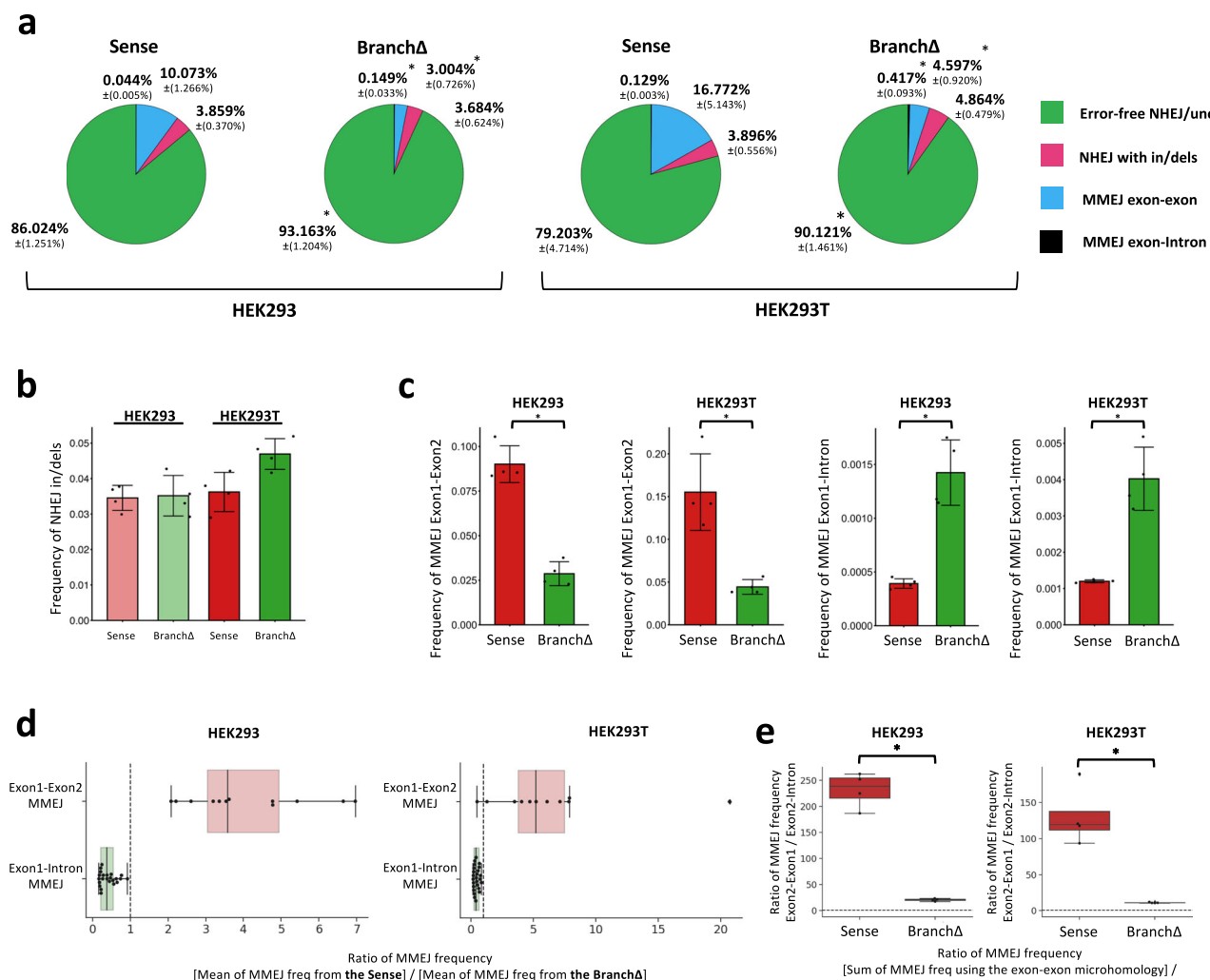

**Fig. 7 | Sequence-dependent RNA-mediated DNA double-strand break repair by end joining is independent of DNA replication. a** Pie charts showing frequencies of sequencing reads in the categories Error-free NHEJ/uncut sequence (green), NHEJ with in/dels (red), MMEJ exon-exon (blue), or MMEJ exon-intron (black) following a DSB by the sgRNA A in the Sense and BranchΔ constructs of HEK293 and HEK293T cells. Percentages represent the average of 4 repeats with standard deviations in parenthesis; $N = 4$. *$p = 0.029$ comparing frequencies of the BranchΔ with those of the Sense construct via the two-tailed Mann–Whitney $U$ test. **b** Frequencies of NHEJ in/dels observed following a DSB by the sgRNA A in the Sense (red) and BranchΔ (green) constructs of HEK293 (transparent, left) and HEK293T (solid, right) cells. Plotted data are the mean ± s.d. of the 4 biological replicates with the individual values shown as dots; $N = 4$. **c** Sum of MMEJ frequencies from all microhomology pairs of exon-exon MMEJ (two left graphs) or exon-intron MMEJ

(two right graphs) following a DSB by the sgRNA A (see Supplementary Figs. 9b and 13) in the Sense (red) and BranchΔ (green) constructs of HEK293 (left) and HEK293T (right) cells. Plotted data are the mean ± s.d. of the 4 biological replicates with the individual values shown as dots; $N = 4$. *$p$ value = 0.029 (two-tailed Mann–Whitney $U$ test). **d** Boxplot showing the ratios of MMEJ frequencies from exon-exon (red box) and exon-intron (green box) microhomologies between the Sense and the BranchΔ constructs of HEK293 and HEK293T cells after DSB by sgRNA A. For details, see legend of Fig. 3c. **e** Ratio of MMEJ frequencies for the exon-exon and exon-intron following a DSB by the sgRNA A in the Sense (red) and BranchΔ (green) constructs of HEK293 (left) and HEK293T (right) cells. Each ratio was calculated by using an average of 4 repeats of MMEJ frequencies from each construct, $N = 4$. Other details are as in the legend of Fig. 3c. *$p = 0.029$ (two-tailed Mann–Whitney $U$ test). Source data are provided as a Source Data file.

the exon-intron MMEJ had higher frequency of DSB repair in the construct without splicing than in that with splicing (Fig. 7a, c, and results with individual-microhomology pairs in Supplementary Fig. 13). For each microhomology pair, we then calculated the ratio of the mean frequency in the Sense construct to the mean frequency in the BranchΔ construct. Similarly to results for repair of a DSB by sgRNA A or B shown in Fig. 3c, the Sense/BranchΔ ratios for the exon-exon microhomologies were mostly higher than 1, while the Sense/BranchΔ ratios for the exon-intron microhomology pairs were mostly lower than 1 (Fig. 7d). Also, the ratio of the total exon-exon frequencies to the total exon-intron frequencies for MMEJ within the Sense and pCMVΔ libraries was significantly higher than that obtained for the BranchΔ libraries thus suggesting more efficient MMEJ between exon sequences for the constructs with splicing compared to the construct without

splicing (Fig. 7e). These results support the role of spliced RNA in promoting MMEJ between exon-exon microhomologies, and the role of the non-spliced RNA in promoting MMEJ between exon-intron microhomologies which are independent from DNA synthesis.

In the double-strand gap experiment, we compared the NHEJ and MMEJ frequencies of intron pop-out with those obtained for the signatures of intron retention in the Sense or BranchΔ constructs extracted from the HEK293 cells. We found that the construct with splicing (Sense) had a higher frequency of intron pop-out than the BranchΔ construct (percentages in bold in Fig. 8a). Both the frequencies of intron deletion by NHEJ and those by MMEJ for the Sense construct were higher than those obtained for the BranchΔ construct (Fig. 8a–c and Supplementary Fig. 13). In stark contrast, and in line with results obtained for the BranchΔ and the Sense constructs in the

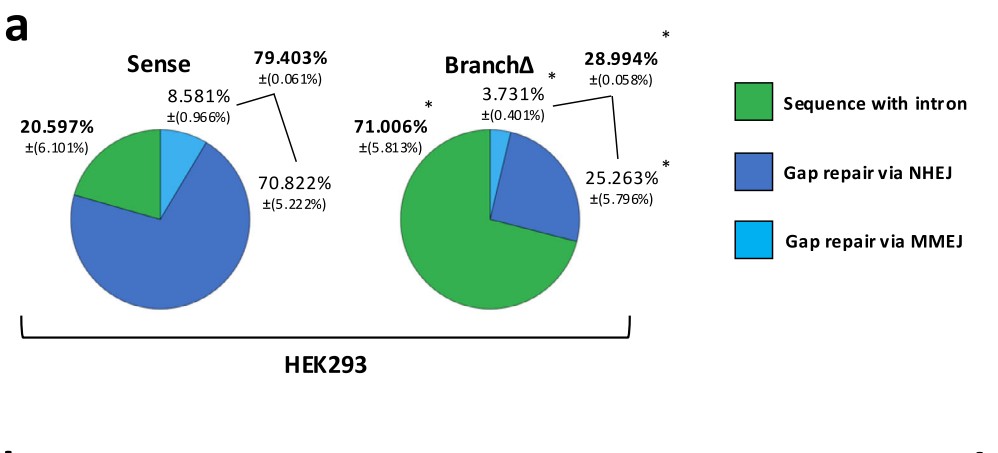

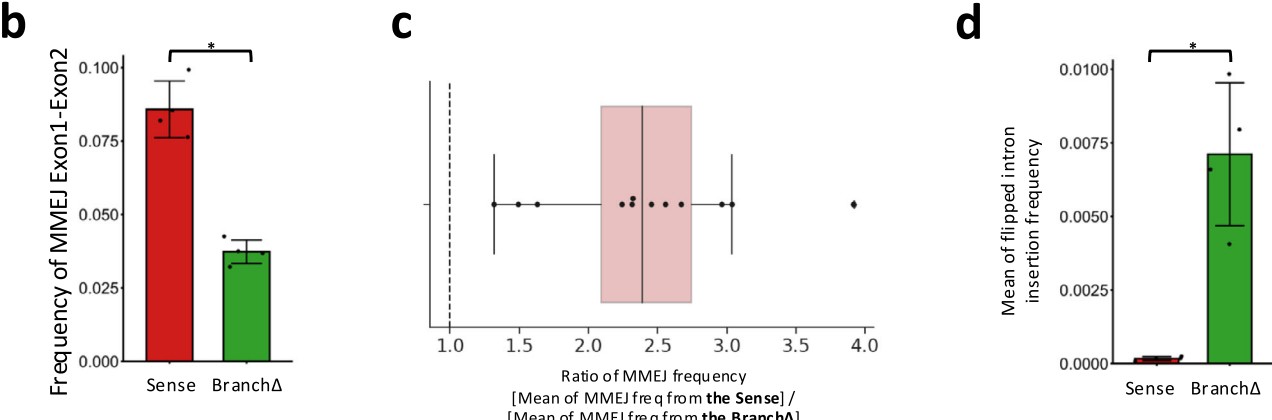

**Fig. 8 | Sequence-dependent RNA-mediated DNA double-strand gap repair by end joining is independent of DNA replication. a** Pie charts showing frequencies of sequencing reads displaying intron retention or pop-out following a double-strand gap by the sgRNAs A and B in the Sense and BranchΔ of HEK293 cells. Percentages represent an average of 4 repeats with standard deviation in parenthesis; $N = 4$. The percentages of sequences with and without intron are bolded. *$p = 0.029$ comparing frequencies of the BranchΔ with those of the Sense construct via the two-tailed Mann−Whitney $U$ test. **b** Sum of MMEJ frequencies from all microhomology pairs of exon-exon MMEJ (see Supplementary Figs. 11b and 13) following a double-strand gap by the sgRNAs A and B in the Sense (red) and BranchΔ (green) constructs of HEK293 cells. Plotted data are the mean ± s.d. of the 4 biological replicates with the individual values shown as dots; $N = 4$. *$p = 0.029$

(two-tailed Mann−Whitney $U$ test). **c** Boxplot showing the ratios of MMEJ frequencies following a double-strand gap by the sgRNAs A and B between the Sense and the BranchΔ constructs of HEK293 cells. The numerator and denominator of each ratio was calculated by using an average of 4 repeats of MMEJ frequencies from each construct. Twelve Exon1-Exon2 microhomologies (black dots) are shown for MMEJ following a double-strand gap by the sgRNAs A and B. Details are as in the legend of Fig. 3c. **d** Frequency of intron flipping caused by re-capture of the intron via NHEJ following a double-strand gap by the sgRNAs A and B in the Sense (red) and BranchΔ (green) constructs of HEK293 cells. Plotted data are the mean ± s.d. of the 4 biological replicates with the individual values shown as dots; $N = 4$. *$p = 0.029$ (two-tailed Mann−Whitney $U$ test). Source data are provided as a Source Data file.

HEK293T cells, we found that the flipped-intron insertion was more frequent in the BranchΔ than in the Sense construct also in the HEK293 cells (Fig. 8d). In sum, the results from the HEK293 cell constructs indicate that RNA transcripts can enhance the repair of a double-strand gap via NHEJ or MMEJ pathways. These repair mechanisms are sequence-dependent and occur at the DNA gene site corresponding to the RNA. Importantly, they function independently of the DNA synthesis status of the gene region.

**An antisense transcript RNA guides double-strand gap repair in its corresponding DNA by Ku70-dependent NHEJ in yeast cells**
To determine whether RNA could directly modulate DSB repair via end joining also in the yeast DNA, we engineered the genome of a set of *Saccharomyces cerevisiae* strains with two constructs (Antisense and BranchΔ) like those used in human cells. A strain containing an integrated *his3* marker gene interrupted by an artificial intron, which can be spliced from an antisense RNA (CM859/860, Supplementary Data 5), was used to integrate the Cas9 nuclease expressed under the galactose-inducible promoter, and two constitutively expressed guide RNAs (sgRNA C and sgRNA D) designed to generate a Cas9-DSB on each side of the intron in chromosomal DNA (Fig. 9a, b and

Supplementary Fig. 14, and see "Methods"). The Antisense construct produces a *his3* antisense RNA from the constitutive pTEF promoter and can splice the intron, while the BranchΔ has a 33-bp deletion of the branch site that prevents splicing (strains CM1033/1035, Supplementary Data 5, Fig. 9a and Supplementary Figs. 14 and 15a, b). RNA-seq data obtained from yeast cells prior to the addition of galactose reveals a significant presence of spliced transcripts (Supplementary Fig. 15c), indicating that the RNA transcripts are already present before the induction of Cas9 transcription. The sgRNAs C and D cut the Antisense and BranchΔ constructs similarly in vitro using Cas9 (Supplementary Fig. 16). To investigate the capacity of the yeast *his3*-antisense RNA to mediate the repair of a double-strand gap in yeast chromosomal DNA, we generated a DSB on each side of the intron in the two constructs by adding galactose to the yeast cell cultures to activate the Cas9 nuclease (Fig. 9b). After two days, the genomic DNAs were extracted from the yeast strains and prepared for NGS to study the sequence of the *his3* gene around the DNA double-strand gap. The NGS data from the Antisense and the BranchΔ constructs were analyzed to determine the frequency of signatures of NHEJ with intron deletion, and MMEJ with intron deletion by recombination between exon-exon microhomologies (Fig. 9b and Supplementary Fig. 17a, see "Methods"). The

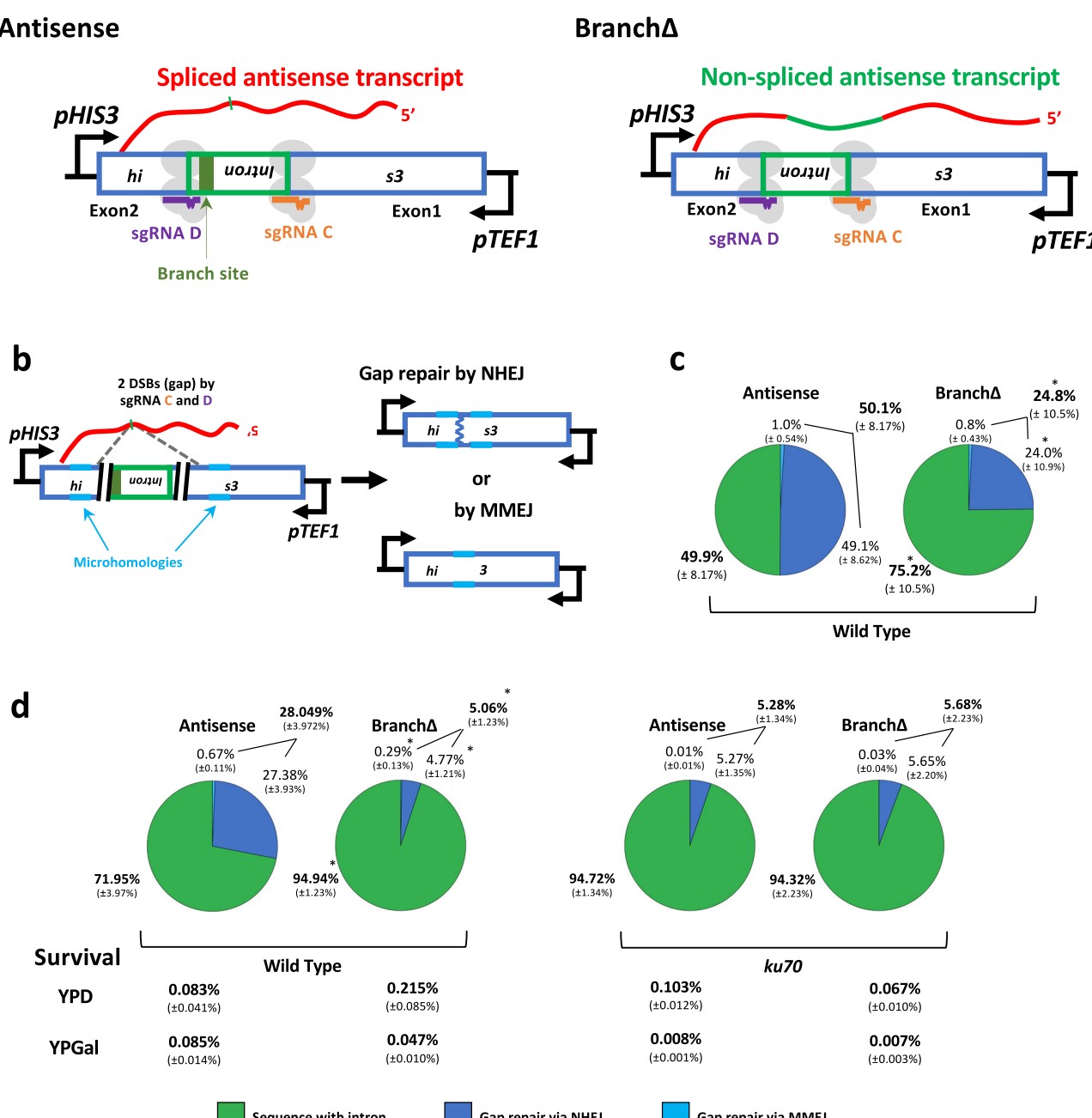

**Fig. 9 | Transcript RNA promotes double-strand gap repair in a sequence-dependent manner in yeast cells. a** Schemes of yeast constructs expressing RNA transcripts that differ by sequence. All these constructs contain the *his3* gene (blue-framed box) with an intron (green-framed box) in the antisense orientation. The antisense transcript RNAs (in red) are depicted after intron splicing (thin green mark) or carrying the intron (thick green line) for the BranchΔ construct. The single-guide RNA (sgRNA) C (orange line) and sgRNA D (purple line) with Cas9 endonuclease (light gray ovals) bind to the complementary DNA of the yeast constructs to generate a DSB. Black arrows: *pHIS3*, *his3* promoter, or *pTEF1*, *TEF1* promoter. Dark green box: branch site of the intron. **b** Scheme of DNA products for the yeast constructs obtained following different DSB repair mechanisms. NHEJ, non-homologous end joining; MMEJ, microhomology-mediated end joining; DSB (black parallel lines); an example of a microhomology pair (light blue lines); DNA repaired by NHEJ (blue zigzag line). **c** Pie charts showing frequencies of sequencing reads displaying intron retention or pop-out following a double-strand gap by the sgRNAs C and D in the Antisense and BranchΔ constructs of wild-type cells following 48 h DSB induction. The NHEJ blue sector contains a DNA product having an identical sequence with the spliced RNA which could be a result of RNA-templated DNA repair (R-TDR), cDNA-templated DNA repair (c-TDR), or NHEJ. The frequencies of this DNA product are 0.03% (Antisense, ±0.03%) and 0.03% (BranchΔ, ±0.01%). Percentages represent an average of 4 repeats with standard deviation in parenthesis; *N* = 4. The percentages of sequences with and without intron are bolded. *p = 0.029 comparing frequencies of the BranchΔ with those of the Antisense construct via the two-tailed Mann–Whitney *U* test. **d** Pie charts showing frequencies of sequencing reads displaying intron retention or pop-out following a double-strand gap in the Antisense and BranchΔ constructs of wild-type and *ku70* mutant cells following 18 h DSB induction. Survival of each construct on YPD and YPGal medium is shown under each pie chart. Percentages represent an average of 4 repeats with standard deviation in parenthesis; *N* = 4. Source data are provided as a Source Data file.

in/del signature was much reduced or practically absent in the No-DSB controls, in which galactose was not added to activate Cas9 expression (Supplementary Fig. 17a). We compared the frequencies of intron deletion obtained for the Antisense construct with those obtained for the BranchΔ construct. Like the results in human cells, we found that the construct with splicing, Antisense, had a higher frequency of intron pop-out than the construct without splicing, BranchΔ (percentages in bold in Fig. 9c). The frequencies of MMEJ were also higher for the Antisense construct compared to the BranchΔ construct, although, due to a larger frequency variation for the yeast chromosomal samples compared to the human plasmid samples, the difference was not significant (Fig. 9c). Overall, these results suggest a role of the spliced RNA in promoting intron deletion via NHEJ and possibly MMEJ, and, vice versa, a role of the non-spliced RNA in maintaining the intron in its original locus also in yeast cells and on chromosomal DNA.

Because elimination of both RNase H1 and H2 function by deletion of the *RNH1* gene and the gene expressing the catalytic subunit of RNase H2 (*RNH201*) markedly promotes RNA-templated DSB repair (R-TDR) in yeast cells[8], we hypothesized that in an *rnh1 rnh201*-null background, RNA may also promote NHEJ more efficiently. We therefore deleted *RNH1* and *RNH201* in strains containing either the Antisense or the BranchΔ construct, expressing Cas9 and both sgRNAs C and D. To avoid cDNA interfering with double-strand gap repair via NHEJ in yeast chromosomal DNA of the *rnh1 rnh201*-null cells, we deleted the *SPT3* gene that is required for reverse transcription and formation of cDNA in yeast[8]. After induction of the two DSBs by sgRNAs C and D in the RNase H-wild type (CM876, YL016 for Antisense, and YL027/028 for BranchΔ, Supplementary Data 5) and the *rnh1 rnh201*-null cells (YL037/038 for Antisense, and YL033/034 for BranchΔ, Supplementary Data 5) containing the Antisense or the BranchΔ construct, genomic DNA was extracted from the cells and prepared for NGS to study the sequence of the *his3* gene around the DNA double-strand gap (Supplementary Fig. 17b, c, see "Methods"). The in/del signature was much reduced or practically absent in the No-DSB controls, in which galactose was not added to activate Cas9 expression (Supplementary Fig. 17b, c). As expected, and opposite to the results of RNase H1 overexpression in the human cells, in the yeast *rnh1 rnh201*-null cells, the frequency of intron deletion was further enhanced in the Antisense construct with splicing vs. the BranchΔ construct without splicing (Supplementary Fig. 18a). The ratio of frequencies of intron retention for the Antisense vs. the BranchΔ construct decreased, while that of intron loss increased in the *rnh1 rnh201*-null cells compared to the RNase H-wild-type cells (Supplementary Fig. 18b). These results further corroborate a direct role of transcript RNA in modulating DSB-repair via end joining mechanisms.

Finally, to examine whether the RNA-driven double-strand gap repair via NHEJ occurred through the Ku70/Ku80-NHEJ factor, we deleted the *KU70* gene in the yeast strains containing the *his3*-Antisense or the BranchΔ constructs generating strains YL085,6,7,8 which also express the sgRNAs C and D and Cas9 under the galactose promoter (Supplementary Data 5). Wild-type and *ku70*-null strains were incubated in the presence of galactose to activate the Cas9 nuclease and generate a double-strand gap. The galactose-incubation time was reduced from 48 h to 18 h to minimize variability across the different strains (see Methods). The genomic DNAs were then extracted from the wild-type and *ku70*-null strains and prepared for NGS to determine the frequency of the different repair outcomes following generation of the double-strand gap in the *his3* gene of the Sense and BranchΔ constructs. The results of the data analysis are shown in Fig. 9d and No-DSB controls, in which galactose was not added preventing Cas9 expression, are shown in Supplementary Fig. 18c. We compared the frequencies of intron deletion obtained for the Antisense construct with those obtained for the BranchΔ construct. Like the results with 48 h incubation in galactose, we found that the construct with splicing, Antisense, had a higher frequency of intron pop-out than the construct

without splicing, BranchΔ (over a factor of 5) also when the cells were incubated in galactose for 18 h (percentages in bold in Fig. 9d). Not only the NHEJ frequency but also the frequency of MMEJ was significantly higher for the Antisense construct compared to the BranchΔ construct, likely due to the short (18 h) incubation in galactose that reduces sample variability (Fig. 9d). Notably, deletion of *KU70* eliminated the frequency difference for intron-pop out between the splicing and the non-splicing constructs (Fig. 9d). These results suggest that the role of the spliced RNA in promoting intron deletion occurs mainly through the Ku70-NHEJ pathway. Interestingly, while yeast cell survival was profoundly reduced in the *ku70*-null cells, both those with the Antisense and those with the BranchΔ construct, as expected when cells were kept on galactose, survival calculated immediately after the 18h-galactose incubation (see Methods) was higher and showed higher percentage for the BranchΔ construct than the Antisense construct possibly indicating a greater capacity of the RNA of the BranchΔ construct to aid in gap repair. In the *ku70*-null mutants, the survival was unchanged for the Antisense construct, while for the BranchΔ construct, the survival was reduced by a factor of 3 (Fig. 9d). The results may suggest that RNA may play a role in maintaining genome integrity, although further experiments are needed to fully characterize this role.

## Discussion

This study provides evidence that a transcript RNA mediates DNA DSB repair via end joining in a sequence-specific manner in human and yeast cells expressing wild-type RNase H genes. We developed genetic constructs expressing RNA transcripts that differ by the presence or absence of an intron, or the level of transcription due to the presence or absence of the CMV promoter. When the transcript RNA is complementary to the DNA sequences at the DSB ends on both sides of the DSB (non-splicing constructs: BranchΔ and 5′-SplicingΔ if the break is near an exon-intron junction, or both splicing and non-splicing constructs if the DSB is within an exon), it more often supports the conservation of the original uncut sequence via NHEJ, compared to the RNA that does not have complementarity to both the DSB ends (splicing constructs: Sense, pCMVΔ, and Antisense if the break is near an exon-intron junction). On the contrary, the transcript RNA that has complementarity distant from the DSB ends more often promotes deletion of the DNA region near the DSB, which it does not share complementarity with, both via NHEJ and MMEJ in double-strand gap repair. We cannot say whether the spliced transcript RNA facilitates intron pop-out more than the non-spliced transcript RNA facilitates intron retention. This may be affected by the level of transcription, by the extension of the RNA-DNA interaction, and/or other factors, and would thus require further studies employing new construct designs and/or targeting different genomic sites. Interestingly, similar results were obtained also when the plasmid constructs could not go through DNA replication inside the HEK293 cells. These findings suggest that transcript RNA could significantly impact DSB repair outcomes not only in dividing cells but also in non-dividing cells, highlighting a potential broader role for RNA in DSB repair mechanisms across different cellular states.

The plasmid system for human cells allows easy engineering of splicing and non-splicing constructs, providing an exportable system for experiments in various cell lines like HEK293T wild-type, RNaseH2A KO cells, and HEK293 cells. Because constructs with the *DsRed* gene transcript, Cas9, and sgRNA are co-transfected in human cells, we assume that by the time Cas9 and sgRNA(s) reach the nucleus to cut the target site(s), *DsRed* gene transcripts are likely already present. This assumption is strongly supported by our yeast cell results, where splicing and non-splicing constructs are integrated into the genome and expressed from the constitutive pTEF promoter. The Cas9 transcript, expressed from a galactose-inducible promoter, is activated by adding galactose to the medium. Indeed, the findings from the plasmid

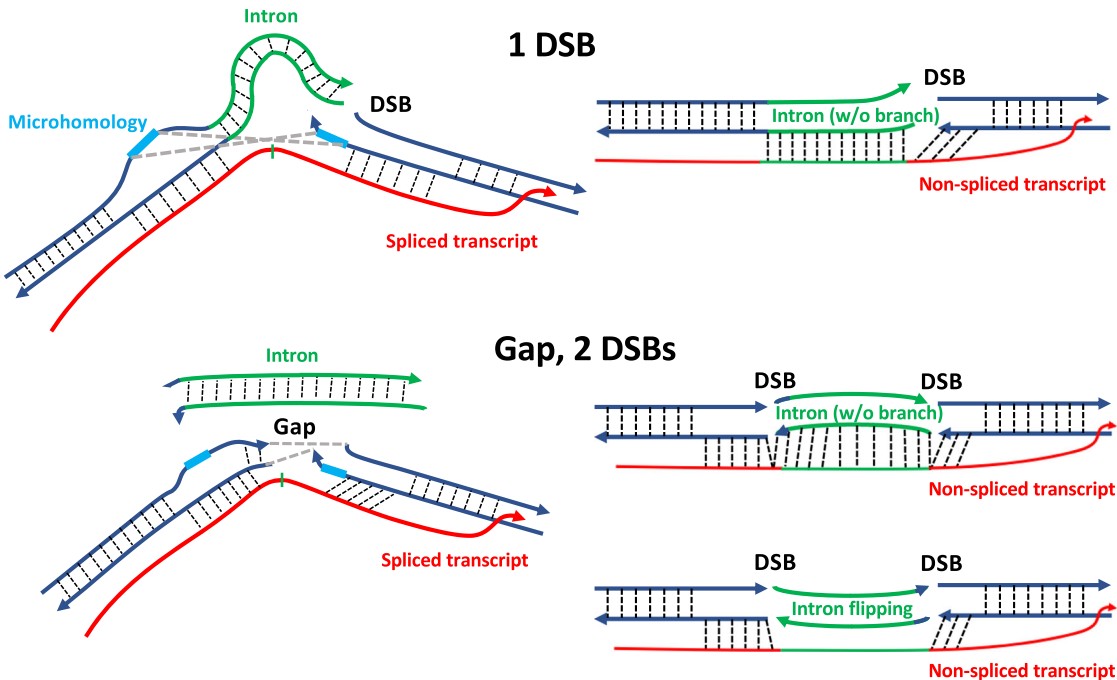

**Fig. 10 | Models of transcript RNA-mediated DSB repair.** When a DSB is generated near one or both exon-intron junctions of a gene, the transcript RNA of that gene supports DSB/double-strand gap repair with intron pop-out if spliced (schemes on the left), or intron retention if non-spliced (schemes on the right). On the other hand, the non-spliced transcript RNA by interacting with the DSB ends of a double-strand gap on both sides of the intron, maintains the double-strand gap ends distant from each other allowing intron flipping (scheme on the bottom right). Dotted lines, hydrogen bonds; blue lines, exon sequences; green lines, intron sequences in DNA or RNA; red line, the gene RNA transcript; thin green mark, splice site of the intron in RNA; light blue lines, microhomology; gray dashed lines, interaction between the broken DNA ends facilitated by the transcript RNA.

constructs in human cells are consistent with the results from yeast chromosomal constructs, suggesting a conserved role for transcript RNA in DSB repair mechanisms across eukaryotic cells. Future directions include integrating the splicing and non-splicing constructs into the human genome.

Additionally, data from double-strand gap-repair experiments, involving RNase H1 overexpression in human cells and RNase H deficiency in yeast, support a direct interaction of transcript RNA with DSB ends and a role for RNA in promoting intron deletion or retention. We propose two mechanisms of RNA-mediated DSB repair: RNA-mediated NHEJ (R-NHEJ) and RNA-mediated MMEJ (R-MMEJ). In these mechanisms, RNA, due to its complementarity to the DNA sequence from which it is transcribed, bridges the DSB ends to facilitate NHEJ or MMEJ in cells expressing wild-type RNase H genes (Fig. 10).

Co-transcriptional RNA-DNA interactions are not rare but rather prevalent and form dynamic structures occupying up to 5% of mammalian genomes under physiological conditions, with RNA-DNA hybrids extending over 50%–100% of gene body for a large fraction of genes[20]. In support of a model in which the transcript RNA interacts with the broken ends of its DNA to modulate DSB repair, the results shown in Figs. 3 and 6 reveal that spliced RNA promotes exon-exon MMEJ, and non-spliced RNA promotes exon-intron MMEJ. The non-spliced transcript from the BranchΔ construct retains the intron in the RNA while the transcripts from the Sense and the pCMVΔ splicing constructs do not. The transcript RNA of the BranchΔ, if it interacts with DNA, can help keep the intron microhomologies near the corresponding exon microhomologies. For the DSB by sgRNA B, the advantage of the non-spliced over the spliced constructs for MMEJ between exon-intron microhomologies is reduced in Exon2-Intron MMEJ when the microhomology in the intron is close to the beginning of Exon1 (compare MMEJ bar graphs for microhomologies EI27R_GTC or EI29R_CCTT with those for microhomologies EI37R_GTC or EI38R_TAGA in Supplementary Fig. 9c, e). When a DSB or a double-

strand gap are generated within an exon sequence, then there is no difference or much less difference in DSB-repair frequency by end joining between the splicing and the non-splicing constructs. In fact, the transcript RNAs generated from the splicing and the non-splicing constructs all retain the exon sequence around the DSB sites. Thus, the transcript RNAs affects similarly the end-joining repair in these constructs allowing the RNA to preferentially maintain the original DNA sequence before breakage. The slightly higher frequency of Exon1 segment pop-out observed in the splicing constructs compared to the non-splicing construct can be explained by a reduced level of complementarity with the DNA sequence downstream of the DSB by sgRNA J of the spliced RNA compared to the non-spliced RNA. Moreover, it is possible that the spliced RNA via its interaction downstream of the intron may enlarge the break at the DSB by sgRNA J facilitating segment loss.

The variation-distance graphs used to analyze the sequencing reads derived from R-NHEJ gave a detailed snapshot of the DSB-repair sequence variation types (insertions and deletions), nucleotide compositions, and their frequencies, displaying the complete results of millions of DSB repair events in a single image. In the 1-DSB system, we found that graphs arising from different guide RNAs had marked differences, while those arising from the same guide RNA were similar to each other. In graphs corresponding to sgRNA A, inserted nucleotides had a preference to start with T, while in those of sgRNA B, inserted nucleotides had no preference among the four bases (Fig. 2b and Supplementary Fig. 8a, c). Such differences in the in/del type of NHEJ repair at these two DSB sites supports marked dependence of Cas9 cleavage on the sequence context, as previously reported[21]. Among the insertions starting with nucleotide T, the mononucleotide T and the dinucleotides TA, TG, and TT had high frequency for NHEJ repair after DSB by sgRNA A, in addition to being more abundant for the BranchΔ construct (Fig. 2b, c and Supplementary Figs. 7 and 8). These T-initiated insertions are most likely due to 5′-sticky end cleavage of

Cas9 on the PAM strand followed by templated[21] (T and TG) or partially templated (TA and TT) insertions. Therefore, it is possible that the biased T-initiated insertions for repair of the DSB caused by sgRNA A reflect a frequent 5′-sticky end cleavage by Cas9, and the non-biased insertions following DSB by sgRNA B reflect a most prevalent blunt-end cleavage by Cas9. Future experiments could be conducted to understand whether transcript RNA can better support repair of blunt vs. sticky-end DSBs.

Interestingly, in all constructs, the generation of a double-strand gap in the 2-DSB system produced greater variety of in/dels (i.e., different vertices in the graph) after repair by NHEJ than in the 1-DSB systems, particularly for deletions (compare graphs in Fig. 4c and Supplementary Fig. 10a, c with those in Fig. 2b and Supplementary Fig. 8a, c). Moreover, although the variation of the in/dels among the different constructs is similar within the 1-DSB and the 2-DSB systems, the relative frequencies of the corresponding in/dels were consistently higher in one construct than in the other as the comparison graphs indicate. The variation-distance graphs also reveal other details that cannot be seen with the bar graphs. For example, when the DSB is caused sgRNA B, there are more chances that the transcript RNAs of the splicing constructs, especially for the pCMVΔ construct (because there are less transcripts), have already lost the intron. The variation-distance graph comparing the NHEJ repair after DSB by sgRNA B for the Sense vs. the pCMVΔ construct shown in Supplementary Fig. 8b highlights a more efficient NHEJ repair for all individual in/dels of the Sense construct. The difference between the NHEJ repair of the Sense vs. the pCMVΔ construct is enhanced in the RNase H2A KO cells (Supplementary Fig. 8d). It is likely that the transcript RNAs of the pCMVΔ compared to those of the Sense construct have already spliced out the intron when the DSB occurs at the Intron-Exon2 junction (DSB by sgRNA B); thus limiting the capacity of the transcript RNA of the pCMVΔ construct to repair the DSB by NHEJ. Instead, for a DSB by sgRNA A the transcripts from the splicing constructs may still carry the intron, and a low level of transcription from the pCMVΔ construct may facilitate the interaction between the RNA that still carries the intron and the DSB ends resulting in higher frequency of NHEJ repair for most of the in/dels of the pCMVΔ construct (Supplementary Fig. 8b). In this case, the difference between the NHEJ repair of the Sense vs. the pCMVΔ construct is reduced in the RNase H2A KO cells (Supplementary Fig. 8d). Most of the variation-distance graphs show that when comparing the NHEJ frequencies between Sense and BranchΔ, Sense and pCMVΔ, or Antisense and 5′-SplicingΔ, the relationship between the overall NHEJ frequencies of the two constructs is reflected in most of the individual sequence frequencies as well. Some graphs show sequences that are exceptions to this rule, such as in Supplementary Fig. 8b, d (sgRNA B, Sense vs. BranchΔ), and 10f (Antisense vs. 5′-SplicingΔ) where we see some sequences that have higher frequency in the construct that has overall lower frequency. Further study will be required to determine why such sequences are enriched differently.

A limitation of the 1-DSB system is that we cannot distinguish error-free NHEJ repaired samples from the uncut samples, and from the samples having error free repair due to recombination between a cut and an uncut plasmid. The error free repaired plasmids can be cut again providing more opportunity for NHEJ as well as in/dels. In fact, we may be underestimating the capacity of RNA to influence NHEJ in the 1-DSB system. The issue was circumvented by generating 2 DSBs to form a double-strand gap between the two exon-intron junctions. In the case of the 2-DSB system for double-strand gap repair, we can analyze all the products of repair by NHEJ, and we do see a major impact of RNA on NHEJ. The 2-DSB system gave consistent results in three different series of constructs: the sense and the antisense constructs for experiments in human cells, and the antisense constructs for the experiments in yeast cells.

We also showed that, contrary to R-TDR[8,9], a low level of transcription of the RNA still has a major impact on R-NHEJ and/or R-MMEJ in the assays we used. Nonetheless, the transcription level does impact R-NHEJ and R-MMEJ. The frequency of NHEJ repair was similar in both the Sense and pCMVΔ constructs for a DSB by sgRNA A, but significantly lower in the pCMVΔ construct for a DSB by sgRNA B (Fig. 2a and bottom right panels in Supplementary Fig. 8b, d), suggesting that the amount of RNA from the pCMVΔ construct still carrying the intron over the DSB site of sgRNA B is likely less than that from the Sense construct. Thus, in this case, the low level of transcription of the spliced RNA in the pCMVΔ construct decreases the efficiency of NHEJ repair of a DSB at the 3′-intron-exon junction compared to not only the non-spliced RNA of the BranchΔ construct, but also the highly-expressed spliced RNA of the Sense construct. Moreover, we established that sense (coding) and antisense (non-coding) RNAs guide DSB repair via NHEJ and MMEJ. Because cellular non-coding RNAs are usually expressed at a lower level compared to mRNAs of coding genes[22], the results suggest that transcript RNA, regardless of its type and level of transcription, may markedly influence the efficiency of DSB-repair mechanisms by end joining. It is interesting to note that while introns in mRNAs tend to be removed co-transcriptionally, introns in long non-coding RNAs (lncRNAs) tend to be spliced post-transcriptionally[23–25]. This suggests that non-coding RNA, like lncRNAs, even when transcribed at low levels, could serve a more general function than previously anticipated in maintaining genomic integrity in both replicating and non-replicating DNA.

While we cannot conclusively prove that the splicing process and/or the presence of the transcript directly influence Cas9 activity, the evidence suggests that RNA plays a significant role in mediating repair outcomes post-cleavage, rather than primarily affecting the cleavage efficiency itself. This observation aligns with the similar repair profiles seen in constructs with high and low transcription level, and with the reproducibility of the results across different cell lines, genotypes, and species. The fact that a DSB or a double-strand gap within Exon1 minimized the effect of splicing in the RNA-mediated end-joining repair analyzed in this study underscores that a transcript RNA can impact DSB repair via end joining differently depending on the location of the breaks in DNA. Interestingly, the predominant difference in double-strand gap repair outcomes between splicing and non-splicing constructs in yeast depended on Ku70. These results suggest that RNA-mediated end joining proceeds via a Ku70-dependent mechanism. MMEJ outcomes were generally less prominent in yeast compared to human cells. Unexpectedly, in the *ku70* mutants the MMEJ frequency was even lower than in wild-type cell. Future work will not only focus on investigating genetic factors, such as the effects of different NHEJ mutants in yeast and human cells and the role of the reverse transcriptase activity of Pol ϑ[26] in R-MMEJ in human cells, but will also explore how the position of DSBs relative to exon and intron sequences affects RNA-mediated DSB repair in cells. For example, the choice of using spliced RNA vs. non-spliced RNA (i.e., pre-mRNA) in R-NHEJ and R-MMEJ for the splicing constructs may mainly depend on the position of the DSB relative to the exon and intron sequences, and in part also on the distance of the microhomologies from the 3′-splice site for R-MMEJ. The most efficient mechanism of DSB repair in human cells is NHEJ[1,2]. This study shows that the transcript RNA of a broken gene helps NHEJ when the RNA retains complementarity with the DSB ends. Therefore, the results suggest that DSB repair mediated by a spliced mRNA may more efficiently stabilize gene regions that experience a DSB in exonic sequences rather than near exon-intron junctions or in intronic sequences, for which the spliced mRNA does not have complementarity to one or both broken DNA ends of a DSB, respectively.

RNA could also play a role in destabilizing the genome. The experiments, which involved cuts on both sides of the intron in the

*DsRed* gene of the sense and antisense constructs, revealed that the non-spliced constructs have a higher frequency of intron flipping than the spliced constructs. These results suggest that RNA may hold DNA ends not just close to each other, but also distant from each other (see bottom right scheme in Fig. 10), depending on the sequence complementarity between the RNA and the DNA. Thus, RNA can guide genome alterations. In summary, we identified a previously unknown function of RNA in modulating the repair of DSBs and double-strand DNA gaps. The findings provide new avenues for understanding mechanisms of genome integrity, genome modification, and evolution.

## Methods

### Constructs to study DSB repair mediated by RNA in human cells

To study DSB repair mediated by transcript RNA in human cells, we built constructs expressing RNA transcripts varying in sequence and transcription level, three 'sense' constructs: Sense, BranchΔ, and pCMVΔ, and two 'antisense' constructs: Antisense and 5′-SplicingΔ, in which the transcribed RNA is the sense or the antisense RNA of the *DsRed* gene, respectively. An artificial intron was PCR amplified from the plasmid DNA of pSM50 (provided by Dr. Garfinkel's lab at University of Georgia) attaching the PstI restriction enzyme site at its ends, purified by spin column (QIAquick PCR Purification Kit, Qiagen), and was introduced into the *DsRed* gene of the pDsRed plasmid (Addgene, #54493) in its SbfI site, which is compatible to the PstI site, in the sense orientation to make the Sense construct. The ligated plasmid was transformed into *Escherichia coli* cells (Agilent, XL1-Blue Competent Cells). Plasmids were isolated from the *E. coli* colonies grown on kanamycin (40 μg/ml) contained LB plate by Miniprep (GeneJET Plasmid Miniprep Kit, Thermo Scientific). Each of the plasmids containing the sense or antisense orientation of the artificial intron was confirmed by Sanger sequence analysis (Eurofins Genomics). There was a residual 6 bp of repeated sequence at the junction of the exon-intron inserted during the restriction enzyme cloning process. The 6 bp located next to the 3′-splicing site of the intron was deleted from both the sense and antisense plasmids by the in-vitro mutagenesis (QuikChange II Site-Directed Mutagenesis kit, Agilent) to reconstitute the original sequence of the *DsRed* gene.

Construction of the antisense constructs. The plasmid having the antisense orientation of the artificial intron in the *DsRed* gene was digested by PciI and SspI restriction enzymes (New England BioLabs). The broken ends were filled in by the Klenow fragment (New England BioLabs) to generate blunt ends. Then, the construct was purified by gel extraction (GeneJET Gel Extraction Kit, Thermo Scientific) to isolate the fragment having the *pCMV* and the *DsRed* gene with the antisense intron, and inserted in a vector, the pEGFP-17 (Addgene, #62043) to construct the pSRAIDC. To construct the pSRAIDC, the pEGFP-17 vector was digested by HindIII and BglII restriction enzymes (New England BioLabs), treated by the Klenow fragment, and dephosphorylated with calf intestinal alkaline phosphatase (CIP, New England BioLabs) to prevent self-ligation of the vector. This dephosphorylated vector was purified by spin column and ligated with the insert containing the *pCMV* and the *DsRed* gene with the antisense intron by T4 DNA ligase. The cutting site for the I-SceI-homing endonuclease, TAGGGATAACAGGGTAAT, was inserted in the middle of the intron after a digestion by MluI restriction enzyme (New England BioLabs) and ligation with a short dsDNA oligonucleotide containing the I-SceI restriction enzyme site flanked with MluI restriction enzyme sites by the T4 DNA ligase. The *EF1α* promoter (*EF1α−2* promoter used in Zheng et al.[27]) was amplified by PCR from a plasmid (customized order#18ACPG6P_2379654_EF1a2, Thermo Fisher Scientific) flanked with the AvrII restriction enzyme site to be inserted into the antisense plasmid to drive transcription of the *DsRed* antisense transcript. Both the PCR fragment containing the *EF1α* promoter and the antisense plasmid were digested by AvrII restriction enzyme (New England BioLabs) and ligated by the T4 DNA ligase to merge them.

The sense and antisense constructs were then engineered to contain a protospacer adjacent motif (PAM) sequence in the intron to induce a DSB by sgRNA B for the sense and sgRNA D for the antisense constructs (Supplementary Fig. 1) at the junction between the exon and the intron. The 2 bp in the intron (TA in the 1126-1127 position of the sense plasmid, and in the 1493-1494 position of the antisense plasmid) were substituted to GG by in vitro mutagenesis (QuikChange II Site-Directed Mutagenesis Kit, Agilent). We used an existing XGG sequence for the PAM site for the DSB by sgRNA A for the sense and sgRNA C/C′ for the antisense (Supplementary Fig. 1).

The 55-bp deletion containing the branch site for the splicing mutant of the sense system (BranchΔ) was done by in-fusion reaction (In-Fusion HD Cloning Kit, Clontech). Primers 55bpDel.AI.F and 55bpDel.AI.R (Supplementary Data 1) were used to amplify the plasmid to delete the branch-site containing sequence. A PCR product was purified by gel extraction (GeneJET Gel Extraction Kit, Thermo Scientific). Deleting the first 6 bp of the 5′-splicing site to make the 5′-SplicingΔ construct, was done by in vitro mutagenesis. All the plasmid constructs were confirmed by Sanger sequence analysis (Eurofins Genomics). We used sgRNA C′ to induce a DSB on the 5′-SplicingΔ construct at the 5 bp downstream of the intron-exon junction corresponding to where the sgRNA C induces the DSB on the Antisense construct (Supplementary Fig. 1).

The pCMVΔ plasmid was made by deletion of the CMV promoter from the Sense construct digested by AseI and XhoI restriction enzymes (New England BioLabs). The single-stranded overhang from the digestion of the restriction enzymes was removed by Mung Bean Nuclease (New England BioLabs), then the construct was ligated by the T4 DNA ligase.

The plasmids transcribing single guide RNAs (sgRNAs) were constructed following the method presented in Zhang et al.[28]. The backbone plasmid was provided by Dr. M. Jasin's lab (Developmental Biology Program, Memorial Sloan-Kettering Cancer Center). The in-fusion reaction was used to replace the target sequence. The plasmid expressing Cas9 nuclease was purchased from Addgene (Plasmid #41815).

### Human RNase H1 constructs and overexpression in HEK-293T cells

A plasmid expressing RNASEH1 (Addgene# 111906) was co-transfected into HEK-293T cells with the other plasmids of the assay following the transfection protocol using PEI (see below). A control plasmid expressing a non-functional RNASEH1 gene was constructed by deleting CCGG, 72-75th nucleotides of RNase H1 gene, by the in-vitro mutagenesis (QuikChange II Site-Directed Mutagenesis kit, Agilent). The plasmid construction was confirmed by Sanger sequence analysis (Eurofins Genomics). Overexpression of RNase H1 gene in HEK-293T cells was verified via western blot (RNASEH1 Ab Proteintech, Cat.# 15606-1-AP) (Fig. 6a).

### Yeast strain construction

The yeast strains used in this work were listed in Supplementary Data 4 and derived from CM-510 (Meers et al.[29]). The insertion of Cas9 sequence was done by *delitto perfetto* method[30] by insertion of a CORE cassette into pGal1/10-HO locus to replace HO endonuclease sequence and followed by replacement of CORE using Cas9 coding sequence. A CORE cassette was inserted at the 5′-intron/exon junction and replaced with DNA oligonucleotide sequence generating an SpCas9-compatible PAM (TGG) sequence designed to cleave 1 bp from the intron/exon junction. For a DSB at the 3′-intron/exon junction an existing PAM sequence was used (Supplementary Fig. 13). A CORE cassette was then inserted upstream of the pGal-Cas9 locus and replaced with an empty sgRNA under a constitutive SNR52 promoter

derived from pML104 (Addgene# 67638). The empty sgRNA was replaced by sgRNA C through insertion and replacement of a CORE cassette at the empty sgRNA site. The sgRNA D was inserted into the *TRP1* locus using the same *delitto perfetto* method. For the BranchΔ construct, the branch site (33 bp) was deleted within the artificial intron by insertion of a CORE cassette and the replacement of the CORE cassette by an oligonucleotide sequence lacking the branch site (Supplementary Fig. 13). The RNase H1 and H2-double knockout (*rnh1 rnh201*) strains were constructed using either the G418 or Hygromycin-resistance marker gene to replace the coding sequence of the *RNH1* and *RNH201* genes. The *spt3* mutant strains were constructed by replacement of *SPT3* coding sequence with the *KlURA3* marker gene. The *ku70* mutants were created by replacing the *KU70* gene with *kanMX4*-resistance marker gene.

### Yeast sample preparation and DSB induction

Single-colony isolates from the yeast strains (CM859, CM860, CM1033, CM1035, CM876, YL016, YL027, YL028, YL037, YL038, YL033, and YL034) were incubated in 25 ml YPLac liquid medium for 48 h at 30 °C with shaking. Control samples were collected for both DNA and RNA extraction using 2 ml cell culture. Then, 2.5 ml of 20% galactose was added in the remaining medium to activate Cas9 expression from the pGal promoter, and cells were incubated for 48 h at 30 °C in the shaker, followed by genomic extraction using 1.5 ml cell culture.

For *ku70* mutants and wild-type controls, single colony isolated from the yeast strains (YL085, YL086, YL087, YL088, CM859, CM860, CM1033, and CM1035) were counted and $1.25*10^7$ cells were inoculated in 5 ml YPLac medium and grown for 24 h at 30 °C with shaking. A total of 500 μL of cell culture were used for extracting RNA and DNA as control. Then, 500 μL of 20 % galactose was added in the remaining medium to activate Cas9 expression from the pGal promoter, and cells were incubated for 18 h at 30 °C in the shaker, followed by genomic extraction using 1.5 ml cell culture. $10^5$ cells from the remaining culture were plated on the YPD plates and incubated at 30 °C for 2–3 days to determine survival by counting the individual colonies growing. The survival in YPGal was done by plating $10^5$–$10^6$ cells derived from single colonies on YPD medium to YPGal plates and incubating the cells at 30 °C for 4–5 days. The survival was calculated by dividing the number of colonies grown on the YPGal medium by the number of cells plated on the same medium.

### In-vitro digestion assay with CRISPR-Cas9 system

The sense and antisense plasmids isolated by Midiprep (GeneJET Plasmid Midiprep Kit, Thermo Scientific) were used for the in-vitro digestion assay to test the DSB efficiencies of each sgRNAs in the different constructs. The sense and antisense constructs were PCR-amplified using DsRed.pCis.F8 and DsRed.pCis.R3 primers (Supplementary Data 1) to amplify a region near the DSB sites on the *DsRed* gene (Supplementary Fig. 4). The PCR products were purified by using a 0.8x concentration of magnetic beads (HighPrep PCR Clean-up System, MagBio Genomics) to remove primer dimers. A DSB was induced on the PCR products by using sgRNA A, B or A and B (Supplementary Data 2) synthesized from Integrated DNA Technologies (Alt-R CRISPR-Cas9 sgRNA, IDT) with Cas9 nuclease (*S. pyogenes*, New England BioLabs). The total volume for each assay was 30 μl in a 1.5 ml RNase-free Microfuge tube (Applied Biosystem), which consisted of 30 nM of sgRNA, 30 nM of Cas9 nuclease and 3 nM of plasmid in 1x NEBuffer r3.1 (New England BioLabs). To determine the DSB efficiency by Cas9 with sgRNA A, B, or A and B, 30 nM of each sgRNA, 30 nM of Cas9 nuclease and 3 μl of 10x NEBuffer r3.1 with nuclease-free water for a total volume 29 μl were pre-incubated for 10 minutes at room temperature; then 1 μl of 90 nM PCR construct (from the Sense/pCMVΔ, BranchΔ, Antisense or 5′-SplicingΔ was added into the tube and incubated at 37 °C for 1 h, then 1 μl of Proteinase K (Qiagen) was added into the tube the sample was incubated at 37 °C for 10 min to degrade the

Cas9 nuclease to stop the cleavage. The sizes and molarity of digested fragments were determined by Bioanalyzer High Sensitivity DNA analysis (Agilent).

To prepare the yeast Antisense and BranchΔ constructs by PCR, genomic DNA from yeast strains containing the Antisense and BranchΔ constructs was extracted using MasterPure Yeast DNA Purification Kit with RNaseH treatment. The yeast Antisense and BranchΔ constructs were PCR-amplified using HIS3. 207F and HIS3.206R primers (Supplementary Data 1) from the genomic DNA of the corresponding strains (CM859 and CM1033) and then purified using concentration of magnetic beads (HighPrep PCR Clean-up System, MagBio Genomics) to remove primer dimers. One DSB was induced by either sgRNA C or sgRNA D synthesized from Integrated DNA Technologies (Alt-R CRISPR-Cas9 sgRNA, IDT, Supplementary Data 2) with Cas9 nuclease (S. pyogenes, New England BioLabs). 2 DSBs were induced on the PCR products by using sgRNA C and sgRNA D. The subsequent procedures were the same as previously described.

### Human cell lines and transfection

Human embryonic kidney T (HEK-293T) *RNASEH2A* wild-type and KO cells were provided by Dr. Pursell's lab at Tulane University. The HEK-293 cell line (ATCC Cat.# CRL-1573) was obtained from the American Type Culture Collection. The cells were grown in Dulbecco's modification of Eagle's medium (DMEM) containing 4.5 g/L glucose, L-glutamine and sodium pyruvate (Corning) with 10% fetal bovine serum (Sigma-Aldrich) and 1x penicillin-streptomycin (Gibco). Cells were grown at 37 °C in a 5% $CO_2$ humified incubator. For the construction of the HEK-293T RNase H2A KO clones, *RNASEH2A* (Chr19, exon 2) gRNA (5′-TAACAGATGGCGTAGACCAT-3′) was cloned into GeneArtTM CRISPR Nuclease Vector with OFP reporter (Invitrogen) following manufacturer's protocol. HEK-293T cells were transfected with 6 mL of Lipofectamine 2000 (Invitrogen) and 2.5 ug of gRNA vector DNA when the cells were 60–70% confluent. After 48 h, OFP-positive cells were FACs sorted and serially diluted into 96-well plates at ~1.5 cells per well and incubated for 10-14 days. Single, well-defined clones were expanded to 24-well plates and initially screened with T7 Endonuclease I assay (NEB). KO clones were verified via western blot (α-RNASEH2A Ab Origene Cat.# TA306706) and Sanger sequencing of PCR products (TOPO Cloning Kit Invitrogen). The *RNASEH2A* locus PCR primers used to identify the mutations in the *RNASEH2A* alleles are shown in Supplementary Data 1. We identified a clone, HEK-293T T3-8, having three distinct frameshift mutations consistent with all three alleles modified in hypotriploid 293T cells (Supplementary Fig. 4a). Modification of each allele was determined by TOPO cloning the PCR product (TOPO TA Cloning Kit Invitrogen) into DH5α competent cells (ThermoFisher). Sanger sequencing of the amplified PCR product showed the following observed alterations (all positions indicated with respect to the Cas9 cleavage site on the reference strand): insertion of G at position −1; deletion of five bases at position +2 to +6; complex alteration with deletion of three bases from position −1 to +2, deletion of 2 two bases from position +5 to +6, and CC > TT at position +8 and +9. HEK-293T T3-8 cells were used in this study as RNase H2A KO cells.

For transfection experiments, cells were seeded in 24-well plates at a density of ~50,000 cells per well for transfection and incubated for a day at 37 °C in a 5% $CO_2$ humified incubator. Cells were transfected using polyethylenimine (PEI, Polysciences), 5 μg were used per well for transfection. For the transfection experiments, the sense and antisense plasmids, as well as the Cas9, sgRNA plasmids, and the plasmids used in the experiments for RNASEH1 overexpression were all isolated by Midiprep (GeneJET Plasmid Midiprep Kit, Thermo Scientific) and their concentration was determined using the Nanodrop instrument. In all transfections, plasmids were used in the amount of 0.4 μg each, with a total amount of DNA not exceeding 2 μg per well. Cells were incubated for 6 days, to allow for more constructs to be cut and repaired, at 37 °C in a 5% $CO_2$ humified incubator after the transfection and used for the

following experiments. Because the plasmids do not replicate in the HEK293 cells, to prevent significant dilution of the Sense and the BranchΔ constructs, the plasmid DNA of the Sense and the BranchΔ constructs were extracted after three instead of six days following transfection. In transfections experiments using HEK293 cells and HEK293T cells as control, the human cell were transfected with the same amount of plasmid and incubated for 3 days at 37 °C in a 5% CO$_2$ humified incubator. After 3 days, each HEK293 sample and HEK293T control sample were collected from three wells.

## Assays to test RNase H2 activity from cell extracts to cleave at an rGMP in DNA

A mixture of 2.5 pmol of Cy5 5′-labeled oligonucleotide containing an rGMP (Cy5.3PS.rG, Supplementary Data 1) or the control DNA oligonucleotide (Cy5.3PS, Supplementary Data 1) and 3.75 pmol of complementary oligonucleotide (DNA.comp.3PS, Supplementary Data 1) in 1X Thermopol Reaction Buffer was heat denatured in boiling water for 5 min and cool down slowly to 30 °C to anneal the complementary oligonucleotides. 2.5 pmol of annealed oligonucleotide were incubated with 50 ng of protein extract from HEK-293T RNase H2A wild-type or KO cells prepared by using NP-40 lysis buffer (Alfa Aesar), or just water as the negative control, for 4 h at room temperature. Successively, the mixture was treated with 9.95 ul of formamide (VWR, 0606-100 ML) and incubated for 5 min at 95 °C to denature the double strand substrate, then the mixture was put on ice. The denatured substrate was mixed with 2.22 ul of 10X Orange Loading Dye (LI-COR, C80809-01) and loaded on a 15% 7 M urea denaturing polyacrylamide gel. As a positive control for this experiment, we used 5 units of *Escherichia coli* RNase HII (NEB, M0288L) in place of the protein extract. Results are shown in Supplementary Fig. 4c.

## Quantitative real-time PCR (qRT-PCR)

RNA was isolated from the transfected cells after 6 days using the RNeasy Mini kit (Qiagen) and treatment by DNase (RNase-Free DNase set, Qiagen). RNA was converted into cDNA using QuantiTect Reverse Transcription kit (Qiagen). Customized TaqMan Assay containing primers and probe (Catalog# 4331348, Assay ID APU67JA, DSRED1, Applied Biosystem, Supplementary Data 1) and TaqMan Fast Advanced Master Mix (no UNG, Applied Biosystem) were used for analyzing RNA expression in 96-well plates (Applied Biosystem). The total volume in each well was 20 μl, which consisted of 10 μl of TaqMan Fast Advanced Master Mix, 1 μl of TaqMan Assay (20x) containing primers and probe, 8 μl of nuclease-free water and 1 μl of cDNA. The cDNA level was determined using an ABI Prism 7000 RT-PCR machine (Applied Biosystem). Human GAPD (GAPDH) Endogenous Control (FAM/MGB probe, non-primer limited, Applied Biosystem), which is TaqMan primers and probe, was used for normalization. Values for each sample were normalized by GAPDH (ΔCt), and then each ΔCt value was normalized by ΔCt from the control sample transcribing the spliced transcript (ΔΔCt).

The cDNA synthesized from yeast control RNA samples were used for TaqMan Gene Expression Assays containing primers and probe (catalog# 4351372, Assay ID Sc04167175_s1, HIS3, ThermoFisher) and TaqMan Fast Advanced Master Mix (no UNG, Applied Biosystem) were used for analyzing RNA expression in 96-well plates (Applied Biosystem). UBC6 control (catalog# 4351372, Assay ID Sc04167175_s1, UBC6, ThermoFisher) was used for normalization. The subsequent procedures were the same as previously described.

## Next generation sequencing (NGS) library preparation for HEK-293T & HEK293 cells

Genomic DNA and plasmids were isolated using DNeasy Blood & Tissue kit (Qiagen) from transfected cells after 6 days of incubation at 37 °C in a 5% CO$_2$ humified incubator. The isolated DNA was amplified by a 1st PCR (20x cycles) using F primer 1 (YJo11-14, Supplementary

Data 1) and R primer 1 (YJo15-26, Supplementary Data 1) having 6-nucleotides barcode (NNXXXX), where N is any nucleotide, XXXX is a specific barcode identifying each sample, and a part of adapter sequences for Illumina sequencing. The 1st PCR products were amplified in a 2nd PCR (15x cycles) using F primer 2 (D501-508, Supplementary Data 1) and R primer 2 (D701, 702, 705, 707 and 712, Supplementary Data 1) containing an index and the adapter sequences. The 2nd PCR products were purified by spin column (QIAquick PCR Purification Kit, Qiagen) and then run in the BluePippin (2% agarose DNA size selection cassette, Sage Science) at the Molecular Evolution Core Facility at Georgia Tech to remove the primer dimers. The extraction range was 130–500 bp for the BluePippin. Bioanalyzer High Sensitivity DNA analysis (Agilent) was used to check the sizes of the final sequencing libraries. We used Illumina HiSeq 2×150 for the NGS provided by Admera Health (Biopharma Services, South Plainfield, NJ). NGS libraries for the Antisense and 5′-SplicingΔ constructs were prepared as follows. The isolated DNA was amplified by using DsRed.p-Cis.F3 and DsRed.pCis.R3 primers (Supplementary Data 1) for the 1st PCR (20x cycles), then the amplified products were amplified by using F primer 2 (YJo1,9,10, Supplementary Data 1) and R primer 2 (YJo4-6,8,11, Supplementary Data 1) for the 2nd PCR. The extracted DNAs of the Antisense and 5′-SplicingΔ constructs were used to construct sequencing libraries twice to increase the size of the libraries, using PCR primers with and without a barcode sequence, as described above. Thus, the DNA extracts for the Antisense and 5′-SplicingΔ constructs were sequenced twice.

To construct RNA sequencing libraries, RNA was isolated using the RNeasy Mini kit (Qiagen) from the plasmid-transfected cells after 6 days at 37 °C in a 5% CO$_2$ humified incubator. The isolated RNA was converted to cDNA using the QuantiTect Reverse Transcription kit (Qiagen). The cDNA was PCR amplified by the 1st PCR primers (YJo11-26) by 20x cycles, then the 1st PCR product was amplified by the 2nd PCR primers (D501-508 and D701, 702, 705, 707 and 712) by 15x cycles. All the PCR amplification was done by Q5 High-Fidelity DNA polymerase (New England BioLabs).

## Next generation sequencing (NGS) library preparation for yeast

Genomic DNA from 1.5 ml culture was isolated using the MasterPure Yeast DNA Purification Kit with RNase H treatment. RNA was isolated from 200 μl of the cell culture using RNeasy Mini kit (Qiagen) and treatment by DNase (RNase-Free DNase set, Qiagen). RNA was converted into cDNA using QuantiTect Reverse Transcription kit (Qiagen). The isolated DNA and reverse-transcribed cDNA were amplified by a 1st PCR (20X cycles) using F primer 3 (YLU001-009, YLU011-015, YLU032-033, Supplementary Data 1) and R primer 3 (YLU016-024, YLU026-029, YLU035-036, Supplementary Data 1) having the same design as the F primer 1 and R primer 1. The 1st PCR products were amplified in a 2nd PCR (15x cycles) using F primer 4 (D501-508, Supplementary Data 1) and R primer 4 (D701-702, D704-706, D708, D710, D712, Supplementary Data 1) containing an index and the adapter sequences. The 2nd PCR products were purified by spin column (QIAquick PCR Purification Kit, Qiagen) following BluePippin (2% agarose DNA size selection cassette, selection range 150 bp-700 bp, Sage Science) at the Molecular Evolution Core Facility at Georgia Tech to remove the primer dimers. Bioanalyzer High Sensitivity DNA analysis (Agilent) was used to check the sizes of the final sequencing libraries. Illumina HiSeq 2 × 150 was used for the NGS provided by Admera Health (Biopharma Services, South Plainfield, NJ).

## Trimming of NGS data

The reads in the sequencing libraries of sense constructs (Sense, BranchΔ and pCMVΔ) were tagged by two 6-nucleotides barcodes NNXXXX (5′ end) and YYYYNN (3′ end), where NN is any dinucleotide and XXXX/YYYY represent specific 4-nucleotide barcodes identifying each library. The barcode sequences were located at the 5′ and the 3′

end of the reads. First, the reads were trimmed based on sequencing quality and Illumina adapter sequence using cutadapt 2.10 with the default setting[31]. After trimming by cutadapt, the barcodes were trimmed from the reads. Every forward-strand (reverse-strand) read was required to have sequence NNXXXXTTCAAG (NNXXXXCTTGAA) in its first 12 nucleotides, where TTCAAG (CTTGAA) are the first 6 nucleotides of the forward (reverse) reference sequence. To check the barcode located at the 3′ end of sequencing reads, the length of the reads was measured first. If the reads were longer than 140 nucleotides and did not contain the last 6 nucleotides of the reference sequence (ACTTCC for forward-strand and CTTGAA for reverse-strand reads) in their last 12 nucleotides, the 3′ barcode check was not done, since the 3′ barcode may not have been captured due to the maximum length of reads (150 nucleotides) for Illumina HiSeq 150×2. However, if the length of forward-strand (reverse-strand) sequencing reads were shorter than 140 bp or contained ACTTCC (CTTGA) in their last 12 nucleotides, the read was required to have ACTTCCYYYYNN (CTTGAYYYYNN) in its last 12 nucleotides, where ACTTCC (CTTGA) are the last 6 nucleotides of the forward (reverse) reference sequence. If either barcode requirement failed, the read was discarded, otherwise the read was accepted with the barcodes removed.

Trimming for the Antisense and 5′-SplicingΔ NGS libraries that did not have barcode sequences was only done by cutadapt 2.10 with the default setting. The FASTQ files for the two sequencing libraries derived from the same DNA source of each Antisense and 5′-SplicingΔ sample were concatenated to generate one FASTQ file for each sample.

The yeast sequencing libraries have the same format of barcode with the sense constructs but different sequences. The forward-strand read has NNXXXXGCCTCT (5′) and CTACTGYYYYNN (3′), and the reverse-strand has NNXXXXCAGTAG (5′) and AGAGGCYYYYNN (3′). The same procedure of trimming with the sense constructs was applied to the yeast sequencing libraries with these sequences.

## RNA sequencing data analysis

The trimmed FASTQ files were aligned to the reference sequence of the corresponding construct using hisat2 2.2.1 with the default setting[32]. The SAM files generated by hisat2 were used to categorize the sequencing reads as 'spliced transcript', 'non-spliced transcript', 'alt-spliced transcript', 'non-canonical alt-spliced transcript', and 'unaligned'. Using the SAM files, the occurrence of splicing was checked first by sequence alignment. The reads without any splicing event were counted as non-spliced transcripts, and the reads having a splicing event with the same length as the intron were counted as spliced transcripts. Splicing on other sites starting with the dinucleotide GT and ending with the dinucleotide AG were counted as alt-spliced transcripts. The other sequencing reads having splicing that did not start with GT- and end with -AG were counted as non-canonical alt-spliced transcripts. All the other sequencing reads not aligned to the reference sequence by hisat2 were counted as unaligned.

## Identification of reads as DSB repair by NHEJ using sequence alignment

Each sequence read in each NGS library was aligned to the corresponding reference sequence. The reference sequence for 1-DSB experiments was the entire construct sequence between the two sequencing primers, while for 2-DSB experiments it was the construct sequence between the two sequencing primers with the region between the two DSBs deleted. We used the Bowtie 2 2.4.1 software with the default setting to generate SAM files[33]. On each SAM file obtained from Bowtie 2, we used a custom script that performed the following data processing. We discarded reads that Bowtie 2 failed to align and those that did align but not with the first base pair of the reference sequence. We discarded short reads that were less than the minimum expected length (130 bp for 1-DSB system and 50 bp for the

2-DSB system). To analyze the sequence variations at the DSB site, we realigned each read to the reference sequence so that the in/dels encapsulate (i.e., are adjacent to or around) the DSB position. If this alignment had the same or lower number of mismatches than the original alignment, this new alignment was kept and replaced the original one. We kept only the reads which, after the previous steps, resulted in an alignment such that in/dels were both consecutive and encapsulated the DSB position. For the 1-DSB system, the resulting libraries contained at least 93% of the original reads; for the 2-DSB system, the resulting libraries contained between 14% and 50% of the original reads. In the 1-DSB libraries, the error-free NHEJ/uncut reads were the resulting reads that had only substitutions and no in/dels, while the NHEJ with in/dels reads were those with in/dels. The frequencies of the groups were the respective number of reads in the groups divided by the number of total reads in the library. The resulting frequencies are only reported for the 1-DSB experiments (see Categorizing 1-DSB sequencing reads into three groups: NHEJ in/dels, Error-free NHEJ/uncut, and MMEJ) since an alternative method based on read length was used to determine the NHEJ frequencies for double-strand gap repair in 2-DSB experiments (see Categorizing 2-DSB sequencing reads into three groups: sequence with intron, DSB or double-strand gap repair via NHEJ, and DSB or double-strand gap repair via MMEJ).

## Identification of microhomology pairs and calculation of MMEJ frequency

Microhomology pairs were defined as two identical DNA segments on the reference sequence that were at least 3 bp long. We identified all microhomology pairs that had one segment upstream and another downstream of the DSB and were between the primer sequences. For the sense constructs, we excluded from the analysis pairs whose segments overlapped the branch site (55 bp) since that region would not be present in the BranchΔ reference sequence. For the antisense constructs, we excluded from the analysis pairs whose segments overlapped the 5′-splice site since that region would not be present in the 5′-SplicingΔ reference sequence. We also only considered microhomology pairs whose matching segments were maximal (i.e., could not be extended to a pair of longer matching segments). Considering that the reference sequence is subdivided into 3 regions (Exon1, Exon2, and Intron), the microhomology pairs were categorized into 4 groups (Exon1-Exon2, Exon1-Intron, Exon2-Exon1, and Exon2-Intron) depending on the regions in which their matching segments were located and on the position of the DSB (Supplementary Fig. 9a). A name was assigned to each microhomology pair such that corresponding microhomology pairs in the different constructs (e.g., Sense, BranchΔ, and pCMVΔ, or Antisense and 5′−SplicingΔ) had the same name. The microhomology pairs with "R" in their names are located on the reverse strand. See Supplementary Figs. 9b, c and 11b, c for the locations and nucleotide sequences of the microhomology pairs and the regions Exon1, Exon2, and Intron on the reference sequence. In this analysis, a read was defined to have DSB repair by MMEJ if it contained the exact sequence of a microhomology deletion signature. A microhomology deletion signature was defined as the DNA sequence that results by concatenating the 10 nucleotides upstream of the upstream matching segment, the matching segment, and the 10 nucleotides downstream of the downstream matching segment (i.e., this is a subsequence of the sequence that results by deleting all nucleotides on the reference sequence between the matching segments and deleting one of the matching segments). We detected all MMEJ events from the raw reads and calculated their frequency, which is the count of MMEJ events divided by the total read count. The corresponding Python 3 scripts to calculate the MMEJ frequency are available on GitHub.

## Bar plots and box plots for MMEJ frequencies

Bar plots and box plots were generated to compare the MMEJ frequencies between different constructs of the sense or antisense systems. In the bar plots, the mean frequency of samples is shown, and the standard deviation is indicated by whiskers. In the box plots, the median of the points is shown as the middle line of the box. The first and third quartiles are indicated by the box frames and the whiskers represent the largest point not more than 1.5 interquartile range (IQR) beyond the box frame. All data points outside the whiskers are classified as outliers and shown as diamond points. Two-tailed Mann-Whitney $U$ tests were performed to compare the frequencies of the different constructs.

## Categorizing 1-DSB sequencing reads into three groups: NHEJ in/dels, Error-free NHEJ/uncut, and MMEJ

The sequencing reads were classified into three categories: NHEJ in/del, Error-free NHEJ/uncut, and MMEJ. The NHEJ in/del group was defined as reads that, when aligned to the reference sequence, have in/dels that are consecutive and adjacent to the DSB site. The Error-free NHEJ/uncut group was defined as reads that align to the reference sequence without any in/dels. The reference sequence for 1-DSB experiments was the entire construct sequence between the two sequencing primers, while for 2-DSB experiments it was the construct sequence between the two sequencing primers with the region between the two DSBs deleted. Due to possible sequencing errors, in both the NHEJ in/del and Error-free NHEJ/uncut groups, substitutions were not considered[34]. The MMEJ group was defined as reads that contained the exact sequence of a microhomology deletion signature. A microhomology deletion signature was defined as the DNA sequence that results by concatenating the 10 nucleotides upstream of the upstream matching segment, the matching segment, and the 10 nucleotides downstream of the downstream matching segment (i.e., this is a subsegment of the sequence that results by deleting all nucleotides on the reference sequence between the matching segments and deleting one of the matching segments). There were two subgroups of MMEJ, Exon1(2)-Exon2(1) MMEJ and Exon1(2)-Intron MMEJ, depending on the location of microhomologies and the location of the cut. The frequency of each category was defined as the number of reads in the category divided by the number of total reads in the library. The NHEJ in/del and Error-free NHEJ/uncut groups were non-overlapping. However, the frequency of overlap between the MMEJ and the two other groups was less than 0.00003% in most libraries (likely sequencing errors), except in BranchΔ, sgRNA B experiments where the frequency was less than 0.04% due to some microhomologies that were close to the DSB site and could be classified as both NHEJ and MMEJ deletions (see Supplementary Fig. 9c). For more details about the categorization see Identification of reads as DSB repair by NHEJ using sequence alignment (groups NHEJ in/del and Error-free NHEJ/uncut) and Identification of microhomology pairs and calculation of MMEJ frequency (group MMEJ).

The unclassified reads, which are those that were not among the three groups described above, were further analyzed by realigning them to the corresponding reference sequence using the Bio.Align package of Biopython[35]. The control No-DSB libraries contained between 0.41% and 1.12% unclassified reads, which were mostly due to sequencing errors including insertions and deletions 4 or more nucleotides distant from the DSB sites. The 1-DSB libraries contained between 1.53% and 5.02% unclassified reads, though when discounting the type of reads that were also in the No-DSB controls, this frequency was between 0.86% and 3.70%. The unclassified reads that were in 1-DSB libraries but not like those in the No-DSB control libraries fell into four categories. The first category contained reads that had in/dels shifted up to 3 nucleotides from the DSB site (frequencies between 0.10% and 1.11%). These sequences could be explained by NHEJ repair

following a non-blunt end cleavage by Cas9. The second, third, and fourth category contained deletions like the MMEJ-type deletions (i.e., the range of deleted nucleotide touched the DSB site) where the start and end of the deletion were, respectively, between the exon and exon (exon-exon, frequencies between 0.16% and 2.27%), the exon and intron (exon-intron, frequencies between 0.05% and 1.13%), and the exon and branch site (exon-branch, only for Sense and pCMVΔ, frequencies between 0.09% and 1.01%). In some cases, there were di- or mono-nucleotide matches at the ends of the deleted segments in exon-exon or exon-intron deletions, or even larger matches in exon-branch deletions, but such deletions did not follow the definition of exon-exon and exon-intron MMEJ repair used here (see Identification of microhomology pairs and calculation of MMEJ frequency). Statistical tests revealed that the exon-exon and exon-intron deletion frequencies followed a similar pattern as the exon-exon and exon-intron MMEJ frequencies: all exon-exon deletions were either more frequent in the constructs with splicing (Sense and pCMVΔ) or were not significantly different, and all exon-intron deletions were either more frequent in the construct without splicing (BranchΔ) or were not significantly different (see Supplementary Data 6).

## Categorizing 2-DSB sequencing reads into three groups: sequence with intron, DSB or double-strand gap repair via NHEJ, and DSB or double-strand gap repair via MMEJ

To determine whether the reads contain or not the intron, the reads were filtered by length. Illumina HiSeq 2×150 was used for the NGS in which the sequencing reads have a maximum length of 150 bp. The fragments in the NGS libraries from the sense (antisense) constructs are 229 (253) bp with the intron and 118 bp without the intron if there was no mutation. The fragments in the BranchΔ (5′-SplicingΔ) libraries are 174 (247) bp with the intron and 118 bp without the intron. If the sequencing reads contain the intron, the length of the reads reaches the maximum length of the Illumina reads, but if not, the length of the sequencing reads is around 118 bp with variations at the DSB site caused by the repair. Therefore, all the reads that were longer than 130 bp were classified as sequences with intron. The reads shorter than 130 bp were classified as DSB/double-strand gap repair via NHEJ or MMEJ by checking whether the reads were the products of MMEJ (see Identification of microhomology pairs and calculation of MMEJ frequency). The products of MMEJ were classified as DSB or double-strand gap repair via MMEJ, and all the other reads shorter than 130 bp were classified as DSB or double-strand gap repair via NHEJ.

The category of repaired DNA products following DSBs by sgRNA E and J was determined by alignment using Bowtie2. Three sequences were used as references for Bowtie2 alignment with the default setting: no DSB, gap repair, and flipped segment. Each reference sequence was the product of perfectly repaired sequence in each scenario without indels. All sequencing reads were aligned with all three reference sequences to find the best match and thus be categorized into different groups. The products of MMEJ were classified as gap repair via MMEJ, and all the other reads that aligned with gap repair sequence were classified as gap repair via NHEJ. The reads that aligned with no DSB or flipped-segment were combined together and classified as segment retention.

In the yeast constructs, the length filter could not distinguish whether the reads contain the intron or not, because the yeast libraries have longer sequences (425 and 392 bp with the intron, 196 bp without the intron) than the sense and antisense libraries. The sequencing reads from the yeast constructs were classified as not containing the intron if the reads contained any 20-bp segment of the 3′ exon sequence. The other sequencing reads that did not have the part of the other exon were classified as containing the intron. All the other procedures were the same for determining the frequencies of NHEJ and MMEJ after this step.

## Calculating the frequency of flipped-intron insertion following 2 DSBs

The frequency of flipped-intron insertion was determined by detecting 20 bp of the complementary sequence of the intron sequence specifically located 10 bp away from the DSB site so that it could be captured within the sequencing reads. The antisense constructs have a sequence that is different from the sequence obtained by flipping the intron in the sense constructs (Supplementary Fig. 1); thus, reads of the antisense constructs cannot be confused for reads of the flipped intron of the sense constructs.

The frequency of flipped-segment following 2 DSBs by sgRNA E and J was determined using Bowtie2 alignment with perfectly repaired flipped-segment reference sequence.

## Statistical analysis of data

Two-tailed Mann–Whitney $U$ test was used for statistical analysis of sequencing data.

## Obtaining DSB-sequence windows for variation-distance graphs and variation-position histograms

For each sequence library, we use the reads categorized as NHEJ with in/dels or Error-free NHEJ/uncut in Identification of reads as DSB repair by NHEJ using sequence alignment. To concentrate on variations near the break site, we obtained a *DSB-sequence window* from each NHEJ or uncut read corresponding to the nucleotides that align with positions −10 to +10 relative to the DSB site on the reference sequence. To ensure the window reflected only variations near the DSB, we additionally required it to be flanked by two anchor sequences. The left anchor sequence corresponded to nucleotides −30 to −11 relative to the DSB site on the reference sequence, and the right anchor sequence corresponded to nucleotides 11 to 30 relative to the DSB site on the reference sequence (Supplementary Fig. 1). We discarded reads whose left or right anchor sequences had alignments with ≥2 mismatches or ≥1 in/del. Because substitutions are common NGS sequence errors[34] and substitutions were also abundant in the No-DSB negative control samples (Supplementary Fig. 5), we only considered in/dels in the DSB-sequence windows in the NHEJ analysis. To do so we replaced all alignment substitutions with the corresponding nucleotide of the reference sequence. Then the sequence reads with the same DSB-sequence window were combined by summing their frequencies. Since the Antisense and 5′-SplicingΔ libraries were sequenced twice, we merged the two outputs of the previous steps and recomputed their frequency accordingly. This process of obtaining windows discarded at most an additional 3.8% of each library. For each experiment, we obtained a table of DSB-sequence windows by assigning each DSB-sequence window the average frequency of all repeats of the same experiment. These tables were then used to construct the variation-distance graphs and variation-position histograms (see the captions for Fig. 2 and Supplementary Figs. 5 and 7). The DSB-sequence windows that had a frequency ≤$10^{-5}$ in at least one of the repeats were not used in the variation-distance graphs to avoid visual clutter. The control sequences 30 bp downstream from the DSB site (data shown in Supplementary Fig. 5a, b) were obtained in the same manner except the DSB site was assumed to be 30 bp downstream from its actual position.

## Testing Sense/BranchΔ or Antisense/BranchΔ-frequency ratios between cell types

The Mann-Whitney $U$ test was used to compare the frequency ratio, Sense divided by BranchΔ or Antisense divided by BranchΔ, in different cell types. For a single category of repair (e.g., NHEJ, MMEJ, with intron, without intron, etc.,), let $x_{ijk}$ be the observed frequencies for $i = 1,2$ (for two different cell types), $j = 1,2$ (for two different constructs), and $k = 1,2,3,4$ for the biological replicates (ordered by numerical order of library ID). Define the frequency ratios by $r_{ik} = \frac{x_{i1k}}{x_{i2k}}$.

Then the two-tailed Mann-Whitney $U$ test was used to compare the four ratios in cell type 1, $\{r_{11}, r_{12}, r_{13}, r_{14}\}$, with the four ratios in cell type 2, $\{r_{21}, r_{22}, r_{23}, r_{24}\}$. Depending on the figure/table, $i = 1$, and $j = 1$, in this paragraph may indicate the following (figure labels shown in parenthesis):

- HEK-293T wild-type (WT, $i = 1$), HEK-293T RNase H2A knock-out (KO, $i = 2$), Sense ($j = 1$), and BranchΔ($j = 2$).
- HEK-293T RNase H1 normal expression (Control, $i = 1$), HEK-293T RNase H1 overexpression (H1 OX, $i = 2$), Sense ($j = 1$), and BranchΔ ($j = 2$).
- Yeast wild-type (WT, $i = 1$), yeast *rnh1 rnh201*-null (*rnh1 rnh201*, $i = 2$), Antisense ($j = 1$), and BranchΔ ($j = 2$).

See the corresponding figure/table labels and captions to determine in which cell types and constructs the ratios are being compared.

## Calculating the frequency of RNA-templated DNA DSB repair for yeast samples

Here RNA-templated DNA DSB repair (R-TDR) is defined as the mechanism that produces an identical sequence to the template RNA (with Ts in place of Us). To calculate the frequency of R-TDR, the spliced transcript sequence was searched for in each sequencing library to find reads that contain an identical copy of the transcript RNA. To calculate the frequency, the number of reads containing this sequence was divided by the total read count.

## Reporting summary

Further information on research design is available in the Nature Portfolio Reporting Summary linked to this article.

## Data availability

The data supporting the findings of this study are available within the paper and its Supplementary Information files. The NGS data generated in this study have been deposited in NCBI's Sequence Read Archive (SRA) under accession code BioProject PRJNA883674. Source data are provided with this paper.

## Code availability

Customized Python 3 scripts for all NGS data analysis in this study are available on GitHub under GPLv3.0 license (https://github.com/xph9876/RNA-mediated_DSB_repair).

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

## Acknowledgements

We are grateful to D. Garfinkel for plasmid pSM50, and to M. Jasin for the enhanced empty gRNA expression vector. We thank N. Djeddar and A. Bryksin for their technical assistance with sequencing. We thank K. Lobachev and J. Dahlman for critical reading of the manuscript, and all the members of the Storici and Jonoska groups for assistance and feedback on this research. We acknowledge funding from the National Institute of General Medical Sciences (NIGMS) of the NIH grant GM115927 (F.S.), the National Institute of Environmental Health Sciences (NIEHS) of the NIH grant ES028271 (Z.F.P.), the National Science Foundation grant MCB-1615335 (F.S.), the Howard Hughes Medical Institute Faculty Scholar grant 55108574 (F.S.), grants from the Southeast Center for Mathematics and Biology NSF DMS-1764406 and the Simons Foundation grant 594594 (F.S. and N.J.), the NSF grants CCF-2107267 and DMS-2054321 (N.J.), and the Georgia Tech Tom and Marie Patton Distinguished Professor grant DE00024645 (F.S.) for supporting this work.

## Author contributions

F.S. together with Y.J., M.M.F. and N.J. conceived the project and designed experiments. F.S. wrote the manuscript with help from Y.J., M.M.F., T.C., P.X., N.J., and help from all authors. Y.J. performed most of the experiments in human cells. Y.L. handled significant work in both human and yeast cells. C.M., S.B., S.M., and Y.Z. helped in the experimental part. Y.J., Y.L., M.M.F., T.C., and P.X. worked on the sequencing data analyses. M.M.F. and T.C. worked on the variation-distance graphs. V.S.P. and Z.F.P. constructed the RNase H2A KO cell lines. All authors commented on and approved the manuscript.

## Competing interests

The authors declare no competing interests.
