## [Peer Review File · Nature Communications]

RNA-mediated double-strand break repair by end-joining mechanismsReviewers' comments:

Reviewer #1 (Remarks to the Author):

In sum, the authors describe unusual phenomena where products of DSB by end joining pathways appear effected by the identity of transcribed RNA. For example, after cutting at one side of an intron, repair products consistent with inaccurate NHEJ at that DSB are slightly enriched by an unspliced transcript. Perhaps more compelling, after cutting at both sides of an intron, repair products consistent with NHEJ after deletion of the intron are more frequent when a version of the transcript with the intron spliced out is present. Effects of transcription are also increased in RNaseH2A deficient cells, suggesting they are mediated by a RNA-DNA hybrid.

The authors refer to transcript RNA playing a “significant role in genome stability”. This claim would need support by experiments addressing chromosomal repair.

Unfortunately, end joining is measured only by sequencing. There was no attempt to validate repair according to pathway by genetic means (or using an inhibitor). Most importantly, they do not consider other outcomes that would require very different interpretation of the data, including still broken DNA ends, and especially uncut substrate and accurate end joining. Importantly, a control asserting equal cutting by Cas9 in vitro isn't useful, since the authors are studying effects of transcription of DSB target sites in cells. They must be able to exclude a possible influence on cutting by transcription or r-loops.

The latter issue is particularly notable, given the following statement of the authors: “The frequencies of error-free NHEJ/uncut constructs did not compensate for such difference in the NHEJ in/del frequency between the non-splicing and the splicing constructs (Extended Data Figure 6a, b).” This seems like gaslighting, since the data show otherwise. For extended 6a, NHEJ in/del is increased 1% in Branchdel (non-splicing, 4% to 5%), while uncut constructs are reduced by about 1% (94.9% to 94%) in the same sample.

The final paragraph of the results appears non-sensical. It is led by the phrase “... we search for the R-TDR signature...”. What exactly is this signature? In methods, the authors indicate they identify R-TDR according to this sentence: “We consider RNA-templated DNA DSB repair to be the mechanism that produces an identical sequence with the template RNA (except for Ts in place of Us).” This seems to imply the DNA repair product is made from an “RNA template”. Since the RNA template is made from the uncut original DNA template, how can the resulting DNA product sequence be distinguished from uncut DNA sequence?

Minor concerns.

- 1) The authors employ a method of data representation (“individual variation-distance graphs”) that seems needlessly complicated, and confound this problem by employing jargon-heavy descriptions. Worse, these graphs serve little obvious purpose, since individual “DSB-sequence windows” are either broadly enriched or broadly depleted by a given experimental variable. The bar graphs are sufficient. The sole exception (a few rarely used DSB sequence windows in the Fig 2C RNaseH2A KO) isn't interpreted.
- 2) The authors do not complement RNaseH2A deficiency to validate this result.
- 3) The methods are generally oddly chosen. For example, why assess products 6 days later, since repair should be finished far earlier?

Reviewer #2 (Remarks to the Author):

In this manuscript, Jeon et al used a DsRed-containing plasmid system, in combination with targeted amplicon deep sequencing of the repair junctions, to study Cas9-induced double strand break repair in HEK293T cells and test the hypothesis that an RNA transcript could mediate DSB repair in human cells. On the basis of the data, the authors stated that “both coding and non-coding transcript RNA facilitates DSB repair in a sequence-specific manner in human cells” in the abstract and concluded that RNA transcripts can promote NHEJ or MMEJ of Cas9-induced DSBs or even serve as a template for NHEJ in human cells. Although a large amount of data are provided with some particular design of the plasmid system in an attempt to support the authors' claims in this manuscript, the approaches taken and evidence provided are too weak and not convincing. It provided no direct answers into the question whether or not RNA-mediated DSB repair really occurs. I will explain these major issues as well as some minor concerns in details below.

1. The #1 issue of this manuscript is that this manuscript is completely based upon the plasmid assays. There are several problems with this approach. Firstly, many copies of the reporter plasmids (i.e., the Sense, Branch Δ and pCMV Δ constructs) were delivered into the cells by transient transfection. In addition to Cas9-induced DSBs, the plasmids often harbor spontaneous nicks and DSBs and even more after being delivered into the cells. Due to spontaneous nicks and DSBs along with Cas9-induced DSBs, recombination between different copies of these plasmids could complicate the analysis and outcomes of the RNA-mediated DSB repair assays, potentially producing misleading results. Second, the plasmids the authors chose to use contains the SV40 origin and the cells used also express SV40 large T antigen. These plasmids transfected would replicate in HEK293T cells. Replication of these plasmids can cause additional DSBs (one-ended and two-ended DSBs) and nicks, complicating the outcomes of the plasmid assays further. It is not clear why the authors added DNA replication into this plasmid approach. In addition, if RNA-mediated DSB repair occurs in human cells, it should occur in the chromatin context. However, the plasmids transiently transfected into 293T cells do not form chromatin structure resembling that of the human genome. Any meaningful observation on RNA-mediated DSB repair requires further confirmation in the context of chromatin. Because of these problems, there is a possibility that RNA-mediated DSB repair observed in this manuscript is an experimental artifact and does not really take place in the genome. That is not to mention the over-interpretation of the data by the authors throughout the manuscript.

2. The weakness in the plasmid approach can be easily addressed with the DsRed reporters integrated in a single copy into specific sites of the genome in mammalian cells. In this setting where a DSB can be induced at a specific genome site, coding and non-coding RNA (antisense RNA) can be provided either to mediate DSB repair or as a template for DSB repair. Any alterations on RNA transcripts could be introduced to determine whether the pairing between DNA and complementary RNA really facilitates DSB repair or even complementary RNA directly serves as the template for DSB repair. However, the manuscript has not done this type of experiments. Furthermore, the fluorescence of DsRed in the reporter plasmids was neither used nor essential for the assays in this manuscript. The authors instead used NGS to analyze the efficiency of RNA-mediated DSB repair and RNA-templated repair. Therefore, a couple of endogenous genomic sites in the cells could be a better choice than the DsRed plasmids in this manuscript for similar analysis. Surely, the results from these natural sites could be more convincing in either supporting or rejecting the authors' conclusions.

3. The authors over-interpreted the data throughout the manuscript from Figure 2-Figure 5,

on page 4 lines 110-113, on page 5 lines 145-146 and lines 157-158, on page 6 lines 178-181 and on page 7 lines 221-222. There is no direct evidence to support these statements. For instance, on page 4 lines 110-113, the authors stated “Overall, these results provide evidence that the sequence of a transcript RNA, its complementarity with the DSB ends, and the level of transcription directly influence the efficiency of DSB repair by NHEJ and the frequency of the repaired products by NHEJ.” However, the data only show that these three different constructs (i.e., the Sense, Branch Δ and pCMV Δ construct) exhibit a difference in the efficiency of Cas9-induced NHEJ. This difference could be caused by the variations in target DNA sequences, DNA metabolism at the target, and the target binding, cleavage and post-cleavage dissociation of the Cas9-sgRNA complex. Recent studies have shown that transcription and DNA replication affect the target binding, cleavage and post-cleavage dissociation of the Cas9-sgRNA complex, thus adding a layer of control in DSB repair pathway choices and influencing the repair outcomes (Clarke et al, Mol Cell 71:42–55.e8, 2018; Ivanov et al, PNAS 117:5853–5860, 2020; Liu et al, Genome Biol 23:165, 2022). Thus, the results of DSB repair in the plasmids may have nothing to do with the RNA transcripts, but be influenced by these above possibilities that this manuscript did not exclude. The statement “The results were stronger in the RNase H2A KO cells, suggesting direct RNA-DNA interaction (Figure 4a and Supplementary Table 3)” on page 5 lines 157-158 was also over-interpreted. Direct evidence is needed for the enrichment of direct RNA-DNA interaction. Another example of over-interpretation is the statement on page 6 lines 178-181. The slightly higher frequency of NHEJ in/dels in gap repair in the Sense construct only reveals the association with the Sense construct. Whether or not spliced RNA is used for gap repair has not been determined in this manuscript. In summary, in order to support the statements on RNA-mediated NHEJ, RNA-mediated MMEJ, RNA-mediated gap repair and RNA-templated repair in the manuscript, the authors have to provide more direct and stronger evidence.

4. The data analysis and presentation in this manuscript are quite complicated and confusing to me. It is really difficult to read through. Based on the Method description, the frequencies of MMEJ and RNA-templated repair were calculated as the number of reads for MMEJ and RNA-templated repair divided by the total read count. However, it is not clear how to calculate the frequencies of NHEJ. I assume that they were calculated as the number of reads for NHEJ divided by the total read count. Further, it is not clear whether NHEJ measured includes NHEJ products with longer than 130 bp (with intron), NHEJ products with shorter than 130 bp without microhomology or with less than 3-nt microhomology or both types of NHEJ products. In addition, given so many microhomology pairs used, the frequency of MMEJ for individual microhomology pairs was surprisingly high as compared to the frequency of total NHEJ. For example, the frequency of individual MMEJ Exon1-Exon 2 (EE4_GACGGC) was about 1% for the Sense construct and the frequency of NHEJ in/dels for the same Sense construct was about 4%. Lastly, considering the variations in DNA sequences and the activities of transcription and DNA replication among the Sense, Branch Δ , and pCMV Δ construct, it is problematic to use only two individual MMEJ products to represent overall MMEJ in Figure 3a. It would be better if the frequencies of overall MMEJ had been given and compared in Figure 3a.

5. In Figure 1, in order to minimize the effect of the sequence variations between the reporter plasmids on DSB repair, the Branch Δ plasmid could be replaced by keeping the Branch site but mutating the splicing site GT/AG. In addition, sgRNA A or B without Cas9 was used as a control as stated on page 3 lines 78-79; however, Cas9 with the sgRNA empty vector or a sgRNA scramble vector should be a better control.

6. In Extended Data Fig. 9, it is clear that PCR and targeted amplicon deep sequencing are biased towards to deletion on the intron side due to the primer design. In our experience, the length of deletions in Cas9-induced NHEJ is largely symmetric among a large pool of NHEJ

products. It is unclear whether this bias interferes with the analysis of MMEJ.

7. Although the authors provided some data to support the hypothesis that RNase H2A may disrupt RNA-mediated NHEJ, RNA-mediated MMEJ, RNA-mediated gap repair and RNA-templated repair, the observed effects of RNase H2A KO on NHEJ, MMEJ, gap repair and RNA-templated repair are not always or strongly consistent with the hypothesis. For example, in Figure 2a, RNase H2A KO does not increase NHEJ in/dels in either of the Sense, Branch Δ , and pCMV Δ constructs. In Figure 3a and 3b, RNase H2A KO does not enhance MMEJ with sgRNA A in either of the Sense, Branch Δ , and pCMV Δ constructs. In Figure 5a panel 2 vs. 4, RNase H2A KO does not enhance RNA-templated repair with sgRNA B in the Sense construct. In addition, wild-type and RNase H2A KO should have been placed together for direct comparison with either sgRNA A or B in these analyses.

8. In Figure 4, paired Cas9-sgRNAs are used to induce a “gap”. However, while simultaneous cleavage of target DNA by paired Cas9-sgRNAs could generate a gap and pop-out of the intron, sequential cleavage would not generate a gap and likely retains the intron. For a better analysis, possible NHEJ outcomes from simultaneous or sequential cleavage by paired Cas9-sgRNAs should be classified as did in a previous study (Guo et al, Genome Biol 19:170, 2018). In addition, accurate NHEJ of DSBs generated by simultaneous cleavage of target DNA by paired Cas9-sgRNAs can be distinguished from the uncut target and should be specifically examined. The term of “gap repair” is different from “gap repair” generally used in the field of DNA repair. This should be noted in the main text of the manuscript.

9. The authors performed the NHEJ analysis with antisense RNA in Figure 4, but draw a conclusion mostly as non-coding RNA. Antisense RNA and non-coding RNA are not the same. This extension from antisense RNA to non-coding RNA should have been more cautious.

10. In Figure 5, the frequency of “RNA-templated repair” is extremely low and around 10^{-5} . This frequency is even lower than the substitution error rate ($\sim 10^{-3}$ - 10^{-4}) of NSG itself. It is possible that NHEJ could generate such a low level of the same products as the RNA-templated repair. In order to identify the product of true “RNA-templated repair”, it could be ideal that the RNA-DNA hybrid over the DSB site is at least detected. In addition, because the Branch Δ construct produces no spliced RNA, the products of “RNA-templated repair” presented for this construct should be the products of NHEJ and be subtracted from those of the Sense and pCMV Δ constructs to get accurate measurement if RNA-templated repair were real.

11. This manuscript should address or discuss a couple of issues related to the mechanisms of RNA-mediated repair and RNA-templated repair. For example, how do spliced and non-spliced RNA mediate DSB repair or serve as a template for DSB repair? When both spliced RNA and non-spliced RNA (i.e., pre-mRNA) are provided in the nucleus, how do cells choose the RNA template between spliced and non-spliced RNA for DSB repair? It is also unclear whether cleavage by Cas9 occurs before or after transcription for RNA-mediated repair or RNA-templated repair. It is intuitive to speculate that cleavage by Cas9 occurs just after transcription for RNA-mediated repair or RNA-templated repair. It would be better that this were discussed. If RNA transcripts mediate DSB repair or serve as a template for DSB repair, how much pairing between RNA and DNA at the broken site is required? It is unfortunate that complementarity between RNA and DNA was little addressed in this manuscript except for the disruption of RNA splicing.

12. In this manuscript, RNA sequencing is included in the Methods section. However, no results are found from RNA sequencing in the manuscript.

13. In Extended Figure 5a and 5b, a significant level of substitutions appears to occur in the setting of the no-DSB control. Why is that?

Reviewer #3 (Remarks to the Author):

DNA double strand breaks (DSBs) are among the most lethal form of DNA lesion; however, the mammalian cells employ multiple DSB repair pathways, such as HR, NHEJ and MMEJ. These repair pathways are highly complex, multistep process and distinct set of proteins are involved in each pathway, and the repair could be error-free or error-prone. In addition to protein factors, a plethora of studies are now showing the role of RNA in DSB repair. Storici's group have earlier provided evidence of RNA-mediated genetic recombination and published several papers and reviews on this topic. The present manuscript by Jeon et al. provided ample data to establish the role of RNA in both NHEJ and MMEJ. The authors have shown that both coding and non-coding transcript RNA facilitates DSB repair in a sequence-specific manner in human cells. However, additional experiments along with some clarifications could help strengthen the mechanistic details regarding the role of RNA in DSB repair pathways.

1. The experimental design seems to have some inherent limitation. Multiple factors are involved that each may add up to the outcome of the experiment and interpretation of results.

a. The assay involved transfection of multiple plasmid constructs, such as, Cas9/gRNA to induce DSBs at specific site along with plasmid substrate, and DSB-containing plasmids are then allowed to repair for 6 days. It is difficult to envisage how CRISPR/Cas9 mediated DSBs can generate a gapped substrate in a synchrony. If one site is preferentially cut involving a specific sgRNA, then cells will attempt to repair the DSB.

b. Even if such gapped plasmid is generated in cell (which would be expectedly low in frequency) and the gap is repaired by the DSB repair pathways, then the residual CRISPR/Cas9 will continue to induce DSBs.

c. The other concern is that the DSB-containing plasmid construct is not replication defective, enabling a plasmid to clonally expand itself and thus, making it difficult to interpret the results.

d. The authors have mostly analyzed the in/dels (i.e., error prone DSB repair) as a marker of NHEJ and have themselves suggested that ".....it was not possible to distinguish the sequence of error-free repair by NHEJ from that of the uncut Constructs". This remains a concern of the study.

e. Once the DSBs are induced, then how the full-length RNA can be produced to serve as template for gap filling. There is a strong possibility that elongating RNA polymerase II will stall at the first site of DSB repair (exon 1). Thus, in the case of gap repair, transcript would not be available as a template.

The authors need to provide further clarifications regarding those concerns to establish this assay as a reliable system to measure RNA-mediated DSB repair.

2. The authors have proposed LINE 1 as the reverse transcriptase required for R-TDR in mammalian cells. LINE 1 expression is usually highly regulated and restricted in somatic tissues and cells. However, its expression is generally higher in tumors and cancer cell lines. Thus, LINE 1 is unlikely to be the major RT in normal somatic cells for such major repair pathways. Nonetheless, to conclusively prove any involvement of LINE1 in such repair, LINE1 inhibitor should be used to evaluate its potential role in R-TDR.

3. Although rarely, Cas9-induced DSBs can lead to unwanted in/dels, particularly in HEK293T cells. Efficiency of L1 transposition in HEK293T is lot higher even when compared to highly proliferative HeLa cells (PMID: 29016854). Hence, HEK293T cells may not be appropriate for such assay, as formation of in/dels can be misinterpreted as DSB repair

activity.

4. Three different scenarios have been tested for the role of RNA in NHEJ: a plasmid construct without capable of splicing (Branch Del), had higher frequency of NHEJ in/dels than the constructs with splicing (Sense and pCMV Del) in wild-type and more evidently in the RNase H2A KO cells. A higher frequency of in/dels for Sense than pCMV Del (in DSB by sgRNA B). The majority of in/dels in RNase H2A KO cells had higher frequency for the Branch compared to the Sense construct. However, the specific implications of these individual observations have not been clearly explained.

5. Recent report has clearly indicated that RT activity of Pol θ plays a critical role in MMEJ (PMID: 34117057). Based on this result, authors need to evaluate such R-TDR or R-MMEJ in Pol θ depleted cells. Also, the authors' conclusions are mostly based on a massive scale of NGS data to identify the signature motifs for NHEJ or MMEJ. However, they have not spent much effort to biochemically characterize and confirm whether the DSB repair took place via these pathways. In this context, at least one key enzyme for NHEJ (such as Ligase IV) and MMEJ (Lig III or XRCC1) should be depleted and effect on such R-TDR should be evaluated to confirm that the repair of the DSB-containing plasmid indeed proceeded via such pathways.

6. Lines 129-130: "...while the exon-intron....with splicing.? Can the authors explain why it is so? Does splicing proteins play any role? Some discussion will be helpful to understand this section.

7. The authors have provided massive amount of high-quality sequence data and these data have been carefully analyzed; however, this reviewer can't comment on bioinformatic analyses of such large amount of data due to lack of expertise.

Major points of the response to Reviewers and/or modifications included in the revised manuscript:
We strengthened the conclusions of the study about the role of transcript RNA in NHEJ and MMEJ by adding new results and better emphasizing the significance of our findings. Specifically:

- i) To support the role of transcript RNA in RNA-mediated DSB repair by NHEJ and MMEJ, we have overexpressed human ribonuclease (RNase) H1 in the human cells. A new section in the Results was added titled ‘Overexpression of RNase H1 reduces the efficiency of RNA-mediated DSB repair’, and the new results are shown in **Figure 5**.
- ii) To support our findings in the human cells and extend these to chromosomal DNA, we have engineered chromosomal DNA of yeast *Saccharomyces cerevisiae* to contain constructs like those used in the assay in human cells, including Cas9 and sgRNAs. These results are presented in a new section titled ‘An antisense transcript RNA guides double-strand gap repair in its DNA by NHEJ in yeast cells’ and are shown in **Figure 6**. The experiments in yeast have been conducted by Dr. Chance Meers (now at Columbia University) and Yilin Lu in the Storici lab, who have been included among the authors of the manuscript.
- iii) Using the yeast chromosomal system we have introduced a mutant of ribonuclease (RNase) H1 and H2 and run the DSB repair assay. The results are supportive of the role of RNA in DSB repair. The results of this experiment are discussed in the manuscript and shown in **Extended Data Figure 17**.
- iv) We considered that because the R-TDR results do not represent the main findings of this study, we preferred to omit them from this manuscript to compensate for the addition of the 2 new sections in the Results presenting the experiments with the overexpression of RNase H1 and those of the yeast system, which are both supportive of our findings of RNA-mediated DSB repair via NHEJ and MMEJ.
- v) We have better explained the design of the plasmid constructs used in the experiments of RNA-mediated DSB repair in human cells. We underscored the value of these constructs. This is presented at the beginning of the Results.
- vi) We have emphasized the value of the variation-distance graphs to characterize the millions of sequences of DSB repair per sample in a single graph image. This is presented in the Discussion.
- vii) We have added more details about the data analyses in the Methods. We performed further analyses of the reads derived from the repair of 1-DSB that did not correspond to NHEJ in/del, Error-free NHEJ/uncut, and MMEJ. These previously unclassified reads were classified in subcategories presented in Supplementary Table 6. Furthermore, we have substantially modified the text of the manuscript to better explain the procedures used, better reflect our findings, and to avoid overinterpretations and misinterpretations. Starting from the abstract, all sections of the manuscript have been modified.

Overall, we clarified all the points of misinterpretation and rigorously addressed all the Referee criticisms and suggestions, as presented in the point-by-point-response shown below. As stated by Reviewer 3, “The authors have provided massive amount of high-quality sequence data and these data have been carefully analyzed”; also including the new experimental data, we believe that our study has been considerably strengthened and it provides results that significantly advance our knowledge about the role of RNA in DSB repair. We are grateful to the Reviewers for their comments/critiques that made us strengthen the findings of this study. In the new version of the manuscript, we hope that all the Reviewers will recognize the technical robustness of our approaches to study DSB repair by transcript RNA, the richness of novel

findings that we obtained, and the value that our study adds to the field of DNA repair and genome stability.

Point-by-point-response to each comment of the reviewers:

Reviewers' comments:

Reviewer #1 (Remarks to the Author):

1) In sum, the authors describe unusual phenomena where products of DSB by end joining pathways appear effected by the identity of transcribed RNA. For example, after cutting at one side of an intron, repair products consistent with inaccurate NHEJ at that DSB are slightly enriched by an unspliced transcript. Perhaps more compelling, after cutting at both sides of an intron, repair products consistent with NHEJ after deletion of the intron are more frequent when a version of the transcript with the intron spliced out is present. Effects of transcription are also increased in RNaseH2A deficient cells, suggesting they are mediated by a RNA-DNA hybrid.

- We thank the Reviewer for the positive comments.

2) The authors refer to transcript RNA playing a “significant role in genome stability”. This claim would need support by experiments addressing chromosomal repair.

- We thank the Reviewer for the comment. Our results show that transcript RNA can influence mechanisms of DSB repair in human cells on plasmid DNA, suggesting that RNA could play a role in genome stability. To support our findings in the human cells and extend these to chromosomal DNA, we have engineered chromosomal DNA of yeast *Saccharomyces cerevisiae* to contain constructs similar to those used in the assay in human cells: the Antisense and the Branch Δ constructs (Figure 6a, b). We have added a new section of Results on page 9 to 10 describing the experiments in yeast cells titled ‘An antisense transcript RNA guides double-strand gap repair in its DNA by NHEJ in yeast cells’. As described in this section of the Results, the findings obtained for double-strand gap repair in yeast chromosomal DNA were consistent with those obtained for double-strand gap repair in the plasmid constructs in human cells. The intron sequence was maintained more frequently in the presence of the transcript containing the intron following the DSBs. On the other hand, the spliced transcript enhanced double-strand gap repair by pop-out of the intron following the DSBs (Figure 6c). These results obtained on the yeast chromosomal DNA support the capacity of transcript RNA to influence genome stability.

3) Unfortunately, end joining is measured only by sequencing. There was no attempt to validate repair according to pathway by genetic means (or using an inhibitor).

- The high-throughput sequencing analyses provide a strong support to DSB repair by NHEJ and MMEJ because of the specific sequence signatures identified. While we agree that it will be important to target NHEJ factors, as well as MMEJ factors to further study DSB repair-mediated by RNA, we think it is the follow up from this study and is beyond the scope of the current work. Nonetheless, to support the role of RNA in mediating DSB repair by NHEJ and MMEJ, we performed a series of genetic experiments using the plasmid constructs in human cells in the condition of ribonuclease H1 (RNASEH1) overexpression and

engineering the splicing and non-splicing constructs in yeast chromosomal DNA. Specifically, a plasmid containing the human *RNASEH1*-coding sequence was co-transfected together with the Sense and Branch Δ constructs to study double-strand gap repair in the HEK-293T wild-type cells. A new section in the Results was added on page 7 and 8 entitled ‘Overexpression of RNase H1 reduces the efficiency of RNA-mediated DSB repair’. The results of this experiment showed that when RNase H1 is overexpressed the difference between the intron pop-out frequency for the splicing vs. that of the non-splicing construct was reduced compared to the control (overexpression of a mutant form of RNase H1) (Figure 5). Experiments in budding yeast using wild-type RNase H cells, in which the DSBs were induced on chromosomal DNA were in line with those obtained in the human cells, supporting a role of RNA in NHEJ and MMEJ. Moreover, we constructed the null mutants for RNase H1 (*rnh1*) and the catalytic subunit of RNase H2 (*rnh201*-null) in the yeast strains carrying the splicing and the non-splicing constructs in chromosomal DNA. In the *rnh1 rnh201*-null cells, the frequency of sequences without intron was significantly increased for the splicing construct (Extended Data Figure 17). All the experiments using yeast cells are presented on page 9 and 10 in a new section entitled ‘An antisense transcript RNA guides double-strand gap repair in its DNA by NHEJ in yeast cells.’ Overall, these new genetic experiments strengthen our findings that transcript RNA can direct the way DSBs are repaired in DNA.

4) Most importantly, they do not consider other outcomes that would require very different interpretation of the data, including still broken DNA ends, and especially uncut substrate and accurate end joining. Importantly, a control asserting equal cutting by Cas9 in vitro isn’t useful since the authors are studying effects of transcription of DSB target sites in cells. They must be able to exclude a possible influence on cutting by transcription or r-loops.

- Our experiments are focused on studying the sequences around the sites of DSB and on comparing the frequencies of sequences with intron retention vs. those with intron loss after induction of DSB(s) when there is splicing or not. The promoter that drives the transcription for the Sense construct is the same as the one driving the transcription for the Branch Δ construct. The Sense and Branch Δ constructs have identical sequence except for 55-bp difference on the intron, which does not affect the sgRNA targeting sites. The pCMV Δ construct has exactly the same sequence of the exons and intron with the Sense. It is not possible to obtain an accurate measure of DSB efficiency when transcription is active because the transcript RNA is involved in the repair of the DSB. Instead, the in-vitro cleavage results provided support to the fact that the 55-bp difference between the splicing and non-splicing constructs (due to the deletion of the branch site in the non-splicing construct) does not impact the cleavage efficiency. For the experiments to compare DSB repair sequences between the splicing and non-splicing constructs, we used exactly the same cell culture, the same Cas9 plasmid, the same sgRNAs to induce the DSBs in the same position on the three constructs (Sense, Branch Δ and pCMV Δ), and in the same conditions. We do consider uncut substrate and accurate end joining. These results are shown in Extended Data Figure 6. These results show that the most prevalent mechanism of DSB repair in all constructs is NHEJ. We cannot distinguish uncut from perfect repair w/o in/dels. Moreover, it is likely that perfect repair is subject to more cleavage. The construct with no splicing has higher frequency of NHEJ events than the constructs with splicing. We did of course think about alternative explanations for the DSB repair frequencies that we observed, but we could not find any alternative explanations that would match our results. In the 1-DSB system, the data analyses of MMEJ exon-exon vs. exon-intron clearly show prevalent MMEJ exon-exon for the splicing constructs vs. prevalent MMEJ exon-intron for the non-splicing construct. These results cannot not be explained by more efficient cleavage of sgRNA A or B in the splicing or in the non-splicing constructs. Although not all reads in the 1-DSB experiments were classified as error-free NHEJ/uncut, NHEJ with in/dels, or MMEJ, a thorough analysis was done of the previously unclassified reads (now classified in subcategories presented in Supplementary Table 6), which revealed that they were either likely sequencing errors (i.e., such reads appeared in No-DSB control experiments as well) or followed the same pattern as the classified reads (reads

with higher complementarity to the spliced transcript appears more often in the constructs with splicing while reads with higher complementarity to the non-spliced transcript appeared more often in the construct without splicing). See Categorizing 1-DSB sequencing reads into three groups: NHEJ in/dels, Error-free NHEJ/uncut, and MMEJ in the Methods for more details.

In the 2-DSB system (double-strand gap repair experiments), if for example the frequency of DSB would be higher for the splicing constructs, not only would the frequency of double-strand gap repair by NHEJ and MMEJ be higher in these constructs compared to the non-splicing construct, but also that of inverted intron capture, which was instead higher for the non-splicing construct. The experiments using the antisense constructs have results in line with those obtained for the sense constructs despite using different sgRNAs. The yeast constructs also share the same Cas9, and sgRNAs, among each other, and differ by the deletion of the branch site (33 bp), which also does not affect the binding sites of the sgRNAs C and D. *In vitro*-cleavage data are also provided for the yeast constructs (Extended Data Figure 15). The results obtained using the yeast chromosomal system are also in line with those obtained in human cells. In addition, the experiments with overexpression of RNase H1 in human cells and those in yeast with RNase H1 and H2-defective mutants provide further support to the role of RNA in mediating DSB repair via NHEJ and MMEJ. In response of this point of the Reviewer, we have worked throughout the manuscript to increase clarity of our experiments, emphasize the control level of our systems, the consistent results despite using different sgRNAs, the reproducibility of the results even between human and yeast data, and the support provided by the new experiment with RNase H1 overexpression and the KO of *rnh1* and *rnh201* in yeast cells.

5) The latter issue is particularly notable, given the following statement of the authors: “The frequencies of error-free NHEJ/uncut constructs did not fully compensate for such difference in the NHEJ in/del frequency between the non-splicing and the splicing constructs (Extended Data Figure 6a, b).” This seems like gaslighting, since the data show otherwise. For extended 6a, NHEJ in/del is increased 1% in Branchdel (non-splicing, 4% to 5%), while uncut constructs are reduced by about 1% (94.9% to 94%) in the same sample.

- We thank the Reviewer for pointing this out. The sentence was confusing. It meant that in addition to NHEJ we anticipated another mechanism of DSB repair, MMEJ, specifically exon-exon MMEJ, that is compensating for the NHEJ difference between the splicing and the non-splicing constructs. We modified the text in two points, on page 3, lines 96-98 as follows: <<While it was not possible to distinguish the sequence of error-free repair by NHEJ from that of the uncut constructs, we searched for in/dels near the DSB site as the signature for NHEJ (see Methods). The in/del signature was practically absent in the No-DSB controls, in which galactose was not added to activate Cas9 expression (Extended Data Figure 5a, b). As expected, the most prevalent mechanism of DSB repair in all constructs was NHEJ (Extended Data Figure 6a, b). The analysis of the sequencing data revealed that the construct without splicing, Branch Δ , had higher frequency of NHEJ in/dels than the constructs with splicing (Sense and pCMV Δ) in wild-type and more evidently in the RNase H2A KO cells (Figure 2a and Supplementary Table 3). >>, and on page 5, line 134-136: <<This higher exon-exon MMEJ frequency in the splicing constructs seems to compensate for the higher frequency of NHEJ in the non-splicing construct (Figure 3a and Extended Data Figure 6)>>.

6) The final paragraph of the results appears non-sensical. It is led by the phrase “... we search for the R-TDR signature...”. What exactly is this signature? In methods, the authors indicate they identify R-TDR according to this sentence: “We consider RNA-templated DNA DSB repair to be the mechanism that produces an identical sequence with the template RNA (except for Ts in place of Us).” This seems to imply the DNA repair product is made from an “RNA template”. Since the RNA template is made from the uncut original DNA template, how can the resulting DNA product sequence be distinguished from uncut DNA sequence?

- The R-TDR, which means RNA-templated DNA repair (see Meers et al., Mol Cell 2020 and Keskin et al., Nature 2014), signature corresponds to the pop-out of the intron in the DNA exactly as it is in the RNA after splicing of the intron. Therefore, the resulting DNA product sequence can be distinguished from uncut DNA sequence because it lost the intron in the splicing constructs. The non-splicing construct functions as negative control because it cannot produce the perfect sequence w/o the intron via R-TDR, as first described in Keskin et al., Nature 2014. In this new version of the manuscript, we considered that the R-TDR results do not represent the main findings of this study, and we preferred to remove them from this study to leave more space for the experiments on the overexpression of RNase H1 and those of the yeast system, which are both supportive of our findings of RNA-mediated DSB repair via NHEJ and MMEJ.

Minor concerns.

7) The authors employ a method of data representation (“individual variation-distance graphs”) that seems needlessly complicated, and confound this problem by employing jargon-heavy descriptions. Worse, these graphs serve little obvious purpose, since individual “DSB-sequence windows” are either broadly enriched or broadly depleted by a given experimental variable. The bar graphs are sufficient. The sole exception (a few rarely used DSB sequence windows in the Fig 2C RNaseH2A KO) isn’t interpreted.

- We thank the Reviewer for the critique. While we agree that the bar graph provide a summary of the results, the individual variation-distance graphs display the complete results of millions of DSB repair events in a single image. Thus, the variation-distance graphs provide a lot of information that cannot be shown with the bar graphs. We believe that it is important to note that the splicing and the non-splicing constructs repair the DSB/s with the same mechanisms, but transcript RNA influences the frequency of these mechanisms. Therefore, the variation-distance graphs have a lot of value and also provide a novel approach to characterize sites of DSB repair. To better emphasize the relevance of the variation-distance graphs, we have expanded the section of the Discussion about the significance of the variation-distance graphs from page 11 line 365 to page 13, line 402: <<The variation-distance graphs used to analyze the sequencing reads derived from R-NHEJ gave a detailed snapshot of the DSB-repair sequence variation types (insertions and deletions), nucleotide compositions, and their frequencies, displaying the complete results of millions of DSB repair events in a single image. In the 1-DSB system, we found that graphs arising from different guide RNAs had marked differences, while those arising from the same guide RNA were similar to each other. In graphs corresponding to sgRNA A, inserted nucleotides had a preference to start with T, while in those of sgRNA B, inserted nucleotides had no preference among the four bases (Figure 2b, Extended Data Figure 8a, c). Such differences in the in/del type of NHEJ repair at these two DSB sites supports marked dependence of Cas9 cleavage on the sequence context, as previously reported¹⁹. Among the insertions starting with nucleotide T, the mononucleotide T and the dinucleotides TA, TG, and TT had high frequency for NHEJ repair after DSB by sgRNA A, beyond been more abundant for the Branch Δ construct (Figure 2b,c and Extended Data Figures 7 and 8). These T-initiated insertions are most likely due to 5'-sticky end cleavage of Cas9 on the PAM strand followed by templated¹⁹ (T and TG) or partially templated (TA and TT) insertions. Therefore, it is possible that the biased T-initiated insertions for repair of the DSB caused by sgRNA A reflect a frequent 5'-sticky end cleavage by Cas9, and the non-biased insertions following DSB by sgRNA B reflect a most prevalent blunt-end cleavage by Cas9. Future experiments could be conducted to understand whether transcript RNA can better support repair of blunt vs. sticky-end DSBs.

Interestingly, in all constructs, the generation of a double-strand gap in the 2-DSB system produced greater variety of in/dels (i.e., different vertices in the graph) after repair by NHEJ than in the 1-DSB systems, particularly for deletions (compare graphs in Figure 4c and Extended Data Figure 10a, c with those in Figure 2b and Extended Data Figure 8a, c). Moreover, although the variation of the in/dels among the different constructs is similar within the 1-DSB and the 2-DSB systems, the relative frequencies of the corresponding in/dels were consistently higher in one construct than in the other as the comparison graphs

indicate. The variation-distance graphs also reveal other details that cannot be seen with the bar graphs. For example, when the DSB is caused by sgRNA B, there are more chances that the transcript RNAs of the splicing constructs, especially for the pCMV Δ construct (because there are less transcripts), have already lost the intron. The variation-distance graph comparing the NHEJ repair after DSB by sgRNA B for the Sense vs. the pCMV Δ construct shown in Extended Data Figure 8b highlights a more efficient NHEJ repair for all individual in/dels of the Sense construct. The difference between the NHEJ repair of the Sense vs. the pCMV Δ construct is enhanced in the RNase H2A KO cells (Extended Data Figure 8d). It is likely that the transcript RNAs of the pCMV Δ compared to those of the Sense construct have already spliced out the intron when the DSB occurs at the Intron-Exon2 junction (DSB by sgRNA B); thus limiting the capacity of the transcript RNA of the pCMV Δ construct to repair the DSB by NHEJ. Instead, for a DSB by sgRNA A the transcripts from the splicing constructs may still carry the intron, and a low level of transcription from the pCMV Δ construct may facilitate the interaction between the RNA that still carries the intron and the DSB ends resulting in higher frequency of NHEJ repair for most of the in/dels of the pCMV Δ construct (Extended Data Figure 8b). In this case, the difference between the NHEJ repair of the Sense vs. the pCMV Δ construct is reduced in the RNase H2A KO cells (Extended Data Figure 8d)>>.

The Reviewer correctly identifies that there are sequences in our variation-distance graphs that do not follow the general pattern of most sequences having higher frequency in Sense and Antisense (for 2-DSB experiments) or Branch Δ (for 1-DSB experiments). We have added a description of this phenomenon in the text and indicated that a careful examination of this will be required to determine possible biological interpretations but is currently outside the scope of this article: page 13, lines 402-409: <<Most of the variation-distance graphs show that when comparing the NHEJ frequencies between Sense and Branch Δ , Sense and pCMV Δ , or Antisense and 5'-Splicing Δ , the relationship between the overall NHEJ frequencies of the two constructs is reflected in most of the individual sequence frequencies as well. Some graphs show sequences that are exceptions to this rule, such as in Extended Figures 8b and 8d (sgRNA B, Sense vs. Branch Δ), and 10f (Antisense vs. 5'-Splicing Δ) where we see some sequences that have higher frequency in the construct that has overall lower frequency. Further study will be required to determine why such sequences are enriched differently >>.

8) The authors do not complement RNaseH2A deficiency to validate this result.

- The results obtained using the RNase H2A KO cells are in line with those obtained in the wild-type RNase H cells, and even stronger than these in some instances. While RNase H1 is expressed throughout the cell cycle, RNase H2A expression levels change during the cell cycle (Parajuli, S. et al., 2017, doi:10.1074/jbc.M117.787473 and Lockhart, A. et al., 2019, doi:10.1016/j.celrep.2019.10.108). Moreover, overexpression of the single subunits of RNase H2, like RNase H2A, could be toxic for the cells. Therefore, we preferred not to stress the RNase H2A KO cells with transfection of an overexpression plasmid to complement RNaseH2A deficiency. Instead of complementing RNaseH2A deficiency, we overexpressed human RNaseH1 in the wild-type RNase H cells, because both RNase H1 and H2 target RNA-DNA hybrids in cells. We have conducted new experiments with the overexpression of RNase H1, and these are described in a new section of the Results on pages 7-8 and entitled 'Overexpression of RNase H1 reduces the efficiency of RNA-mediated DSB repair'. These results support the role of RNA in mediating DSB repair in human cells.

9) The methods are generally oddly chosen. For example, why assess products 6 days later, since repair should be finished far earlier?

- The incubation time (6 days) was chosen to give time to Cas9 to break again the constructs if there were perfect repair events, to enrich for the repaired sequences. All our experiments were performed in the same

exact conditions for all the constructs studied, all samples were treated in the same manner. Always the splicing (Sense and pCMV Δ) and the non-splicing (Branch Δ) constructs were transfected in the same experiment, side by side using the same cell culture, with the same Cas9 and sgRNA plasmids in the same exact conditions, and the plasmid DNA was extracted from all these samples at the same time and using the same procedure. We have edited the text on page 2 from line 65 as follows: <<In all three constructs, we induced a DSB in the *DsRed* gene on either side of the intron, or on both sides using two single guide RNAs (sgRNAs) with Cas9 endonuclease using the same sgRNAs cutting in the same sites on these three constructs. >>, on page 3, lines 75-78: <<Individually, the plasmids with the Sense, Branch Δ , and pCMV Δ constructs were transfected four independent times each into cells of the same culture of HEK-293T wild-type cells, as well as HEK-293T knock-out cells having mutations in the catalytic subunit of RNase H2 (RNase H2A KO) >>, and on page 3, lines 85-86: <<After a few days, the plasmid DNAs of the three constructs were extracted from the cells at the same time and prepared for next generation sequencing (NGS)...>>.

Reviewer #2 (Remarks to the Author):

In this manuscript, Jeon et al used a DsRed-containing plasmid system, in combination with targeted amplicon deep sequencing of the repair junctions, to study Cas9-induced double strand break repair in HEK293T cells and test the hypothesis that an RNA transcript could mediate DSB repair in human cells. On the basis of the data, the authors stated that “both coding and non-coding transcript RNA facilitates DSB repair in a sequence-specific manner in human cells” in the abstract and concluded that RNA transcripts can promote NHEJ or MMEJ of Cas9-induced DSBs or even serve as a template for NHEJ in human cells. Although a large amount of data are provided with some particular design of the plasmid system in an attempt to support the authors’ claims in this manuscript, the approaches taken and evidence provided are too weak and not convincing. It provided no direct answers into the question whether or not RNA-mediated DSB repair really occurs. I will explain these major issues as well as some minor concerns in details below.

- We thank the Reviewer for the comments. We have worked to better emphasize the results of this study, the value of the designs used, the controls used, the reproducibility of the results, and the significance of the study. We have also performed additional experiments in human cells, and we have added experiments using the budding yeast cells that altogether provide new support to RNA-mediated DSB repair. All the issues and concerns of this Reviewer have been addressed below.

1. The #1 issue of this manuscript is that this manuscript is completely based upon the plasmid assays. There are several problems with this approach. Firstly, many copies of the reporter plasmids (i.e., the Sense, Branch Δ and pCMV Δ constructs) were delivered into the cells by transient transfection. In addition to Cas9-induced DSBs, the plasmids often harbor spontaneous nicks and DSBs and even more after being delivered into the cells. Due to spontaneous nicks and DSBs along with Cas9-induced DSBs, recombination between different copies of these plasmids could complicate the analysis and outcomes of the RNA-mediated DSB repair assays, potentially producing misleading results. Second, the plasmids the authors chose to use contains the SV40 origin and the cells used also express SV40 large T antigen. These plasmids transfected would replicate in HEK293T cells. Replication of these plasmids can cause additional DSBs (one-ended and two-ended DSBs) and nicks, complicating the outcomes of the plasmid assays further. It is not clear why the authors added DNA replication into this plasmid approach. In addition, if RNA-mediated DSB repair occurs in human cells, it should occur in the chromatin context. However, the plasmids transiently transfected into 293T cells do not form chromatin structure resembling that of the human genome. Any

meaningful observation on RNA-mediated DSB repair requires further confirmation in the context of chromatin. Because of these problems, there is a possibility that RNA-mediated DSB repair observed in this manuscript is an experimental artifact and does not really take place in the genome. That is not to mention the over-interpretation of the data by the authors throughout the manuscript.

- The new version of our manuscript presents data supporting RNA-mediated DSB repair that are not completely based upon the plasmid assay because we included chromosomal assays in budding yeast cells. First, we note that we chose to work with the constructs on plasmids for this study in human cells because we could engineer the desired modifications on the plasmids in a much more proficient manner than if these constructs were on human chromosomal DNA. Moreover, the plasmid constructs represent a stable system because the constructs on plasmid DNA are less prone to rearrangements and/or mutations than if they would be integrated in the genome of the HEK-293T cells. If the constructs were engineered on chromosomal DNA, the cell line containing the Sense would be different from the one containing the Branch Δ , and the pCMV Δ . There could be variations and mutations not just at the locus of the constructs but elsewhere in the genome of the cells that could affect the DSB-repair mechanisms. Instead, when we transfect the different plasmid constructs into the HEK-293T cells, we use the same cell culture for all the constructs; therefore, the system is more uniform and the differences among the constructs that we detect are dependent on the different constructs not on the cell lines used. To better describe the plasmid system, we modified the text on page 2, line 55-58: <<The constructs were put on plasmid DNA to facilitate the engineering and maintain a controlled (less affected by genome mutations and/or rearrangements) and exportable (allowing to introduce and study the constructs in cells of different genotypes) system.>>, page 2 line 65 to page 3, line69: <<In all three constructs, we induced a DSB in the *DsRed* gene on either side of the intron, or on both sides using two single guide RNAs (sgRNAs) with Cas9 endonuclease using the same sgRNAs cutting in the same sites on these three constructs. In all constructs, the sgRNA A binds near the junction of the 5' exon, Exon1, and the intron, and the sgRNA B binds near the junction of the intron and the 3' exon, Exon2 (Figure 1a).>>, on page 3, from line 75: <<Individually, the plasmids with the Sense, Branch Δ , and pCMV Δ constructs were transfected four independent times each into cells of the same culture of HEK-293T wild-type cells, as well as HEK-293T knock-out cells... >>, and on page 3, from line 85: << ... the plasmid DNAs of the three constructs were extracted from the cells at the same time and prepared for next generation sequencing (NGS) to study the sequence around the DSB site or the double-strand gap ...>>.

As the Reviewer indicates, many copies of the plasmids are delivered to the cells. However, in the experiments we do make multiple repeats (4) of the transfection for each construct. The transfected cells are from the same culture for the different constructs in every experiment, and are transfected following the exact same protocol and the same amount of the plasmid constructs. The Cas9 and the sgRNA plasmids are always from the same batch in each experiment for the different constructs. The Sense, Branch Δ , and the pCMV Δ are very similar to each other: the Branch Δ has a 55-bp deletion containing the branch site, and the pCMV Δ does not have the CMV promoter. Therefore, it is expected that spontaneous nicking would affect the different constructs in the same manner. The results are reproducible. In fact, we obtained similar results for the Sense, Branch Δ , and the pCMV Δ constructs in the wild-type and in the RNase H2A KO cells. It is our goal also to develop an approach to study RNA-mediated DSB repair at the chromosomal level in the human cells; however, this is beyond the scope of the current study. To better clarify this point, the text was edited on page 2, from line 55: <<The constructs were put on plasmid DNA to facilitate the engineering and maintain a controlled (less affected by genome mutations and/or rearrangements) and exportable (allowing to introduce and study the constructs in cells of different genotypes) system >>, as well as other points as described as presented just above.

All the splicing and non-splicing constructs contain the SV40 origin and were treated in the same manner; therefore, we do not expect any bias on the interpretation of our result. We chose to work with HEK-293T cells and plasmids with the SV40 origin to have DNA replication as on chromosomal DNA. We chose to work with the replicating system. For future work we had considered transfecting the same

plasmids in the HEK-293 cells that are not expressing the T antigen. These experiments would involve DSBs in a non-replicating system; thus, a different situation from what occurs in the genome of replicating cells. Studying DSB repair by RNA in a non-replicating system is in our plan for future studies. Again, the plasmid constructs represent an advantage vs. the integrated constructs because they can be introduced into different cell lines (e.g., HEK-293T and HEK-293). In this study, we use controls in which the DSBs were not induced, and we use another control which is a sequence 30 bp downstream from the DSB site of each construct. Our control results showed negligible in/dels among the different constructs (see Extended Data Figure 5 and 16). Therefore, spontaneous nicks or DSBs would be too infrequent to have an impact on the results. As indicated above, the Sense, Branch Δ , and the pCMV Δ are very similar to each other: the Branch Δ has a 55-bp deletion containing the branch site, and the pCMV Δ does not have the CMV promoter. It is expected that spontaneous nicking, if occurring, would affect the different constructs in the same manner. In the experiments we do not see major variation among the same type of constructs and the results are also reproducible across different cell lines. In fact, we obtained similar results for the Sense, Branch Δ , and the pCMV Δ constructs in the HEK-293T wild-type and in the HEK-293T RNase H2A KO cell lines. To better clarify this point, the text was edited as already discussed in the paragraphs above and on page 3, from line 84: <<In addition, as discussed below, a sequence 30 bp distant from the DSB site was also used for no-DSB control>> and on page 3, from line 98: <<The in/del signature was practically absent in the no-DSB controls, as well as in the control sequences 30 bp downstream from the DSB sites (Extended Data Figure 5a, b)>>.

While several reports have shown that plasmids rapidly form chromatin upon transfection in mammalian cells becoming like minichromosomes (Reeves et al NAR 1985; Bai et al. Bioscience Reports, 2017 with references therein), we do not know at which point chromatin is formed on the plasmid constructs used in this study. Nonetheless, we believe that the results on the plasmid constructs in human cells represent a significant step forward towards understanding the capacity of RNA to play a direct role in DNA DSB repair. As also indicated above, the use of the plasmid system provides an additional value, that is the ability to transfect the constructs into HEK-293T or HEK-293 cells of different genotypes, e.g., RNase H2A KO cells and in future to cell lines with defects in NHEJ or MMEJ genes, allowing broader characterization of the mechanisms by which RNA can mediated DSB repair in the human cells. For better clarity, we have included a statement in the Discussion on page 11, lines 343-346: <<The findings obtained from the plasmid constructs in human cells were supported by the results produced using the yeast chromosomal constructs, possibly indicating a conserved capacity of transcript RNA to play a direct role in DSB-repair mechanisms in eukaryotic cells>>.

While integrating the current plasmid constructs into human cells is beyond the scope of this work, we integrated analogous constructs in chromosomal DNA of budding yeast cells. We engineered the genome of yeast *Saccharomyces cerevisiae* with constructs carrying the *his3* gene with an intron, like the splicing construct and the non-splicing constructs used in human cells. We genetically engineered an Antisense and a Branch Δ constructs in yeast RNase H wild-type cells. We also integrated two guide RNAs and the Cas9 gene under the inducible galactose promoter to generate DSBs on each side of the intron within the *his3* gene. With such yeast chromosomal constructs, we induced DSBs and then studied the DNA repaired region around the sites of breakage. The results of the yeast experiments, which are described in the last section of the Results on pages 9-10, entitled ‘An antisense transcript RNA guides double-strand gap repair in its DNA by NHEJ in yeast cells’, are consistent with those obtained using the plasmid constructs in the human cells.

We provide here above (and below) several clarifications and explanations, as well as new results that are supportive and consistent with a role of RNA in mediating DNA DSB repair in human and yeast cells.

It was not our intention to overinterpret the data. In the new version of the manuscript, the findings were strengthened by new results, and we have substantially modified the text to better reflect our findings and to avoid overinterpretations and misinterpretations. Starting from the abstract, all sections of the manuscript have been modified.

2. The weakness in the plasmid approach can be easily addressed with the DsRed reporters integrated in a single copy into specific sites of the genome in mammalian cells. In this setting where a DSB can be induced at a specific genome site, coding and non-coding RNA (antisense RNA) can be provided either to mediate DSB repair or as a template for DSB repair. Any alterations on RNA transcripts could be introduced to determine whether the pairing between DNA and complementary RNA really facilitates DSB repair or even complementary RNA directly serves as the template for DSB repair. However, the manuscript has not done this type of experiments. Furthermore, the fluorescence of DsRed in the reporter plasmids was neither used nor essential for the assays in this manuscript. The authors instead used NGS to analyze the efficiency of RNA-mediated DSB repair and RNA-templated repair. Therefore, a couple of endogenous genomic sites in the cells could be a better choice than the DsRed plasmids in this manuscript for similar analysis. Surely, the results from these natural sites could be more convincing in either supporting or rejecting the authors' conclusions.

- As indicated above in response to point 1) for this Reviewer, the plasmid system allowed a series of genetic engineering steps to study DSB repair by RNA. For example, we first inserted the intron, then we deleted the branch site; this was not trivial, as we had to test different sizes for the branch-site deletion. The fact that we used the *DsRed* gene helped us to determine whether there was or not splicing of the intron despite the deletion of the branch site. Red fluorescent cells still had a functional branch site, while no red fluorescent cells lost the functionality of the branch site. Thus, the choice of the *DsRed* was very helpful for the design even if we did not measure the frequency of the red fluorescent cell for DSB repair because we performed next generation sequencing of the region around the site of DSB. Moreover, we had to edit the PAM site near the intron sequence. It took a lot of designing and engineering to develop and optimize the constructs we have presented in this study. To have the antisense (non-coding) RNA, we introduced the E1alpha promoter. The plasmid system is a good starting point allowing relatively easy engineering and a lot of optimizations. All this work would have been too extensive to be performed directly on human chromosomal DNA. In addition, as indicated above, the plasmid constructs represent a highly controlled system that has allowed us to detect even small differences in DSB repair in the different splicing and non-splicing constructs. Not to mention that the plasmid constructs can easily be introduced in different cell types, like wild-type HEK-293T and RNase H2A KO, and can allow us to study DSB repair by RNA even in condition of no DNA replication if the plasmids are transfected in HEK-293 cells that do not have the T antigen for the replication of the plasmid. Now that we have established the constructs to study DSB repair by RNA in human cells, and we have also obtained consistent results from the chromosomal constructs in yeast genomic DNA, yes, we could integrate the splicing and non-splicing constructs in the genome, and we may. However, this would be a separate work that is beyond the scope of the current study. We also agree with the Reviewer that targeting endogenous sites in human chromosomal DNA would be valuable. However, it is hard to pick specific endogenous sites on chromosomal DNA having the same expression level and same sequence context but with and without splicing like we have in the plasmid constructs. Again, having now established the capacity of RNA to mediate DSB repair in human cells via different DSB-repair mechanisms (e.g., NHEJ and MMEJ) using the plasmid constructs, and having the supporting results for RNA-mediated DSB repair on the chromosomal constructs in yeast DNA, we are certainly more knowledgeable for how to design a new project in which we can target endogenous sites in the human genome, via a new study, which is beyond the scope of the current work. To better clarify this Reviewer's point, as indicated in point 1) above, we have better emphasized the value of the plasmid constructs, and we have added new data using the chromosomal constructs in the yeast cells as described above.

3. The authors over-interpreted the data throughout the manuscript from Figure 2-Figure 5, **a)** on page 4 lines 110-113, **b)** on page 5 lines 145-146 and lines 157-158, **c)** on page 6 lines 178-181 and **d)** on page 7 lines 221-222. There is no direct evidence to support these statements. **a)** For instance, on page 4 lines 110-

113, the authors stated “Overall, these results provide evidence that the sequence of a transcript RNA, its complementarity with the DSB ends, and the level of transcription directly influence the efficiency of DSB repair by NHEJ and the frequency of the repaired products by NHEJ.” However, the data only show that these three different constructs (i.e., the Sense, Branch Δ and pCMV Δ construct) exhibit a difference in the efficiency of Cas9-induced NHEJ. This difference could be caused by the variations in target DNA sequences, DNA metabolism at the target, and the target binding, cleavage and post-cleavage dissociation of the Cas9-sgRNA complex. Recent studies have shown that transcription and DNA replication affect the target binding, cleavage and post-cleavage dissociation of the Cas9-sgRNA complex, thus adding a layer of control in DSB repair pathway choices and influencing the repair outcomes (Clarke et al, Mol Cell 71:42–55.e8, 2018; Ivanov et al, PNAS 117:5853–5860, 2020; Liu et al, Genome Biol 23:165, 2022). Thus, the results of DSB repair in the plasmids may have nothing to do with the RNA transcripts, but be influenced by these above possibilities that this manuscript did not exclude. **b)** The statement “The results were stronger in the RNase H2A KO cells, suggesting direct RNA-DNA interaction (Figure 4a and Supplementary Table 3)” on page 5 lines 157-158 was also over-interpreted. Direct evidence is needed for the enrichment of direct RNA-DNA interaction. **c)** Another example of over-interpretation is the statement on page 6 lines 178-181. The slightly higher frequency of NHEJ in/dels in double-strand gap repair in the Sense construct only reveals the association with the Sense construct. Whether or not spliced RNA is used for double-strand gap repair has not been determined in this manuscript. **e)** In summary, in order to support the statements on RNA-mediated NHEJ, RNA-mediated MMEJ, RNA-mediated double-strand gap repair and RNA-templated repair in the manuscript, the authors have to provide more direct and stronger evidence.

- We respectfully disagree with the interpretation of this Reviewer. As discussed above, thanks to the suggestions of all Reviewers, we have introduced new experimental data in the study that have strengthened our conclusions that RNA does play a significant role in DSB-repair mechanisms. We have omitted the results on the RNA-templated DSB repair because these were not in the center of the study. We have instead provided new supportive results about the effect of RNA on NHEJ and MMEJ, which represent the major findings of this study. Two new sections of the Results have been added and the corresponding results are presented in Figures 5 and 6, and Extended Data Figure 17. The experiment employing overexpression of human ribonuclease H1 (RNase H1) in the human cells shows that when the level of RNase H1 is increased the impact of transcript RNA on DSB repair is reduced. The experiments using the yeast chromosomal system are in line with those obtained from the plasmid systems in human cells. The RNase H1 and H2-null mutants tested in the yeast system lead to an opposite result to that obtained with RNase H1 overexpression, causing increased impact of the transcript RNA on DSB repair. Moreover, we have substantially modified the text throughout the manuscript to avoid any wording that could reflect overinterpretation and/or misinterpretation of our results. For better clarity, we have broken up the critique of the Reviewer point 3) and the corresponding response in 5 subpoints from a) to e):

a) Regarding the statement on page 4 lines 110-113 of the previous version of the manuscript, the data show that the Sense, Branch Δ , and pCMV Δ constructs exhibit a difference in the efficiency of Cas9-induced NHEJ. The Sense and Branch Δ constructs have identical sequence except for 55-bp difference on the intron, which is outside of the sgRNA targeting sites; and the pCMV Δ has the same sequence of the exons and intron with the Sense. The same Cas9 and sgRNA plasmids are used to generate a DSB/s in the Sense, Branch Δ , and pCMV Δ constructs, in the same transfection experiments using the same cell culture and the same conditions and procedure. The RNA-seq data did not reveal significant difference in the expression level of the DsRed RNA between the Sense and the Branch Δ constructs. Moreover, it is not possible to obtain an accurate measure of DSB efficiency when transcription is active because the transcript RNA is involved in the repair of the DSB. Instead, the in-vitro cleavage results provide support to the fact that the 55-bp difference between the splicing and non-splicing constructs (due to the deletion of the branch site in the non-splicing construct) does not impact the cleavage efficiency.

DNA replication of the plasmids is expected to affect the constructs in the same manner because the plasmids are very similar to each other. The Sense and the Branch Δ only differ for the deletion of the

branch site (55bp), the CMV promoter driving the expression of the transcript from these constructs is the same. The sgRNAs with Cas9 recognize the same sites on the Sense, Branch Δ and pCMV Δ constructs. Therefore, we believe that it is unlikely that DNA replication differently affects the cleavage by Cas9 in the splicing vs. the non-splicing constructs.

In response to this subpoint, the specific statement in question has been modified to <<These results provide evidence that the presence or absence of the intron region in the transcript RNA, which affects the transcript's complementarity to the DSB ends, can directly influence the frequency of the repaired products by NHEJ>> on page 4 lines 114 to 116. Moreover, as also stated in the response to point 1) for this Reviewer, we modified the text on page 2, line 55 : <<The constructs were put on plasmid DNA to facilitate the engineering and maintain a controlled (less affected by genome mutations and/or rearrangements) and exportable (allowing to introduce and study the constructs in cells of different genotypes) system.>>, page 2, line 65: <<In all three constructs, we induced a DSB in the *DsRed* gene on either side of the intron, or on both sides using two single guide RNAs (sgRNAs) with Cas9 endonuclease using the same sgRNAs cutting in the same sites on these three constructs. In all constructs, the sgRNA A binds near the junction of the 5' exon, Exon1, and the intron, and the sgRNA B binds near the junction of the intron and the 3' exon, Exon2 (Figure 1a).>>, on page 3, lines 75-77: <<Individually, the plasmids with the Sense, Branch Δ , and pCMV Δ constructs were transfected four independent times each into cells of the same culture of HEK-293T wild-type cells, as well as HEK-293T knock-out cells... >>, and on page 3, lines 85-87: << ... the plasmid DNAs of the three constructs were extracted from the cells at the same time and prepared for next generation sequencing (NGS) to study the sequence around the DSB site or the double-strand gap ...>>.

b) The sentences on page 5 lines 145-146 and lines 157-158 were changed to <<Altogether, these findings support the capacity of RNA to promote DSB repair via MMEJ in a sequence-specific manner>> and <<We found that the constructs with splicing (Sense and pCMV Δ) had a higher frequency of intron pop-out than the Branch Δ construct (percentages in bold in Figure 4a). The individual frequencies of intron pop-out by NHEJ were also higher for the splicing constructs compared to the Branch Δ construct (Figure 4a). The results were stronger in the RNase H2A KO cells, suggesting direct RNA-DNA interaction (Figure 4a and Supplementary Table 3). >> on page 5, lines 159 to 160 and page 6, lines 170 to 174, respectively. Supplementary Table 3 shows the statistical analysis of these data.

We disagree that evidence of RNA-DNA interaction will help to understand the role of RNA in DSB repair. We know that the RNA is transcribed because we have the RNA-seq data showing it. Revealing RNA-DNA hybrids would not inform about what is the role of the RNA in DSB repair. The RNA-DNA hybrids may also be transient but still play an important role in DSB repair. Instead, to strengthen the role of the RNA in DSB repair, we have introduced two new sections in our study, one showing results following overexpression of RNase H1 and another showing results in yeast chromosomal DNA that include the use of RNase H1 and H2-null mutants. Both these experiments support the role of RNA interacting with the DNA at the DSB sites. These results are presented in the new Figures 5 and 6 and Extended Data Figure 17, and are discussed in the text.

c) The sentence on page 7, lines 203-206 was not changed because it reflects what we observed and what we propose: <<These results, for the repair of the double-strand gap generated by two DSBs near the intron-exon junctions, point towards a role of the spliced RNA in promoting intron deletion, and, vice versa, a role of the non-spliced RNA in maintaining the intron in its original locus or in facilitating its flipping>>. To emphasize this point, we modified the text on page 9 from line 274 to note that similar results were also obtained using the antisense RNA constructs that are presented in Figure 4a,c,d and e: <<Overall, the results obtained using the antisense constructs suggest that not just sense, but also antisense RNA has the capacity to promote double-strand gap repair by NHEJ or MMEJ in a sequence-dependent manner.>>. In addition, also the experiments in yeast supported a role of transcript RNA in DSB repair: <<Overall, these results suggest a role of the spliced RNA in promoting intron deletion via NHEJ and possibly MMEJ, and, vice versa, a role of the non-spliced RNA in maintaining the intron in its original locus also in yeast cells and

on chromosomal DNA>>, page 10 lines 304-307. The use of the yeast *rnh1 rnh201*-null cells, further corroborates a direct role for transcript RNA in modulating DSB-repair mechanisms.

The results presented in Figure 4a for the sense constructs do not reveal a “slightly higher frequency of NHEJ in/dels in double-strand gap repair in the Sense construct” as the Reviewer states. The overall frequency of intron pop-out for the Sense construct is 51.0% ($\pm 2.44\%$) and that one for the Branch Δ is 26.6% ($\pm 0.79\%$), with pop-out by NHEJ frequency of 46.9% ($\pm 2.37\%$) for the Sense and 25.6% ($\pm 0.84\%$) for the Branch Δ . This is a significant and large difference.

The results show that the non-splicing construct (Branch Δ) has significantly higher frequency of intron retention (74.3% $\pm 0.79\%$) compared to the Sense construct (49.1% $\pm 2.44\%$). This is also a significant and large difference. This is what we observe from the results. The data analysis for the RNase H1 overexpression experiment also reveal a significant difference between the splicing and non-splicing construct <<We found that the construct with splicing (Sense) had a reduced frequency of intron pop-out when RNase H1 was overexpressed (percentages in bold in Figure 5b, c and Supplementary Table 4), while the ratio of the intron retention frequencies of the Sense vs. the Branch Δ was closer to 1 when RNase H1 was overexpressed (Figure 5c and Supplementary Table 4). In particular, the difference between the Sense and Branch Δ frequency of intron deletion via MMEJ was less prominent when RNase H1 was overexpressed (Figure 5b), with the median Sense/Branch Δ ratio of individual MMEJ frequencies dropping from 1.95 to 1.45 (Figure 5d). Interestingly, the outlier-data points in Figure 5d, which represent ratios obtained from the EE2R_GAAG microhomologies (thick, red arrows) that are the most distant from each other, and from the EE12R_GTC microhomologies (thin, yellow arrows) that are the closest to each other (Extended Data Figure 9X), highlight a major effect of RNase H1 on microhomologies that are distant from each other vs. those that are close to each other (Figure 5d, e, and Extended Data Figure 12a).>> on page 7, lines 221-232. We cannot say whether the spliced transcript RNA facilitates intron pop-out more than the non-spliced transcript RNA facilitates intron retention. This may be affected by the level of transcription, by the extension of the RNA-DNA interaction, and/or other factors, and would thus require further studies employing new construct designs and/or targeting different genomic sites. To clarify this point, we included these sentences in the Discussion on page 11, lines 340-343: <<We cannot say whether the spliced transcript RNA facilitates intron pop-out more than the non-spliced transcript RNA facilitates intron retention. This may be affected by the level of transcription, by the extension of the RNA-DNA interaction, and/or other factors, and would thus require further studies employing new construct designs and/or targeting different genomic sites>>, as well as on page 14, lines 435-438:<<Future work, directed at investigating not just genetic players, e.g., the role of the reverse transcriptase activity of Pol θ ²⁴ in R-MMEJ, but also how the position of the DSB(s) relative to exon and intron sequences of genes affects RNA-mediated DSB repair, will help us understand the dynamics and the impact of RNA-mediated DSB repair in human cells.>>.

d) The statement on page 7 lines 221-222 was modified to <<Overall, the results obtained using the antisense constructs suggest that not just sense, but also antisense RNA has the capacity to promote double-strand gap repair by NHEJ or MMEJ in a sequence-dependent manner>> on page 9, lines 274-276.

e) As suggested by this Reviewer and the other Reviewers, we have performed additional experiments that have strengthened our conclusions supporting the role of transcript RNA in DSB repair. Two new result sections have been added on pages 7-8 and 9-10, respectively, and the corresponding findings have been presented in Figures 5 and 6, and in Extended Data Figure 17.

4. The data analysis and presentation in this manuscript are quite complicated and confusing to me. It is really difficult to read through. Based on the Method description, the frequencies of MMEJ and RNA-templated repair were calculated as the number of reads for MMEJ and RNA-templated repair divided by the total read count. However, it is not clear how to calculate the frequencies of NHEJ. I assume that they

were calculated as the number of reads for NHEJ divided by the total read count. Further, it is not clear whether NHEJ measured includes NHEJ products with longer than 130 bp (with intron), NHEJ products with shorter than 130 bp without microhomology or with less than 3-nt microhomology or both types of NHEJ products. In addition, given so many microhomology pairs used, the frequency of MMEJ for individual microhomology pairs was surprisingly high as compared to the frequency of total NHEJ. For example, the frequency of individual MMEJ Exon1-Exon 2 (EE4_GACGGC) was about 1% for the Sense construct and the frequency of NHEJ in/dels for the same Sense construct was about 4%. Lastly, considering the variations in DNA sequences and the activities of transcription and DNA replication among the Sense, Branch Δ , and pCMV Δ construct, it is problematic to use only two individual MMEJ products to represent overall MMEJ in Figure 3a. It would be better if the frequencies of overall MMEJ had been given and compared in Figure 3a.

- We have taken a lot of care in organizing the large amount of data generated in this study. In the new version of the manuscript, we have worked further to make the data and results not just stronger but clearer. For example, we have removed the section on R-TDR as this did not represent the major findings of the study. Instead, we have included two new sections, as discussed above, that provide support to the main findings of this study. We have also taken care to explain more clearly the experiments and the results presented.

The Reviewer is correct for the R-TDR and MMEJ frequency. In the Methods, we have provided more details about the classification of the reads in the following six sections: ‘Identification of reads as DSB repair by NHEJ using sequence alignment’ on page 6-7, ‘Identification of microhomology pairs and calculation of MMEJ frequency’ on page 7, ‘Categorizing 1-DSB sequencing reads into three groups: NHEJ in/dels, Error-free NHEJ/uncut, and MMEJ’ on page 8-9, ‘Categorizing 2-DSB sequencing reads into three groups: sequence with intron, DSB or double-strand gap repair via NHEJ, and DSB or double-strand gap repair via MMEJ’ on page 9, ‘Calculating the frequency of flipped-intron insertion following 2 DSBs’ on page 9, and ‘Calculating the frequency of RNA-templated DNA DSB repair for yeast samples’ on page 10. We cannot distinguish NHEJ events that restore the original sequence before the DSB because these will have the same sequence as that of the uncut constructs. The perfect repaired sequences can also be cut again until a mutation is generated in the repair process. Thus, we measured the frequency of NHEJ in/dels following 1 DSB. All Figures indicate the % of NHEJ in/dels. Therefore, 4% for the Sense shown in Figure 2 (see also Extended Data Figure 6) is the measure of NHEJ in/dels. The microhomologies identified are shown in Extended Data Figure 9b, and c, and the MMEJ frequencies obtained for each individual microhomology pair are shown in Extended Data Figure 9d. We have included a sentence on page 5, lines 134-136 to clarify this point: <<This higher exon-exon MMEJ frequency in the splicing constructs seems to compensate for the higher frequency of NHEJ in the non-splicing construct (Figure 3a and Extended Data Figure 6).

We agree with the Reviewer that it is more informative to provide the frequencies of overall MMEJ in Figure 3a, as well as 3b. The new Figure 3a and 3b shows the overall MMEJ frequencies. The MMEJ frequencies obtained for each individual microhomology pair are shown in Extended Data Figure 9d. The same was done for MMEJ results after 2 DSBs with results presented in Figure 4c. Likely, all the MMEJ frequencies obtained for each individual microhomology pair for the 2 DSB experiments (see scheme in Extended Data Figure 11b and c) are shown in Extended Data Figure 11d for the sense constructs and Extended Data Figure 11e for the antisense constructs.

Moreover, in Figure 3c we have presented a boxplot showing the ratios of MMEJ frequencies from exon-exon and exon-intron microhomologies between the Sense and the Branch Δ constructs of wild-type and RNase H2A KO cells after DSB by sgRNA A or B. For this analysis all the microhomology pairs have been considered. And we have also calculated the ratio of the total exon-exon frequencies to the total exon-intron frequencies for MMEJ within the Sense, Branch Δ , and pCMV Δ libraries, shown in Extended Data Figure 9e. To better emphasize the analyses we performed on the frequencies of MMEJ we modified the text on page 4 line 130 to page 5, line 134: <<The exon-exon MMEJ had higher frequency of DSB repair for the constructs with splicing (Sense and pCMV Δ) than for the construct without splicing (Branch Δ)

(Figure 3a, and results with individual-microhomology pairs in Extended Data Figure 9d), while the exon-intron MMEJ had higher frequency of DSB repair in the construct without splicing than for those with splicing (Figure 3b, and results with individual-microhomology pairs in Extended Data Figure 9d). >>, on page 5, line 137-139: <<Next, for each microhomology pair, we calculated the ratio of the mean frequency in the Sense construct to the mean frequency in the Branch Δ construct. As shown in Figure 3c, the Sense/Branch Δ ratios for the exon-exon microhomologies were mostly higher than 1...>>, and on page 5, line 140-142: <<Also, the ratio of the total exon-exon frequencies to the total exon-intron frequencies for MMEJ within the Sense and pCMV Δ libraries was significantly higher...>>.

5. In Figure 1, in order to minimize the effect of the sequence variations between the reporter plasmids on DSB repair, the Branch Δ plasmid could be replaced by keeping the Branch site but mutating the splicing site GT/AG. In addition, sgRNA A or B without Cas9 was used as a control as stated on page 3 lines 78-79; however, Cas9 with the sgRNA empty vector or a sgRNA scramble vector should be a better control.

- The intronic splicing sites GT/AG are next to the junction with the exons. Because we generate the DSBs near the exon-intron junctions using the Cas9 endonuclease, which requires a 20-nt sgRNA to be complementary to the target sites, if we mutate one of the splicing site to make the construct with no splicing we need to use a different sgRNA to make the DSB near this position. We preferred to use the exact same sgRNAs to make the DSBs in the Sense, Branch Δ , and pCMV Δ constructs. See Figure 1a and Extended Data Figure 1 with all the sequence details. We also used the strategy suggested by this Reviewer when we worked with the antisense constructs, because we eliminated the 5'-splice site for the non-splicing construct. This is shown in Figure 1a, and all sequence details are shown in Extended Data Figure 1. In this case, we had to use two different sgRNAs (sgRNA C for the Antisense construct, and sgRNA C' for the 5'-splicing Δ construct) to induce the DSB near that specific 5'-exon-intron junction. We did take care although that the sgRNA C and C' would share the same PAM sequence (CGG) and 8 nt of the sgRNA sequence on the 3' side, which is the most important for Cas9 binding (see details in Extended Data Figure 1). To better explain the designs used for the sense and antisense constructs, we modified the text on page 2 lines 65 to page 3, line 69 and on page 8, line 243-247: <<In all three constructs, we induced a DSB in the *DsRed* gene on either side of the intron, or on both sides using two single guide RNAs (sgRNAs) with Cas9 endonuclease using the same sgRNAs cutting in the same sites on these three constructs. In all constructs, the sgRNA A binds near the junction of the 5' exon, Exon1, and the intron, and the sgRNA B binds near the junction of the intron and the 3' exon, Exon2 (Figure 1a)>>, and <<To investigate the capacity of the antisense RNA to mediate the repair of a double-strand gap in DNA, we generated a DSB on each side of the intron by using both sgRNA C and D to cut the Antisense construct, and C' and D to cut the 5'-Splicing Δ construct (Figure 1a and Extended Data Figure 1). The respective sgRNAs cut the Antisense and 5'-Splicing Δ constructs similarly *in vitro* using Cas9 (Extended Data Figure 3b)>>, respectively. For the Branch Δ construct in yeast chromosomal DNA, we deleted 33 bp around the branch site so that we could use the same sgRNA C and D both for the Antisense and the Branch Δ constructs. This is described on page 9, lines 281-285: <> and detailed in Extended Data Figure 13.

The control without Cas9, lacking the sgRNA, or carrying a scrambled sgRNA would give a similar result in terms of not inducing a DSB on any of our plasmid constructs. However, we note that the control without Cas9 is not our sole negative control, in fact, we also used as negative control the analysis of a sequence 30 bp downstream of the DSB site in all the constructs that were co-transfected with the Cas9 and sgRNA plasmids. The analysis of such control sequence revealed no in/dels (this is shown in Extended

Data Figure 5). To better emphasize our controls, we added a sentence in the text on page 3, lines 83-85 <<As a No-DSB control, each plasmid was also co-transfected in the wild-type or RNase H2A KO cells with sgRNA A and B, but without Cas9. In addition, as discussed below, a sequence 30 bp distant from the DSB site was also used for no-DSB control>>, on page 3, lines 98-99 <<The in/del signature was practically absent in the no-DSB controls, as well as in the control sequences 30 bp downstream from the DSB sites (Extended Data Figure 5a, b).>>, and in the legend of Extended Data Figure 5: <<including the controls (No-DSB and DSB-sequence windows 30 bp downstream from the DSB);>>. For the yeast experiments, as no-DSB control, we did not add the galactose into the culture that is needed to activate the expression of the Cas9 endonuclease gene. This control for the yeast experiment is shown in the new Extended Data Figure 16. Specifically, we state on page 9, line 297-298: <<The in/del signature was much reduced or practically absent in the No-DSB controls, in which galactose was not added to activate Cas9 expression (Extended Data Figure 16a)>>, and on page 10, line 319-321: <<The in/del signature was much reduced or practically absent in the No-DSB controls, in which galactose was not added to activate Cas9 expression (Extended Data Figure 16b, c)>>.

6. In Extended Data Fig. 9, it is clear that PCR and targeted amplicon deep sequencing are biased towards to deletion on the intron side due to the primer design. In our experience, the length of deletions in Cas9-induced NHEJ is largely symmetric among a large pool of NHEJ products. It is unclear whether this bias interfere with the analysis of MMEJ.

- Extended Data Figure 9 in the old and new version of the manuscript shows the microhomology schemes and the results of MMEJ for the individual microhomologies following DSB by sgRNA A or B, as well as the ratio of MMEJ frequency for the exon-exon and exon-intron microhomologies. We stress that the splicing and the non-splicing constructs are treated in the same manner, using the same sgRNAs, the same Cas9, and have been amplified in PCR and sequencing using the same primer sequences. Following DSB by sgRNA A or B, the distance between the two exons is shorter in the Branch Δ construct compared to the Sense or the pCMV Δ construct because the intron is lacking the branch site. Thus, this should facilitate MMEJ between exon-exon microhomologies in the Branch Δ construct, but we consistently observe the opposite results. Moreover, for the DSB by sgRNA A the distance between the Exon1-Intron microhomologies is the same for the Sense, pCMV Δ and Branch Δ constructs (see Extended Data Fig. 9b), yet Exon1-Intron MMEJ is more efficient for the Branch Δ construct. We added a few sentences on page 5, lines 146-153: <<Following DSB by sgRNA A or B, the distance between the two exons is shorter in the Branch Δ construct compared to the Sense or the pCMV Δ construct because the intron lacks the branch site. This shorter distance should facilitate MMEJ between exon-exon microhomologies in the Branch Δ construct, but we consistently observe the opposite result. Moreover, for the DSB by sgRNA A the distance between the Exon1-Intron microhomologies is the same for the Sense, pCMV Δ , and Branch Δ constructs (see Extended Data Fig. 9b), yet MMEJ group Exon1-Intron is more efficient for the Branch Δ construct. These results argue against biased Cas9 cleavage of the different constructs and support the role of transcript RNA in guiding MMEJ>>.

7. Although the authors provided some data to support the hypothesis that RNase H2A may disrupt RNA-mediated NHEJ, RNA-mediated MMEJ, RNA-mediated double-strand gap repair and RNA-templated repair, the observed effects of RNase H2A KO on NHEJ, MMEJ, double-strand gap repair and RNA-templated repair are not always or strongly consistent with the hypothesis. For example, in Figure 2a, RNase H2A KO does not increase NHEJ in/dels in either of the Sense, Branch Δ , and pCMV Δ constructs. In Figure 3a and 3b, RNase H2A KO does not enhance MMEJ with sgRNA A in either of the Sense, Branch Δ , and pCMV Δ constructs. In Figure 5a panel 2 vs. 4, RNase H2A KO does not enhance RNA-templated repair with sgRNA B in the Sense construct. In addition, wild-type and RNase H2A KO should have been placed together for direct comparison with either sgRNA A or B in these analyses.

- The effect of RNase H2A KO on the frequency of RNA-mediated DSB repair was minor, but still evident in some experiments, and it was pointed out when significant. We did not expect that the mutation in RNase H2A would have a major effect on the results, because not just RNase H2 but ribonuclease (RNase) H1 can also target RNA-DNA hybrids in the cells. In fact in the study by Keskin et al 2014 we had to eliminate both RNase H1 and H2 function to detect RNA-templated DNA repair (R-TDR) in the yeast cells, and elimination of single RNase H1 or H2 function did not show any R-TDR activity. While providing only a small enhancement of the capacity of RNA to impact DSB repair by NHEJ and MMEJ, the results from the RNase H2A KO cells provide also strong reproducibility of the findings. We have included new results with the overexpression of RNase H1, which are described in the text on page 7-8 in the section ‘Overexpression of RNase H1 reduces the efficiency of RNA-mediated DSB repair’ and presented in Figure 5. These new analyses show a significant effect in the opposite direction to RNase H2A KO as expected to diminish the impact of RNA in DSB repair. Moreover, as part of the work in yeast cells that we have introduced into this study, we have also examined the effect of eliminating both RNase H1 and H2 function (using the yeast *rnh1 rnh201* mutant cells) on the frequency of intron retention vs. intron pop-out for the splicing and non-splicing constructs in the yeast chromosomal DNA. In this experiment, we did see a more significant effect of lack of RNase H function on RNA-mediated DSB repair. These results are shown in Extended Data Figure 17 of the new version of the manuscript and are described in the text on page 9-10 in the new section entitled ‘An antisense transcript RNA guides double-strand gap repair in its DNA by NHEJ in yeast cells’.

The Results presented in the old version of the manuscript in Figure 5 showed frequencies of RNA-templated DNA repair (R-TDR) in both wild-type and RNase H2A KO cells. They were put on different panels because the experiments were performed separately. Our goal was to emphasize the difference of R-TDR frequency among the different constructs. We have decided to remove these results from the new version of the manuscript because they were not related to the major findings of this study, and we instead preferred to add new experiments that were in support of our major findings, shown in Figures 5 and 6, and in Extended Data Figure 17 in the new version of the manuscript.

8. In Figure 4, paired Cas9-sgRNAs are used to induce a “gap”. However, while simultaneous cleavage of target DNA by paired Cas9-sgRNAs could generate a gap and pop-out of the intron, sequential cleavage would not generate a gap and likely retains the intron. For a better analysis, possible NHEJ outcomes from simultaneous or sequential cleavage by paired Cas9-sgRNAs should be classified as did in a previous study (Guo et al, Genome Biol 19:170, 2018). In addition, accurate NHEJ of DSBs generated by simultaneous cleavage of target DNA by paired Cas9-sgRNAs can be distinguished from the uncut target and should be specifically examined. The term of “gap repair” is different for “gap repair” generally used in the field of DNA repair. This should be noted in the main text of the manuscript.

- The 2-DSB system was designed to study transcript RNA-mediated double-strand gap repair. We agree with the Reviewer that there can be sequential cleavage in addition to simultaneous cleavage when using sgRNAs targeting both sides of the intron in the examined constructs. The product of sequential cleavage would not generate the gap, like the Reviewer writes, would retain the intron as in the uncut sequence, and would contain some in/dels at the intron exon junctions or MMEJ products only on one side of the DSB. All these products of uncut and possible sequential cleavage (which are expected to be longer than 130 bp, see Methods) were classified as sequences retaining the intron. As written in the Methods on page 9 in the section titled: ‘Categorizing 2-DSB sequencing reads into three groups: sequence with intron, DSB or double-strand gap repair via NHEJ, and DSB or double-strand gap repair via MMEJ’ <<To determine whether the reads contain or not the intron, the reads were filtered by length. ... If the sequencing reads contain the intron, the length of the reads reaches the maximum length of the Illumina reads, but if not, the length of the sequencing reads is around 118 bp with variations at the DSB site caused by the repair. Therefore, all the reads that were longer than 130 bp were classified as sequences with intron. The reads

shorter than 130 bp were classified as DSB/double-strand gap repair via NHEJ or MMEJ by checking whether the reads were the products of MMEJ (see Identification of microhomology pairs and calculation of MMEJ frequency). The products of MMEJ were classified as DSB or double-strand gap repair via MMEJ, and all the other reads shorter than 130 bp were classified as DSB or double-strand gap repair via NHEJ.>>. Therefore, accurate NHEJ of DSBs generated by simultaneous cleavage of target DNA by paired Cas9-sgRNAs were clearly distinguished from the uncut target and were included in the count of NHEJ sequences. All the NGS datasets are available in the NCBI Sequence Read Archive via BioProject “PRJNA883674”. Our focus in this experiment is to study the sequences without the intron and compare their frequency between the splicing and the non-splicing constructs.

We note that when we generated a single DSB by either sgRNA A or B, only a small fraction of the reads had lost the intron (1.8% or less for the splicing constructs and 0.9% or less for the non-splicing construct), and these were mainly exon-exon MMEJ sequences (Extended Data Figure 6). Instead, when we used both sgRNA A and B to induce two DSBs, the fraction of cells lacking the intron was much larger than when we generated a single DSB (compare Figure 4a with Extended Data Figure 6). Thus, the products of intron pop-out by sequential cleavage are too low to influence the results of simultaneous cleavage with pop-out of the intron. Also the experiment of *in vitro* cleavage shows that when 2 sgRNAs and Cas9 are used to target the sequence of the splicing and non-splicing constructs, a product of 2-DSB cleavage is quite abundant (Extended Data Figure 3, for human cells constructs, and Extended Data Figure 15, for yeast cell constructs). Moreover, we remark that the splicing and non-splicing constructs were always used in the same experiment transfecting cells of the same culture at the same time and using the same sgRNAs and Cas9 plasmid sources and following the same technical procedure. We therefore believe that the frequency differences that we obtained for the splicing constructs vs. the non-splicing construct in the experiments presented in Figure 4 derive from data obtained following simultaneous cleavage of the constructs on both sides of the intron. To clarify this point we modified the text on page 6, lines 174-180: <<Though the aim here was to study double-strand gap repair, it is possible to have a repair of one DSB before the other DSB occurs by using both sgRNA A and B. However, when we generated a single DSB by either sgRNA A or B, only a small fraction of the reads had lost the intron, and these were mainly exon-exon MMEJ sequences (Extended Data Figure 6). Thus, such products of intron pop-out after repair of one DSB before the other occurs were too low to influence the results of double-strand gap repair, and would anyway be additional evidence that the spliced transcript promotes intron pop-out.>>.

The meaning of ‘gap’ or “gap repair” in our manuscript was better clarified and replaced with the term ‘double-strand gap’ or ‘double-strand gap repair’ throughout the manuscript starting from the abstract on page 1, line 25-26: <<...a transcript RNA can promote repair of a DSB or a double-strand gap in its DNA gene...>>.

9. The authors performed the NHEJ analysis with antisense RNA in Figure 4, but draw a conclusion mostly as non-coding RNA. Antisense RNA and non-coding RNA are not the same. This extension from antisense RNA to non-coding RNA should have been more cautious.

- We appreciate the Reviewer comment. While in our work the antisense sequences are non-coding because they cannot give a functional DsRed (in human cells) or His3 (in yeast cells) protein, we have been more cautious in the use of this term. The wording ‘non-coding’, previously present 14 times, has been substituted with the wording ‘antisense’ except for some occurrences mainly in the Discussion in the new version of the manuscript.

10. In Figure 5, the frequency of “RNA-templated repair” is extremely low and around 10^{-5} . This frequency is even lower than the substitution error rate ($\sim 10^{-3}$ - 10^{-4}) of NSG itself. It is possible that NHEJ could generate such a low level of the same products as the RNA-templated repair. In order to identify the product of true “RNA-templated repair”, it could be ideal that the RNA-DNA hybrid over the

DSB site is at least detected. In addition, because the Branch Δ construct produces no spliced RNA, the products of “RNA-templated repair” presented for this construct should be the products of NHEJ and be subtracted from those of the Sense and pCMV Δ constructs to get accurate measurement if RNA-templated repair were real.

- We note that R-TDR is detected following generation of 1 DSB by sgRNA A or B. In line with the thought of the Reviewer, we cannot detect R-TDR in the 2-DSB system because it can be confused with NHEJ. In the 1-DSB systems, the sequencing error occurs near the DSB site (see Extended Data Figure 5); it is highly unlikely that sequencing error leads to exact pop-out of the entire intron. We also note that R-TDR is not just low but even undetectable in yeast RNase H wild-type cells (Keskin et al., 2014; Meers et al., 2020). Revealing RNA-DNA hybrids would not inform about what is the role of the RNA in DSB repair. The RNA-DNA hybrids may also be transient but still play an important role in DSB repair. As described above, we considered that the R-TDR results do not represent the main findings of this study, and we preferred to remove them from this study to leave more space for the experiments on the overexpression of RNase H1 and those of the yeast system, which are both supportive of our findings of RNA-mediated DSB repair via NHEJ and/or MMEJ.

R-TDR sequences for the Branch Δ will look like uncut sequences. Considering the very low frequency of R-TDR detected in the human RNase H wild-type and RNase H2A KO cells, it is unlikely that the R-TDR frequency has any significant impact on the frequency of NHEJ in the 1-DSB system. The R-TDR events indeed cannot be distinguished from NHEJ in the 2-DSB system, but these represent a very small fraction of the NHEJ events, which does not affect the conclusions taken, for which the intron pop-out frequency by NHEJ for the Sense construct is 49.6% (\pm 2.37) and for the Branch Δ is 25.6% (\pm 0.84) (Figure 4a). To address the Reviewer comment in relation to the yeast system, in which not just the RNA (via R-TDR, which is a direct RNA-mediated DSB-repair mechanism) but also the cDNA copy of the RNA (which is an indirect RNA-mediated DSB-repair mechanism) can have an impact on the intron pop-out, we have used the *spt3* mutation that minimizes generation of cDNA. This is described on page 10, lines 312-314: <<To avoid cDNA interfering with double-strand gap repair via NHEJ in yeast chromosomal DNA of the *rnh1 rnh201*-null cells, we deleted the *SPT3* gene that is required for reverse transcription and formation of cDNA in yeast>>.

11. This manuscript should address or discuss a couple of issues related to the mechanisms of RNA-mediated repair and RNA-templated repair. For example, how do spliced and non-spliced RNA mediate DSB repair or serve as a template for DSB repair? When both spliced RNA and non-spliced RNA (i.e., pre-mRNA) are provided in the nucleus, how do cells choose the RNA template between spliced and non-spliced RNA for DSB repair? It is also unclear whether cleavage by Cas9 occur before or after transcription for RNA-mediated repair or RNA-templated repair. It is intuitive to speculate that cleavage by Cas9 occur just after transcription for RNA-mediated repair or RNA-templated repair. It would be better that this were discussed. If RNA transcripts mediate DSB repair or serve as a template for DSB repair, how much pairing between RNA and DNA at the broken site is required? It is unfortunate that complementarity between RNA and DNA was little addressed in this manuscript except for the disruption of RNA splicing.

- The Reviewer questions are highly valuable. It is in fact the purpose of this study to bring more insights on the mechanisms of how transcript RNA both spliced and not-spliced have a role in DSB repair. We have modified the Discussion in the new version of the manuscript to provide more insights on the role of RNA in DSB repair following the findings of the current study. On page 14, lines 435-441, we write: <<Future work, directed at investigating not just genetic players, e.g., the role of the reverse transcriptase activity of Pol η in R-MMEJ, but also how the position of the DSB(s) relative to exon and intron sequences of genes affects RNA-mediated DSB repair, will help us understand the dynamics and the impact of RNA-mediated DSB repair in human cells. For example, the choice of using spliced RNA vs. non-spliced RNA (i.e., pre-mRNA) in R-NHEJ and R-MMEJ for the splicing constructs may mainly depend on the position of the

DSB relative to the exon and intron sequences, and in part also on the distance of the microhomologies from the 3'-splice site for R-MMEJ>>.

For Cas9 to generate a DSB, the Cas9 gene introduced by the plasmid via transfection needs to be transcribed into RNA and the RNA needs to be exported from the nucleus to the cytoplasm to be translated. The Cas9 protein then needs to enter the nucleus to find its target sequence for cleavage. For the generation of the transcript RNAs of the splicing and the non-splicing constructs there is only need of transcription. We expect that the transcript RNA of the splicing and non-splicing constructs are already present at the time Cas9 cleaves the DNA of these constructs. We have included a discussion about this in the text on page 3, line 80-82: <<By the time Cas9 has been transcribed, translated, and imported into the nucleus to cut the target site(s) together with the sgRNA(s), it is expected that there is already transcript from the *DsRed* gene on the different constructs>>. In the yeast system, the Cas9 expression is induced by adding galactose to the yeast culture after growing the cells. While the transcript RNA of the splicing and non-splicing chromosomal constructs is constitutively expressed under the pTEF promoter, thus the transcript RNA is generated continuously also when the Cas9 gene is not active. This is indicated on page 9, lines 285-286:<<The Antisense construct produces a *his3* antisense RNA from the constitutive pTEF promoter...>> and on page 9, lines 289-292: <<To investigate the capacity of the yeast *his3*-antisense RNA to mediate the repair of a double-strand gap in yeast chromosomal DNA, we generated a DSB on each side of the intron in the two constructs by adding galactose to the yeast cell cultures to activate the Cas9 nuclease (Figure 6b)>>. The yeast cell cultures were grown for 2 days before adding galactose as described in the Methods on page 3, in the section titled: 'Yeast sample preparation and DSB induction': <<Single-colony isolates from the yeast strains (CM859, CM860, CM1033, CM1035, CM876, YL016, YL027, YL028, YL037, YL038, YL033, and YL034) were incubated in 25 ml YPLac liquid medium for 48 h at 30 °C with shaking. Control samples were collected for both DNA and RNA extraction using 2 ml cell culture. Then, 2.5 ml of 20 % galactose was added in the remaining medium to activate Cas9 expression from the pGal promoter, and cells were incubated for 48 h at 30 °C in the shaker, followed by genomic extraction using 1.5ml cell culture>>.

The transcripts of the splicing and the non-splicing system have long complementary sequences with the DNA they are generated from, because they are in fact generated from the DNA that is targeted for DSB by Cas9. The RNA that loses the intron after splicing is of course missing this sequence, but the sequences flanking the intron are perfectly complementary with the DNA. Considering the high frequency with which the transcript RNAs of the splicing and non-splicing constructs can affect NHEJ and/or MMEJ, we believe that transient interaction between the DNA and the RNA is sufficient to affect these events of DSB repair. Most likely a more stable RNA-DNA hybrids might be required for R-TDR due to the need of DNA synthesis on the template RNA. In the Discussion, we proposed a model of RNA-mediated DSB repair as shown in Figure 7. In line with the Reviewer queries, as presented in the Discussion, for future studies, it would be very interesting to change the position of the DSB within the intron and exon sequences and determine how this affects the role of RNA in DSB repair: << Future work, directed at investigating not just genetic players, e.g., the role of the reverse transcriptase activity of Pol θ ²⁴ in R-MMEJ, but also how the position of the DSB(s) relative to exon and intron sequences of genes affects RNA-mediated DSB repair, will help us understand the dynamics and the impact of RNA-mediated DSB repair in human cells>>, page 14, lines 435-438.

12. In this manuscript, RNA sequencing is included in the Methods section. However, no results are found from RNA sequencing in the manuscript.

- The results of RNA sequencing are presented in the Extended Data Figure 2a. See page 2, line 63-65: <<We made a third construct, called pCMV Δ , in which we removed the CMV promoter to minimize transcription of the *DsRed* gene while still allowing intron splicing (Figure 1a and Extended Data Figures 1 and 2a, b)>>. The actual RNA-sequencing reads are available in the NCBI Sequence Read Archive via BioProject "PRJNA883674. A statement in the Methods on page 10, in the section titled 'Data availability'

indicates this: <<The NGS datasets, including DNA-seq and RNA-seq data, generated during the current study are available in the NCBI's Sequence Read Archive via BioProject "PRJNA883674">>.

13. In Extended Figure 5a and 5b, a significant level of substitutions appear to occur in the setting of the no-DSB control. Why is that?

- The substitutions represent sequencing errors. This was addressed in the Methods on page 9-10, in the section titled 'Obtaining DSB-sequence windows for variation-distance graphs and variation-position histograms' in which we also provide a relevant reference: <<Because substitutions are common NGS sequence errors⁸ and substitutions were also abundant in the No-DSB negative control samples (Extended Data Figure 5), we only considered in/dels in the DSB-sequence windows in the NHEJ analysis>>. The reference is Chen, L., Liu, P., Evans, T. C., Jr. & Ettiwiller, L. M. DNA damage is a pervasive cause of sequencing errors, directly confounding variant identification. *Science* **355**, 752-756 (2017).

Reviewer #3 (Remarks to the Author):

DNA double strand breaks (DSBs) are among the most lethal form of DNA lesion; however, the mammalian cells employ multiple DSB repair pathways, such as HR, NHEJ and MMEJ. These repair pathways are highly complex, multistep process and distinct set of proteins are involved in each pathway, and the repair could be error-free or error-prone. In addition to protein factors, a plethora of studies are now showing the role of RNA in DSB repair. Storici's group have earlier provided evidence of RNA-mediated genetic recombination and published several papers and reviews on this topic. The present manuscript by Jeon et al. provided ample data to establish the role of RNA in both NHEJ and MMEJ. The authors have shown that both coding and non-coding transcript RNA facilitates DSB repair in a sequence-specific manner in human cells. However, additional experiments along with some clarifications could help strengthen the mechanistic details regarding the of role of RNA in DSB repair pathways.

- We are thankful to the Reviewer for the positive and supporting comments. Following the advice of this Reviewer and the other two Reviewers we have added additional experiments and introduces numerous clarifications to strengthen the mechanistic details regarding the of role of RNA in DSB repair pathway

1. The experimental design seems to have some inherent limitation. Multiple factors are involved that each may add up to the outcome of the experiment and interpretation of results.

a. The assay involved transfection of multiple plasmid constructs, such as, Cas9/gRNA to induce DSBs at specific site along with plasmid substrate, and DSB-containing plasmids are then allowed to repair for 6 days. It is difficult to envisage how CRISPR/Cas9 mediated DSBs can generate a gapped substrate in a synchrony. If one site is preferentially cut involving a specific sgRNA, then cells will attempt to repair the DSB.

- The plasmid constructs are very similar to each other and were treated in the same way with 4 repeats. As discussed in the response to Reviewer 2 point 8). We agree with the Reviewer that there can be sequential cleavage in addition to synchronous cleavage when using sgRNAs targeting both sides of the intron in the examined constructs. The product of sequential cleavage would not generate the gap, would retain the intron as in the uncut sequence, and would contain some in/dels at the intron exon junctions or MMEJ products only on one side of the DSB. All these products of uncut and possible sequential cleavage (which are expected to be longer than 130 bp, see Methods) were classified as sequences retaining the intron. As

written in the Methods on page 9, in the section titled ‘Categorizing 2-DSB sequencing reads into three groups: sequence with intron, DSB or double-strand gap repair via NHEJ, and DSB or double-strand gap repair via MMEJ: <<To determine whether the reads contain or not the intron, the reads were filtered by length. Illumina HiSeq 2x150 was used for the NGS in which the sequencing reads have a maximum length of 150 bp. The fragments in the NGS libraries from the sense (antisense) constructs are 229 (253) bp with the intron and 118 bp without the intron if there was no mutation. The fragments in the Branch Δ (5'-SplicingD) libraries are 174 (247) bp with the intron and 118 bp without the intron. If the sequencing reads contain the intron, the length of the reads reaches the maximum length of the Illumina reads, but if not, the length of the sequencing reads is around 118 bp with variations at the DSB site caused by the repair. Therefore, all the reads that were longer than 130 bp were classified as sequences with intron. The reads shorter than 130 bp were classified as DSB/double-strand gap repair via NHEJ or MMEJ by checking whether the reads were the products of MMEJ (see Identification of microhomology pairs and calculation of MMEJ frequency). The products of MMEJ were classified as DSB or double-strand gap repair via MMEJ, and all the other reads shorter than 130 bp were classified as DSB or double-strand gap repair via NHEJ>>. All the NGS datasets are available in the NCBI Sequence Read Archive via BioProject “PRJNA883674”. Our focus in this experiment is to study the sequences without the intron and compare their frequency between the splicing and the non-splicing constructs.

We note that when we generated a single DSB by either sgRNA A or B, only a small fraction of the reads had lost the intron (1.8% or less for the splicing constructs and 0.9% or less for the non-splicing construct), and these were mainly exon-exon MMEJ sequences (Extended Data Figure 6). Instead, when we used both sgRNA A and B to induce two DSBs, the fraction of cells lacking the intron was much larger than when we generated a single DSB (compare Figure 4a with Extended Data Figure 6). Thus, the products of intron pop-out by sequential cleavage are too low to influence the results of simultaneous cleavage with pop-out of the intron. Also the experiment of *in vitro* cleavage shows that when 2 sgRNAs and Cas9 are used to target the sequence of the splicing and non-splicing constructs, a product of 2-DSB cleavage is quite abundant (Extended Data Figure 3, for human cells constructs, and Extended Data Figure 15, for yeast cell constructs). Moreover, we remark that the splicing and non-splicing constructs were always used in the same experiment transfecting cells of the same culture at the same time and using the same sgRNAs and Cas9 plasmid sources and following the same technical procedure. We therefore believe that the frequency differences that we obtained for the splicing constructs vs. the non-splicing construct in the experiments of double-strand gap repair derive from data obtained following simultaneous cleavage of the constructs on both sides of the intron. To clarify this point we modified the text on page 6, lines 174-180: <<Though the aim here was to study double-strand gap repair, it is possible to have a repair of one DSB before the other DSB occurs by using both sgRNA A and B. However, when we generated a single DSB by either sgRNA A or B, only a small fraction of the reads had lost the intron, and these were mainly exon-exon MMEJ sequences (Extended Data Figure 6). Thus, such products of intron pop-out after repair of one DSB before the other occurs were too low to influence the results of double-strand gap repair, and would anyway be additional evidence that the spliced transcript promotes intron pop-out.>>.

b. Even if such gapped plasmid is generated in cell (which would be expectedly low in frequency) and the gap is repaired by the DSB repair pathways, then the residual CRISPR/Cas9 will continue to induce DSBs.

- The *in vitro* cleavage experiment shows that when 2 sgRNAs and Cas9 are used to target the sequence of the splicing and non-splicing constructs, a product of 2-DSB cleavage is quite abundant ranging from 89% to 79% (Extended Data Figure 3, for human cells constructs, and Extended Data Figure 15, for yeast cell constructs, respectively). Moreover, the frequency of intron pop-out by NHEJ was not low, as shown in Figure 4a, the intron pop-out frequency by NHEJ for the Sense construct is 49.6% (\pm 2.37) and for the Branch Δ is 25.6% (\pm 0.84). Once the intron sequence has been pop-out from the constructs and the gap has been repaired by the DSB repair pathways, the residual Cas9 protein cannot generate more DSBs because the sgRNA binding sites are not present anymore (Extended Data Figures 1 and 13). A statement has been

added on page 6, lines 180-182: <<Moreover, once the double-strand gap has been repaired by the DSB repair pathways there cannot be more cleavage by Cas9 because the sgRNA binding sites are no longer present (Extended Data Figure 1)>>.

c. The other concern is that the DSB-containing plasmid construct is not replication defective, enabling a plasmid to clonally expand itself and thus, making it difficult to interpret the results.

- All the splicing and non-splicing constructs contain the SV40 origin and were treated in the same manner; therefore, we do not expect any bias on the interpretation of our result. We chose to work with HEK-293T cells and plasmids with the SV40 origin to have DNA replication as on chromosomal DNA. We chose to work with the replicating system. For future work we had considered transfecting the same plasmids in the HEK-293 cells that are not expressing the T antigen. These experiments would involve DSBs in a non-replicating system; thus, a different situation from what occurs in the genome of replicating cells. Studying DSB repair by RNA in a non-replicating system is in our plan for future studies. Again, the plasmid constructs represent an advantage vs. the integrated constructs because they can be introduced into different cell lines (e.g., HEK-293T and HEK-293). As indicated above, the Sense, Branch Δ , and the pCMV Δ are very similar to each other: the Branch Δ has a 55-bp deletion containing the branch site, and the pCMV Δ does not have the CMV promoter. It is expected that if any plasmid would clonally expand itself, this would affect the different constructs in a similar manner. In the experiments we do not see major variation among the same type of constructs and the results are also reproducible across different cell lines. In fact, we obtained similar results for the Sense, Branch Δ , and the pCMV Δ constructs in the HEK-293T wild-type and in the HEK-293T RNase H2A KO cell lines. Moreover, the experiments conducted in budding yeast with the constructs inserted in chromosomal DNA produced similar results to those obtained in human cells using the plasmid constructs, presented on page 9 and 10 under the section ‘An antisense transcript RNA guides double-strand gap repair in its DNA by NHEJ in yeast cells’. To strengthen the value of using plasmid constructs and the fact that these are treated in the same manner, we modify the text on page 2, from line 55-58: <<The constructs were put on plasmid DNA to facilitate the engineering and maintain a controlled (less affected by genome mutations and/or rearrangements) and exportable (allowing to introduce and study the constructs in cells of different genotypes) system.>> and on page 2 lines 65 to page 3, line 69: <<In all three constructs, we induced a DSB in the *DsRed* gene on either side of the intron, or on both sides using two single guide RNAs (sgRNAs) with Cas9 endonuclease using the same sgRNAs cutting in the same sites on these three constructs. In all constructs, the sgRNA A binds near the junction of the 5' exon, Exon1, and the intron, and the sgRNA B binds near the junction of the intron and the 3' exon, Exon2 (Figure 1a)>>. Moreover, to point out the fact that results similar to those obtained in the HEK-293T cells using plasmid constructs were obtained in yeast cells using analogous chromosomal constructs, we added the following text in the Discussion on page 11, lines 343-346: <<The findings obtained from the plasmid constructs in human cells were supported by the results produced using the yeast chromosomal constructs, possibly indicating a conserved capacity of transcript RNA to play a direct role in DSB-repair mechanisms in eukaryotic cells>>.

d. The authors have mostly analyzed the in/dels (i.e., error prone DSB repair) as a marker of NHEJ and have themselves suggested that “.....it was not possible to distinguish the sequence of error-free repair by NHEJ from that of the uncut Constructs”. This remains a concern of the study.

- For better clarifying this point we have edited the text on page 13 line 410 to page 14, line 438: <>.

e. Once the DSBs are induced, then how the full-length RNA can be produced to serve as template for gap filling. There is a strong possibility that elongating RNA polymerase II will stall at the first site of DSB repair (exon 1). Thus, in the case of gap repair, transcript would not be available as a template. The authors need to provide further clarifications regarding those concerns to establish this assay as a reliable system to measure RNA-mediated DSB repair.

- For Cas9 to generate a DSB, the Cas9 gene introduced by the plasmid via transfection needs to be transcribed into RNA and the RNA needs to be exported from the nucleus to the cytoplasm to be translated. The Cas9 protein then needs to enter the nucleus to find its target sequence for cleavage. For the generation of the transcript RNAs of the splicing and the non-splicing constructs there is only need of transcription. We expect that the transcript RNA of the splicing and non-splicing constructs are already present at the time Cas9 cleaves the DNA of these constructs. We have included a discussion about this in the text on page 3, line 80-82: <<By the time Cas9 has been transcribed, translated, and imported into the nucleus to cut the target site(s) together with the sgRNA(s), it is expected that there is already transcript from the *DsRed* gene on the different constructs>>. For the pCMV Δ construct the transcription occurs from a cryptic promoter on the plasmid because we do have the transcript although at low level (Extended Data Figure 2). For RNA to transiently interact with the DSB ends it might be even better to have a low than a high level of transcription to allow longer time for the RNA to interact with DNA. The models we suggest for RNA-mediated NHEJ and MMEJ involve a transient interaction of the transcript RNA with the DSB ends (Figure 7), see page 11, from line 349: <<We propose two new mechanisms of RNA-mediated DSB repair: RNA-mediated NHEJ (R-NHEJ) and RNA-mediated MMEJ (R-MMEJ), in which the RNA, due to its complementarity to the DNA sequence from which it is transcribed, bridges the DSB ends in a way that facilitates NHEJ or MMEJ, respectively (Figure 7). Co-transcriptional RNA-DNA interactions are not rare, but rather prevalent and form dynamic structures occupying up to 5% of mammalian genomes under physiological conditions, with RNA-DNA hybrids extending over 50%–100% of gene body for a large fraction of genes¹⁸. In support of a model in which the transcript RNA interacts with the broken ends of its DNA to modulate DSB repair, the results shown in Figure 3 reveal... >>.

In the yeast system, the Cas9 expression is induced by adding galactose to the yeast culture after growing the cells. While the transcript RNA of the splicing and non-splicing chromosomal constructs is constitutively expressed under the pTEF promoter, thus the transcript RNA is generated continuously also when the Cas9 gene is not active. This is indicated on page 9, lines 285-286: <<The Antisense construct produces a *his3* antisense RNA from the constitutive pTEF promoter...>> and on page 9, lines 289-292: <<To investigate the capacity of the yeast *his3*-antisense RNA to mediate the repair of a double-strand gap in yeast chromosomal DNA, we generated a DSB on each side of the intron in the two constructs by adding galactose to the yeast cell cultures to activate the Cas9 nuclease (Figure 6b)>>. The yeast cell cultures were grown for 2 days before adding galactose as described in the Methods on page 3, in the section titled: ‘Yeast sample preparation and DSB induction’: <<Single-colony isolates from the yeast strains (CM859, CM860, CM1033, CM1035, CM876, YL016, YL027, YL028, YL037, YL038, YL033, and YL034) were incubated in 25 ml YPLac liquid medium for 48 h at 30 °C with shaking. Control samples were collected for both

DNA and RNA extraction using 2 ml cell culture. Then, 2.5 ml of 20 % galactose was added in the remaining medium to activate Cas9 expression from the pGal promoter, and cells were incubated for 48 h at 30 °C in the shaker, followed by genomic extraction using 1.5ml cell culture>>.

2. The authors have proposed LINE 1 as the reverse transcriptase required for R-TDR in mammalian cells. LINE 1 expression is usually highly regulated and restricted in somatic tissues and cells. However, its expression is generally higher in tumors and cancer cell lines. Thus, LINE 1 is unlikely to be the major RT in normal somatic cells for such major repair pathways. Nonetheless, to conclusively prove any involvement of LINE1 in such repair, LINE1 inhibitor should be used to evaluate its potential role in R-TDR.

- We agree with the Reviewer. As indicated also in the response to Reviewers 1 and 2, we considered that the R-TDR results do not represent the main findings of this study, and we preferred to remove them from this study to leave more space for the experiments on the overexpression of RNase H1 and those of the yeast system, which are both supportive of our findings of RNA-mediated DSB repair via NHEJ and/or MMEJ.

3. Although rarely, Cas9-induced DSBs can lead to unwanted in/dels, particularly in HEK293T cells. Efficiency of L1 transposition in HEK293T is lot higher even when compared to highly proliferative HeLa cells (PMID: 29016854). Hence, HEK293T cells may not be appropriate for such assay, as formation of in/dels can be misinterpreted as DSB repair activity.

- The point of the Reviewer is valuable. Although, all the splicing and non-splicing constructs have been treated in the same way. In each experiment, all the constructs used were transfected together, side by side using the same cell culture, with the same Cas9 and sgRNA plasmids in the same exact conditions, at the same time, and the plasmid DNA was extracted from all these samples at the same time and using the same procedure. We always used 4 repeats per sample. In addition, for all the sense constructs, we have a double set of results considering that all the experiments have been performed in wild-type as well as in RNase H2A KO cells. To clarify this point, we have edited the text to better emphasize i) the fact that the different plasmid constructs have been treated in very similar manner: on page 2 from line 55 as follows: <<The constructs were put on plasmid DNA to facilitate the engineering and maintain a controlled (less affected by genome mutations and/or rearrangements) and exportable (allowing to introduce and study the constructs in cells of different genotypes) system.>> and on page 2 lines 65 to page 3, line 69: <<In all three constructs, we induced a DSB in the *DsRed* gene on either side of the intron, or on both sides using two single guide RNAs (sgRNAs) with Cas9 endonuclease using the same sgRNAs cutting in the same sites on these three constructs. In all constructs, the sgRNA A binds near the junction of the 5' exon, Exon1, and the intron, and the sgRNA B binds near the junction of the intron and the 3' exon, Exon2 (Figure 1a).>>, and the reproducibility of the results on page 13, lines 415-417: <<The 2-DSB system gave consistent results in three different series of constructs: the sense and the antisense constructs for experiments in human cells, and the antisense constructs for the experiments in yeast cells>>. Furthermore, the yeast experiments presented in Figure 6 and Extended Data Figure 17 are in line with the results obtained in the HEK-293T cells, including those obtained in the absence of the *SPT3* gene which is required for formation of cDNA exploiting the yeast Ty retrotransposon function, discussed on page 9 and 10 in the new section entitled 'An antisense transcript RNA guides double-strand gap repair in its DNA by NHEJ in yeast cells'.

4. Three different scenarios have been tested for the role of RNA in NHEJ: a plasmid construct without capable of splicing (Branch Del), had higher frequency of NHEJ in/dels than the constructs with splicing (Sense and pCMV Del) in wild-type and more evidently in the RNase H2A KO cells. A higher frequency

of in/dels for Sense than pCMV Del (in DSB by sgRNA B). The majority of in/dels in RNase H2A KO cells had higher frequency for the Branch compared to the Sense construct. However, the specific implications of these individual observations have not been clearly explained.

- The main implication of the NHEJ results following a DSB by sgRNA A or B shown in Figure 2a is that the non-splicing construct (Branch Δ) is preferentially repaired via NHEJ compared to the splicing constructs. This is more evident in the RNase H2A KO cells. For a DSB by sgRNA A the transcripts from the splicing constructs may still carry the intron, instead when the DSB is caused by sgRNA B, there are more chances that the transcript RNAs of the splicing constructs, especially for the pCMV Δ construct (because there are less transcripts), have already lost the intron. This could explain the higher frequency of in/dels for the Sense vs. the pCMV Δ construct for a DSB by sgRNA B. To better characterize the capacity of transcript RNA to help in NHEJ, more experiments are needed in which the DSB is generated at several different positions relative to the intron and exon sequences of a gene. To clarify this point, we have edited the text on page 12 lines 389 to page 13, line 402: <<For example, when the DSB is caused by sgRNA B, there are more chances that the transcript RNAs of the splicing constructs, especially for the pCMV Δ construct (because there are less transcripts), have already lost the intron. The variation-distance graph comparing the NHEJ repair after DSB by sgRNA B for the Sense vs. the pCMV Δ construct shown in Extended Data Figure 8b highlights a more efficient NHEJ repair for all individual in/dels of the Sense construct. The difference between the NHEJ repair of the Sense vs. the pCMV Δ construct is enhanced in the RNase H2A KO cells (Extended Data Figure 8d). It is likely that the transcript RNAs of the pCMV Δ compared to those of the Sense construct have already spliced out the intron when the DSB occurs at the Intron-Exon2 junction (DSB by sgRNA B); thus, limiting the capacity of the transcript RNA of the pCMV Δ construct to repair the DSB by NHEJ. Instead, for a DSB by sgRNA A the transcripts from the splicing constructs may still carry the intron, and a low level of transcription from the pCMV Δ construct may facilitate the interaction between the RNA that still carries the intron and the DSB ends resulting in higher frequency of NHEJ repair for most of the in/dels of the pCMV Δ construct (Extended Data Figure 8b). In this case, the difference between the NHEJ repair of the Sense vs. the pCMV Δ construct is reduced in the RNase H2A KO cells (Extended Data Figure 8d).>>, and on page 14 lines 435-438: <<Future work, directed at investigating not just genetic players, e.g., the role of the reverse transcriptase activity of Pol θ in R-MMEJ, but also how the position of the DSB(s) relative to exon and intron sequences of genes affects RNA-mediated DSB repair, will help us understand the dynamics and the impact of RNA-mediated DSB repair in human cells>>.

5. Recent report has clearly indicated that RT activity of Pol θ plays a critical role in MMEJ (PMID: 34117057). Based on this result, authors need to evaluate such R-TDR or R-MMEJ in Pol θ depleted cells. Also, the authors' conclusions are mostly based on a massive scale of NGS data to identify the signature motifs for NHEJ or MMEJ. However, they have not spent much effort to biochemically characterize and confirm whether the DSB repair took place via these pathways. In this context, at least one key enzyme for NHEJ (such as Ligase IV) and MMEJ (Lig III or XRCC1) should be depleted and effect on such R-TDR should be evaluated to confirm that the repair of the DSB-containing plasmid indeed proceeded via such pathways.

- The study by Chandramouly et al., 2021 (PMID: 34117057) was cited in the previous version of the manuscript in the discussion of the R-TDR results (Ref. #23). In the new version of the manuscript, we added a sentence in the Discussion on page 14, lines 435-436: <<Future work, directed at investigating not just genetic players, e.g., the role of the reverse transcriptase activity of Pol θ in R-MMEJ...>>. We appreciate the Reviewer comment. As the Reviewer indicates, it took a huge computational effort and before that also a significant genetic engineering and molecular biology effort to uncover the capacity of transcript RNA to impact DSB-repair mechanisms. It is certainly the direction of our research to investigate the players and the molecular details of the mechanisms how RNA plays a role in DSB repair in human as

well as in yeast cells in a next phase of the work, that is beyond the scope of the current work. Nonetheless, we performed a series of additional experiments to corroborate the role of RNA and its interaction with DNA in DSB repair. We overexpressed the human RNase H1 enzyme, and we performed experiments on yeast chromosomal DNA using mutants of the RNase H1 and RNase H2 enzymes. These experiments are included in two new sections of the manuscript: ‘Overexpression of RNase H1 reduces the efficiency of RNA-mediated DSB repair’ and ‘An antisense transcript RNA guides double-strand gap repair in its DNA by NHEJ in yeast cells’. These new experiments provide further support to the conclusions taken in the manuscript.

6. Lines 129-130: “..while the exon-intron....with splicing.? Can the authors explain why it is so? Does splicing proteins play any role? Some discussion will be helpful to understand this section.

- We added an explanation in the Discussion on page 11, lines 355-364: <<In support of a model in which the transcript RNA interacts with the broken ends of its DNA to modulate DSB repair, the results shown in **Figure 3** reveal that spliced RNA promotes exon-exon MMEJ, and non-spliced RNA promotes exon-intron MMEJ. The non-spliced transcript from the Branch Δ construct retains the intron in the RNA while the transcripts from the Sense and the pCMV Δ splicing constructs do not. The transcript RNA of the Branch Δ , if it interacts with DNA, can help keep the intron microhomologies near the corresponding exon microhomologies. For the DSB by sgRNA B, the advantage of the non-spliced over the spliced constructs for MMEJ between exon-intron microhomologies is reduced in Exon2-Intron MMEJ when the microhomology in the intron is close to the beginning of Exon1 (compare MMEJ bar graphs for microhomologies EI27R_GTC or EI29R_CCTT with those for microhomologies EI37R_GTC or EI38R_TAGA in Extended Data Figure 9c and d)>>.

On the other hand, the spliced RNA transcript can help to keep Exon1 and Exon2 microhomologies in the vicinity to facilitate MMEJ between exon-exon microhomologies. This is particularly evident in the experiment of double-strand gap repair (2 DSBs) with the RNase H1 overexpression and it is discussed on page 7, lines 227-232: <<Interestingly, the outlier-data points in Figure 5d, which represent ratios obtained from the EE2R_GAAG microhomologies (thick, red arrows) that are the most distant from each other, and from the EE12R_GTC microhomologies (thin, yellow arrows) that are the closest to each other (Extended Data Figure 11b), highlight a major effect of RNase H1 on microhomologies that are distant from each other vs. those that are close to each other (Figure 5d, e, and Extended Data Figure 12a)>>.

Figure 7 of the new version of the manuscript presents the proposed models for how a transcript RNA via interaction with complementary DNA at the DSB ends could guide NHEJ or MMEJ *in cis* following a DSB or a double-strand gap in its own DNA gene.

7. The authors have provided massive amount of high-quality sequence data and these data have been carefully analyzed; however, this reviewer can't comment on bioinformatic analyses of such large amount of data due to lack of expertise.

- We thank the Reviewer for appreciating the amount and the quality of our computational work. We have worked further to make our computational work as clear as possible. The Methods section includes detailed explanations of our use of standard bioinformatics tools or our custom scripts where needed. We have also provided all the code and detailed READMEs of how to reproduce our results at the GitHub repository at https://github.com/xph9876/RNA-mediated_DSB_repair. The MMEJ and NHEJ analyses include demonstrations of the code on test data sets. The full raw data sets are available in NCBI's Sequence Read Archive via BioProject "PRJNA883674" (<https://www.ncbi.nlm.nih.gov/bioproject/?term=PRJNA883674>). We hope this will ensure that our bioinformatics analyses are accessible and reproducible to a wide audience.

REVIEWER COMMENTS

Reviewer #2 (Remarks to the Author):

In this revision, some of the issues have been adequately addressed. However, the key issues remain.

First, the revised manuscript remains over-dependent on the plasmid reporters in human cells. In the revision, the authors extended a similar assay to engineered chromosomal DNA of *Saccharomyces cerevisiae*, but not human cells. The authors explained the difficulty to copy the whole plasmid reporter assay into an endogenous site of the genome in human cells. However, it should be doable to validate the key difference in the effect of spliced RNA and unspliced RNA on DSB repair at the chromosomal level in human cells. One could just select an endogenous gene with a known Branch site and splicing sites and then use CRISPR genome editing to precisely delete the known Branch site or splicing sites. Functional disruption of this gene should not have any effect on DSB repair in order to identify the effect of spliced and unspliced RNA transcript on DSB repair at the gene. After a site-specific DSB is induced at the proper position of the endogenous gene in both WT cells and those CRISPRed cells, the effect of Branch Δ or Splicing Δ on DSB repair could be determined by comparing with spliced RNA in WT cells. It would not be that difficult to find a few genomic sites suitable for the experiment proposed in human cells.

Alternatively, the authors could deliver fully or partially complementary RNA directly into human cells where a site-specific DSB is induced at a chosen gene. Any alterations on RNA could help determine whether the pairing between DNA and complementary RNA really facilitates DSB repair or even complementary RNA directly serves as the template for DSB repair. Given that mRNA delivery is much easier and more efficient nowadays, this type of experiments with an endogenous site could be done and would be of great significance in complementing the plasmid reporter assays. Unfortunately, the revised manuscript has not done this type of experiments yet.

In this manuscript, only one construct type was used each for the Sense, Branch Δ and pCMV Δ in the plasmid reporter assay. Is it possible that sequence variations in plasmid reporter DNA, not RNA transcripts, give rise to the different effect on DSB repair, in particular between the Sense and the Branch Δ plasmid reporter? One suggestion is to use a different intron with a Branch site for the same DsRed exons to generate the new Sense construct and analyze whether the new Sense construct would generate the same results in DSB repair as the original Sense construct. Both the new Sense construct and the original Sense construct should generate the same spliced RNA transcript. If spliced RNA transcript mediates DSB repair, the same results in DSB repair would be expected for both the new Sense construct and the original Sense construct. In parallel, given the purpose of Branch Δ is to prevent splicing, deleting part of the Branch site or mutating splicing sites should have similarly different effect on DSB repair as compared to spliced RNA. Therefore, the construct type for unspliced RNA should be expanded in order to better test RNA-mediated DSB repair.

Secondly, to address the concern that multiple rounds of replication in one cell cycle may distort the effect of RNA transcript on DSB repair, the authors argue: "We chose to work with HEK-293T cells and plasmids with the SV40 origin to have DNA replication as on chromosomal DNA. We chose to work with the replicating system." However, the plasmids are replicating by multiple rounds in one cell cycle in 293T cells while the genome is

replicated only once per cycle. The effect of re-replication on DSB repair could not be excluded. It is also unknown whether RNASEH1 KO or RNASEH2A KO affect replication of these plasmids in 293T cells. Any disruption or stimulation of plasmid DNA replication could interfere analysis of RNA-mediated DSB repair.

In addition, as RNase H could disrupt the DNA-RNA hybrid including R-loop, it is possible that RNASEH1 KO or RNASEH2A KO may affect RNA stability and by extension gene expression in a global manner. While the interactions of the transcript RNA with the broken DNA ends in the absence of RNase H2 may directly mediate DSB repair more efficiently, it does not exclude the possibility that the global effect on RNA stability indirectly enhance DSB repair. RNASEH1 KO or RNASEH2A KO might even affect transfection and gene expression of plasmids including reporter plasmids and the expression plasmids for Cas9 and sgRNA in human cells, causing a change in DSB induction and DSB repair. In the end, this effect on DSB induction and DSB repair may interfere analysis of RNA-mediated DSB repair in this manuscript. However, this possibility has not been excluded in the revised manuscript.

Reviewer #3 (Remarks to the Author):

The revised version of the manuscript by Jeon et al. provided new data to establish the role of RNA in both NHEJ and MMEJ in replicating cells. In the previous version, the authors have shown that both coding and non-coding transcript RNA facilitates DSB repair in a sequence-specific manner in human cells. In this revised version, the authors have included two new experimental findings:

- (i) Overexpression of the human ribonuclease (RNase) H1 in the human cells reduces efficiency of RNA-mediated DSB repair and
- (ii) Extended the model of transcript RNA-mediated DSB repair to another higher eukaryote, yeast *Saccharomyces cerevisiae*, to show that an antisense transcript RNA guides double-strand gap repair in its DNA by NHEJ.

They also have provided further clarifications about the experimental design, based on the reviewer's comments, and modified the text accordingly. Overall, the manuscript has improved considerably. However, a few more clarifications (detailed below) would be helpful before this study is suitable for publication in Nature Communications.

1. The authors have provided clarifications about the possibility of sequential cleavage to produce a single DSB vs. synchronized cleavage at two sites to produce a gapped DSB via Cas9-sgRNA. However, it is evident from the data that such efficiency of cutting, in both the forms, are of very low efficiency. The authors admit that they could not distinguish between uncut plasmid vs. error-free NHEJ repair and together they contain more than 90% of the sequence reads. This remains a concern for the efficiency of this plasmid-based assay system to mimic DSB repair under physiological conditions in mammalian cells.

2. The authors have clarified that they reproduced the experiments four times and every time maintaining the same culture and transfection conditions using the same set of plasmid substrates /sgRNA /cas9 constructs. Also, they have provided evidence of similar NHEJ and MMEJ signatures among the various plasmids, indicating similar DSB repair mechanism is operative in each case. However, concerns remain about the transfection efficiency as multiple plasmids are being transfected together and there is no way to judge transfection efficiency of the individual plasmids. Some clarification will help understanding the data.

3. The authors negated the effect of ongoing replication by comparing the efficiency of

NHEJ/MMEJ repair within various constructs in the replicating system, which they intentionally utilized to mimic the physiological situation. Though the authors have mentioned their future goal in studying DSB repair using the same constructs in a non-replicating system, still we suggest that comparing the repair (in at least one representative plasmid) in a replicating vs. non-replicating system will help in alleviating any concern in this regard. Such study will be particularly important in non-replicating cells, as NHEJ is the predominant repair pathway and thus, the role of RNA in such repair can be analyzed.

4. The authors speculate that “by the time Cas9 has been transcribed, translated, and imported into the nucleus to cut the target site(s) together with the sgRNA(s), it is expected that there is already transcript from the DsRed gene on the different constructs”. Therefore, whether Cas9 plasmid will cut one or both sites, full length RNA transcript will be present to stabilize the DSB ends and perform RNA-mediated repair. However, there is no experimental evidence to show the presence of full-length transcript or at least the truncated transcript covering the region between sgRNA A and sgRNA B cut sides. Such evidence will help specifically to address the concern.

5. The authors reiterated the consistency of the cell culture and transfection conditions and similar pattern of results among WT vs. RNase HA KO cells (both representing HEK293T) to address the concern of any Cas9 mediated in/dels being misrepresented as NHEJ repair. However, they did not provide any experimental evidence (may be representative with one plasmid construct) that results are similar in HEK293T cells and in any other cell line, where L1 transpositions are less to begin with.

6. This manuscript is solely based on high-quality NGS data that the authors have further analyzed and extrapolated in this revised version. It remains a strength of this manuscript. However, the biochemical characterization of the pathways remain a weak section of the manuscript. At least one experiment involving depletion of Pol θ to assess its effect on the RNA-mediated MMEJ will be helpful in further strengthening their model.

Reviewer #4 (Remarks to the Author):

The study by Jeon et al. provides evidence based on plasmid assays in human cells that RNA transcripts can participate in non-homologous end-joining (NHEJ) reactions at Cas9-induced DNA double-strand breaks (DSBs). There was unanimous criticism by ref. 1 and the other referees that the results with plasmid assays say little about RNA-mediated NHEJ ever occurring in the human genome, and certainly that these results do not support the notion of a significant role of such repair events for genome stability. Yeast experiments have been added with inducible Cas9-DSB/transcription constructs, like those used on plasmids in human cells, integrated into chromosomal DNA. The results are shown in Fig. 6 and Extended Data Fig. 15.

I agree with the criticism regarding the experimental set-up and concerns that the use of a plasmid-based repair assay in human cells, with repair events inferred from sequencing data only, is prone to artefacts and says little about the biological significance of the apparent influence of RNA transcripts on proximal DSB repair NHEJ.

The addition of experiments in yeast where DSBs are introduced on a chromosome go some way to mitigate some of the concerns. Satisfyingly, the trends observed in yeast can be

interpreted as consistent with an influence of RNA on repair. On the other hand, the yeast data shown in Fig. 6 and Ext. Data Fig. 17 appears to show huge variation of the data from experiment to experiment, and this weakens the new evidence; however, the trends observed between experimental strains appear to remain consistent, despite considerable variation in the data between biological replicates. The opportunity to exploit the yeast system to prove the participation of NHEJ factors in the repair events in questions and mitigate the referees' other concerns has not been taken, but this could probably still be done quite easily. Similarly, yeast might offer a way to address in vivo Cas9 cut efficiency, which is another concern raised and not addresses thus far.

In the revised MS, some referee questions remain unresolved, and I think the yeast system offers room for improvement. However, given the referees largely raise the same main criticisms, I would suggest that if referees 2 and 3 are enthusiastic about the added yeast experiments and satisfied that it mitigates their concerns, there isn't anything particular in referee 1's comments that should block further considerations of the MS.

Below is a point-by-point assessment of referee 1's concerns, pointing out improvements that should be made.

Point 1: Comment only.

Point 2: Requires experiments addressing chromosomal repair to support potential in vivo significance of RNA-mediated NHEJ.

This has been addressed by using the yeast model with chromosomal transcription/Cas9-cleavage construct. While a chromosomal construct in human cells might have been preferable, the results shown in Fig. 6 and Extended Data Fig. 15 show trends for the apparent yeast repair products consistent with an influence of the "splicing status" of the RNA transcribed locally. This in turn supports the notion of participation of RNA in at least some types of DSB repair reactions under the conditions employed. It is unclear why the data presented in Ext Data Fig 17 and Fig. 6c, generated by the same experimental set-up, shows such different gap repair by NHEJ frequencies. While the results do not go against the overall trends observed, this does suggest a high degree of experimental variation, potentially relating to a few other concerns raised by the referees around fluctuations in Cas9 cleavage efficiency in vivo and asynchronous cleavage where two DSBs are expected to occur simultaneously. In the yeast system routine cut efficiency could have been measured by PCR or Southern blotting (if necessary, in a repair deficient strain) to rule out these concerns.

Nonetheless, while the yeast data set is not extensive, the trends observed are consistent with the conclusions drawn from the experiments in human cells.

Point 3: Unfortunately, end joining is measured only by sequencing. There was no attempt to validate repair according to pathway by genetic means (or using an inhibitor).

This is a valid criticism (repair outcomes inferred from DNA sequencing only) that is important to address. It might have been feasible in the revision to address this in the yeast system where a simple KO of a NHEJ repair factor could have been tested for its effect on the observed repair products (potentially wiping out most repair but allowing uncut cells to survive; even simple loss of viability would offer a readout of NHEJ factor involvement).

Point 4: They must be able to exclude a possible influence on cutting by transcription or r-loops.

>This is an issue raised by all referees and is very valid. The uncertainty around in vivo cut

efficiency (see also point 2) weakens the conclusions. Different scenarios with transcription and the type of transcript made (with or without intron) apparently affecting repair, but instead affecting Cas9 cleavage and/or dwell time on the cut (with its own potential influence on the nature and/or efficiency of repair) are possible. While this is not a trivial experimental problem, the yeast system might provide a way to check this by PCR or Southern blotting, if necessary, in NHEJ repair mutants. If this is not possible, it could be argued that whether RNAs truly influence only downstream, i.e., actual repair events, is not proven beyond reasonable doubt (and this would need acknowledgement of this limitation in the MS). Again, a point raised by all referees as a limitation of the experimental set-up, so there will be a verdict on this point from referees 2 and 3.

Points 5 and 6 > resolved by additional clarifications.

Minor point 7 > I agree with the authors that the presentation of variation-distance graphs is useful.

Minor point 8: The authors do not complement RNaseH2A
Adequately mitigated by RNase H1 over-expression experiments.

Minor point 9: why assess products 6 days later, since repair should be finished far earlier?
Satisfactory explanation (sufficient time for multiple rounds of Cas9 cleavage where accurate NHEJ has occurred to allow for enrichment of in/dels to facilitate distinction from uncut) given by the authors.

Response May 18th, 2024
NCOMMS-22-47074B-Z

Summary of major revisions:

Testing new systems as suggested by Reviewers:

- **Exon-based DSB system in human cells:** Following Reviewer #2's suggestion, we generated double-strand breaks (DSBs) in Exon 1 of both splicing and non-splicing constructs. Unlike the outcomes observed for DSBs at exon-intron junctions, the repair results were similar for both constructs, demonstrating a sequence-dependent role of RNA in DSB repair, which supports our model.
- **Verification in the HEK293 cell line:** Following suggestions from Reviewers #2 and #3, we conducted tests in the human embryonic kidney (HEK293) cells, a system in which plasmid constructs do not undergo DNA replication (different from the HEK293T cells previously used). Results here aligned with those from HEK293T cells, reinforcing a direct RNA's role in DSB repair that is independent of DNA synthesis.

Addressing Reviewer concerns on Cas9 efficiency and DNA repair factors:

- **Results and Discussion text modifications:** Following suggestions from Reviewers #3 and #4, we clarified that Cas9 cleavage efficiency is robust both in vitro and in cells. Notably, in the 1-DSB system, breaks are precisely repaired, often restoring the original sequence. Conversely, the 2-DSB system displayed a high frequency of intron loss, suggesting efficient cleavage and varied repair outcomes. In the Discussion, we elaborated on the 2-DSB system's capacity to better analyze NHEJ influenced by RNA, backed by consistent results across different constructs, conditions, and cell systems. The manuscript now also emphasizes the uniformity and replicability of our experimental procedures.
- **Yeast experiments:** Prompted by Reviewer #3 and #4, we deleted the *KU70* gene in yeast, which is necessary for non-homologous end joining (NHEJ). We observed no difference in repair outcomes between splicing and non-splicing constructs, contrary to the results in wild-type cells. This suggests the role of transcript RNA via the Ku70-NHEJ mechanism.

Additional key updates:

- **Consistent and stronger post-transfection results:** The new experiments in HEK293 and HEK293T cells showed enhanced outcomes three days (compared to previously used six days) post-transfection, affirming the robustness of our findings.
- **Reduced data variation in yeast:** Adjusting the incubation time in galactose medium (from 48 h to 18 h) significantly reduced data variation, solidifying our results.
- **Modification of the manuscript title:** Considering the extensive experiments conducted in the yeast cell system and the key finding that transcript RNA significantly impacts DNA break repair outcomes via end-joining mechanisms, we have updated the manuscript title to 'RNA-mediated double-strand break repair by end-joining mechanisms' and revised the text in several places to emphasize this finding.
- **Authorship update:** Yilin Lu and Yiqi Zhang, major contributors to the new experimental work and data analyses, have been added to the list of authors.

We have incorporated significant new findings that considerably reinforce the conclusions of our study regarding the role of transcript RNA in NHEJ and MMEJ. We have thoroughly addressed all the concerns raised by the Reviewers, as detailed in the Point-by-point response file here below. Additionally, we have provided a version of the manuscript, in which the major edits are shown in red font.

Point by point response to Reviewers' comments

Reviewer #2 (Remarks to the Author):

In this revision, some of the issues have been adequately addressed. However, the key issues remain. First, the revised manuscript remains over-dependent on the plasmid reporters in human cells. In the revision, the authors extended a similar assay to engineered chromosomal DNA of *Saccharomyces cerevisiae*, but not human cells. The authors explained the difficulty to copy the whole plasmid reporter assay into an endogenous site of the genome in human cells. However, it should be doable to validate the key difference in the effect of spliced RNA and unspliced RNA on DSB repair at the chromosomal level in human cells. One could just select an endogenous gene with a known Branch site and splicing sites and then use CRISPR genome editing to precisely delete the known Branch site or splicing sites. Functional disruption of this gene should not have any effect on DSB repair in order to identify the effect of spliced and unspliced RNA transcript on DSB repair at the gene. After a site-specific DSB is induced at the proper position of the endogenous gene in both WT cells and those CRISPRed cells, the effect of Branch Δ or Splicing Δ on DSB repair could be determined by comparing with spliced RNA in WT cells. It would not be that difficult to find a few genomic sites suitable for the experiment proposed in human cells.

-We appreciate the Reviewer's comment and encouragement. While integrating the cassette into the genome is theoretically feasible (as we have successfully done in the *Saccharomyces cerevisiae* genome), achieving this in the human genome is far more complex. It is not a simple additional experiment but rather a completely new and extensive study. i) First, selecting an endogenous gene with a known branch site and splicing sites and then use CRISPR-genome editing to precisely delete the known branch site or a splicing site, the disruption of which should not have any effect on DSB repair, is not simple. The choice of the gene or genes to target is not easy. It requires not only literature search but also analysis of the RNA-seq data of the HEK293T cells to be sure the chosen gene/s do have abundant splicing and without abundant forms of alternative splicing. ii) Once selected a gene or a few genes to target, in a diploid genome both alleles of the chosen gene/s should be targeted, and the correction needs to be the same in both alleles of the chosen gene/s. For the experiments of MMEJ, the microhomologies need to be the same; thus, the two alleles need to have the same sequence. Therefore, we will not be able to perform CRISPR KO, we would need knock-in. While gene KO in human cells is relatively straight forward, gene knock-in is not. iii) After having made the constructs of branch deletion in each chromosomal allele for each chosen gene/s, it would be important to make an RNA-seq analysis of the new clones to verify that the modification introduced into the splicing site does not generate splicing forms or alternative splicing forms. If splicing still occurs, the genetic construct/s done will be useless and would need to be redone in a different locus/i or different gene/s. We note that the Branch Δ construct we made from the *DsRed* gene took a lot of engineering of the plasmid. We tried multiple deletion sizes of the branch site. Just small deletions around it always resulted in splicing of the intron because the branch site can change locus within the same intron. Only a deletion of 55 bp around the branch site in the intron of the *DsRed* gene eliminated splicing. iv) We would need to have independent clones for the branch-deletion cell line, ideally 3. v) If we were interested to test more than one genetic locus, then all experiments described here should be doubled. vi) Finally, we would be able to design

different sgRNAs and perform the DSB repair experiment and then conduct the bioinformatics analyses of the data. We also note that the newly generated cell lines with the branch deletion will be distant in terms of generations and genetic alterations from the starting cell line, and this will introduce significant variability into the experiments. Overall, these experiments will require a significant time and effort. Thus, while theoretically doable, we feel that such effort for the construction of the branch deletion in a chosen gene/s is beyond the scope of the current study.

We have addressed the Reviewer comment in two ways:

- 1) Exon-based DSB system in human cells: We have performed an additional experiment in the human cells to induce a DSB or a double-strand gap within the first exon of the *DsRed* gene in the splicing and non-splicing constructs. In this case, differently from a DSB or a double-strand gap at the intron-exon junction, both the splicing and non-splicing constructs repair the damage with similar frequency of NHEJ or MMEJ as expected because the transcript RNA shares the same complementarity with the broken DNA in both the splicing and the non-splicing constructs. These results provide a strong support to our model of RNA-mediated NHEJ and MMEJ. These new experiments and results are described in the Results in two new sections entitled 'A DSB in the exon sequence is repaired with similar end-joining frequency in the splicing and non-splicing constructs' on page 6 from line 167 to 181, and 'A double-strand gap within an exon sequence is repaired with similar end-joining frequency in the spliced and non-spliced constructs' on page 8 from line 234 to 261. The new findings are shown in Figure 1a, 2a, 3d,e, and 4a,c-e, Extended Data Figure 1, 3b, 6c,d,f, 9a,d,f, and 11a,c,f, and are discussed in the Discussion on page 16 from line 508 to 518: <<When a DSB or a double-strand gap are generated within an exon sequence, then there is no difference or much less difference in DSB-repair frequency by end joining between the splicing and the non-splicing constructs. In fact, the transcript RNAs generated from the splicing and the non-splicing constructs all retain the exon sequence around the DSB sites. Thus, the transcript RNAs affects similarly the end-joining repair in these constructs allowing the RNA to preferentially maintain the original DNA sequence before breakage. The slightly higher frequency of Exon1 segment pop-out observed in the splicing constructs compared to the non-splicing construct can be explained by a reduced level of complementarity with the DNA sequence downstream of the DSB by sgRNA J of the spliced RNA compared to the non-spliced RNA. Moreover, it is possible that the spliced RNA via its interaction downstream of the intron may enlarge the break at the DSB by sgRNA J facilitating segment loss>>, and on page 18, line 594 to 597: << The fact that a DSB or a double-strand gap within Exon1 minimized the effect of splicing in the RNA-mediated end-joining repair analyzed in this study underscores that a transcript RNA can impact DSB repair via end joining differently depending on the location of the breaks in DNA>>.
- 2) Verification in the HEK293 cell line: To follow a suggestion of this Reviewer and of Reviewer #3, to address the possibility that the plasmid constructs, going through several rounds of DNA replication in the HEK293T cells, may alter the interpretation of the results of the current study, we have performed the DSB repair experiments in the HEK293 cells, in which the plasmid constructs cannot undergo DNA replication. The results are very much in line with those obtained in the HEK293T cells, in which instead the plasmid constructs undergo DNA replication. The new findings show that RNA impacts the DSB repair outcomes in human cells independently from DNA synthesis. The results of this new set of experiments and analyses are presented in the Results in the section entitled 'The sequence of a transcript RNA guides DSB and double-strand gap repair in the corresponding DNA by NHEJ and MMEJ, independently from DNA synthesis (page 10 to 12 from line 330 to 383) and mainly displayed in Figure 6. Page 12, line 380 to 383: << In sum, the results from the HEK293 cell constructs indicate that RNA transcripts can enhance the repair of a double-strand gap via NHEJ or MMEJ pathways. These repair mechanisms are sequence-dependent and occur at the RNA corresponding DNA gene site. Importantly, they function independently of the gene region's DNA replication status>>.

Alternatively, the authors could deliver fully or partially complementary RNA directly into human cells where a site-specific DSB is induced at a chosen gene. Any alterations on RNA could help determine whether the pairing between DNA and complementary RNA really facilitates DSB repair or even complementary RNA directly serves as the template for DSB repair. Given that mRNA delivery is much easier and more efficient nowadays, this type of experiments with an endogenous site could be done and would be of great significance in complementing the plasmid reporter assays. Unfortunately, the revised manuscript has not done this type of experiments yet.

-We appreciate the Reviewer for the suggestion. Experiments in *trans* were performed in yeast and human cells using synthetic RNA-containing or RNA-only oligonucleotides and it was shown that the RNA can modify DNA although at a low frequency (see Storici et al., *Nature* 2007, and Shen et al., *Mutat Res* 2011). Transcript RNA-templated DSB repair works preferentially in *cis*, compared to in *trans* (see Keskin et al. *Nature* 2014). In RNA-mediated NHEJ and RNA-mediated MMEJ the transcript RNA needs to be present and nearby before the DSB occurs. Thus, providing the RNA in *trans* would not be optimal. While certainly doable, experiments in *trans* would represent a new study, which is beyond the scope of the current study.

In this manuscript, only one construct type was used each for the Sense, Branch Δ and pCMV Δ in the plasmid reporter assay. Is it possible that sequence variations in plasmid reporter DNA, not RNA transcripts, give rise to the different effect on DSB repair, in particular between the Sense and the Branch Δ plasmid reporter? One suggestion is to use a different intron with a Branch site for the same DsRed exons to generate the new Sense construct and analyze whether the new Sense construct would generate the same results in DSB repair as the original Sense construct. Both the new Sense construct and the original Sense construct should generate the same spliced RNA transcript. If spliced RNA transcript mediates DSB repair, the same results in DSB repair would be expected for both the new Sense construct and the original Sense construct. In parallel, given the purpose of Branch Δ is to prevent splicing, deleting part of the Branch site or mutating splicing sites should have similarly different effect on DSB repair as compared to spliced RNA. Therefore, the construct type for unspliced RNA should be expanded in order to better test RNA-mediated DSB repair.

-We appreciate the Reviewer comment and effort in finding suggestions to corroborate the validity of our findings and the suggestion to use different plasmid constructs. We do have used a different gene from the *DsRed*, which is the *his3* gene for the experiments in the yeast genome. Using the *his3* gene, which has different sequence from the *DsRed*, and using different sgRNAs from those used to target the constructs in human cells, we did obtain very similar results. All these experiments in yeast are described in manuscript in the section entitled 'An antisense transcript RNA guides double-strand gap repair in its corresponding DNA by Ku70-dependent NHEJ in yeast cells' on page 12 to 15 from line 386 to 466.

Moreover, as indicated above, we have performed an additional experiment in the human cells to induce a break or a double-strand gap within the first exon of the *DsRed* gene in the splicing and non-splicing constructs. In this case, differently from a DSB or a double-strand gap at the intron-exon junction, both the splicing and non-splicing constructs repair the damage with similar frequency of NHEJ or MMEJ as expected because the transcript RNA shares the same complementarity with the broken DNA in both the splicing and the non-splicing constructs. These results provide a strong support to our model of RNA-mediated NHEJ and MMEJ. These new experiments and results are described in the Results in two new sections entitled 'A DSB in the exon sequence is repaired with similar end-joining frequency in the splicing and non-splicing constructs' on page 6 from line 167 to 181, and 'A double-strand gap within an exon sequence is repaired with similar end-joining frequency in the spliced and non-spliced constructs' on page 8 from line 234 to 261. The new findings are shown in Figure 1a, 2a, 3d,e, and 4a,c-e, Extended Data Figure

1, 3b, 6c,d,f, 9a,d,f, and 11a,c,f, and are discussed in the Discussion on page 16 from line 508 to 518: <<When a DSB or a double-strand gap are generated within an exon sequence, then there is no difference or much less difference in DSB-repair frequency by end joining between the splicing and the non-splicing constructs. In fact, the transcript RNAs generated from the splicing and the non-splicing constructs all retain the exon sequence around the DSB sites. Thus, the transcript RNAs affects similarly the end-joining repair in these constructs allowing the RNA to preferentially maintain the original DNA sequence before breakage. The slightly higher frequency of Exon1 segment pop-out observed in the splicing constructs compared to the non-splicing construct can be explained by a reduced level of complementarity with the DNA sequence downstream of the DSB by sgRNA J of the spliced RNA compared to the non-spliced RNA. Moreover, it is possible that the spliced RNA via its interaction downstream of the intron may enlarge the break at the DSB by sgRNA J facilitating segment loss>>, and on page 18, line 594 to 597: << The fact that a DSB or a double-strand gap within Exon1 minimized the effect of splicing in the RNA-mediated end-joining repair analyzed in this study underscores that a transcript RNA can impact DSB repair via end joining differently depending on the location of the breaks in DNA>>.

Indeed, as the Reviewer suggests, to inactivate splicing, we have not only employed a construct with the branch deletion (Branch Δ) but also, we have inactivated splicing by mutating the 5'-splice site in the antisense construct. The experiments performed using the intron 5'-splice inactivation construct (5'-Splicing Δ) site also utilize different guide RNAs from those used in the Sense and Branch Δ constructs. The experiments using the 5'-Splicing Δ constructs give results that are in line with those obtained using the Branch Δ construct. These experiments and findings are described in the section 'An antisense transcript RNA guides double-strand gap repair in its DNA by NHEJ and MMEJ' on page 9 to 10, from line 289 to 328.

Secondly, to address the concern that multiple rounds of replication in one cell cycle may distort the effect of RNA transcript on DSB repair, the authors argue: "We chose to work with HEK-293T cells and plasmids with the SV40 origin to have DNA replication as on chromosomal DNA. We chose to work with the replicating system." However, the plasmids are replicating by multiple rounds in one cell cycle in 293T cells while the genome is replicated only once per cycle. The effect of re-replication on DSB repair could not be excluded. It is also unknown whether RNASEH1 KO or RNASEH2A KO affect replication of these plasmids in 293T cells. Any disruption or stimulation of plasmid DNA replication could interfere analysis of RNA-mediated DSB repair.

-We appreciate the Reviewer comment and suggestion. We agree that the plasmids can have multiple rounds of replications and that the plasmid replication can impact DSB repair of the plasmid possibly via recombination. However, all the constructs of the Sense and Antisense system would be affected in the same manner as they have small differences from each other and are treated all in the same way. Studying DSB repair by RNA in condition of no replication on the plasmid, which would occur for example if the plasmid would be introduced into HEK-293 cells, matches with our research plans. As indicated above, we have performed the DSB repair experiments in the HEK293 cells, in which the plasmid constructs cannot undergo DNA replication. The results are very much in line with those obtained in the HEK293T cells, in which instead the plasmid constructs undergo DNA replication. The new findings show that RNA impacts the DSB repair outcomes in human cells independently from DNA synthesis. The results of this new set of experiments and analyses are presented in the Results in the section entitled 'The sequence of a transcript RNA guides DSB and double-strand gap repair in the corresponding DNA by NHEJ and MMEJ independently from DNA synthesis' (page 10 to 12 from line 330 to 383) and mainly displayed in Figure 6. Page 12, line 380 to 383: << In sum, the results from the HEK293 cell constructs indicate that RNA transcripts can enhance the repair of a double-strand gap via NHEJ or MMEJ pathways. These repair mechanisms are sequence-dependent and occur at the RNA corresponding DNA gene site. Importantly,

they function independently of the gene region's DNA replication status>>. In addition, we have used the yeast cell system in which the constructs have been integrated into the genome and follow the chromosomal replication process. The results obtained with the yeast chromosomal system are in line with those we obtained with the plasmid constructs in the human cells (both in the HEK293T and in the HEK293 cells).

We underscore that the main value of our findings is revealing a new function of transcript RNA in cells, i.e., a direct function of RNA in end joining mechanisms (NHEJ and MMEJ). To better emphasize the value of our findings, we have edited the manuscript in a few sites starting from the title <<RNA-mediated double-strand break repair by end-joining mechanisms>>. The corresponding will be marked in red in the text.

We agree that it is unknown whether RNASEH1 KO or RNASEH2A KO affect replication of the plasmids in HEK293T cells. We have not done RNASH1 KO, we have done the overexpression. However, we disagree that any disruption or stimulation of plasmid DNA replication could interfere with the analysis of RNA-mediated DSB repair because all the constructs we have used in each experiment have been subjected to the same conditions and have been treated in the same manner. If for example, there is a condition that favors intron pop-out after the DSB/s, this should affect all the constructs in the same way. The splicing and non-splicing constructs are on the same plasmid backbone (pDsRed plasmid, Addgene, #54493) and are very similar to each other at the DNA level (see Methods and Figure 1a), and we use the same sgRNAs for the splicing and non-splicing constructs, which are transfected into the cells in the same experiment using the same cell culture and the same procedure. The text was edited on page 2 from line 62 to 66: <<From the Sense, we made a second construct, called Branch Δ , in which we deleted the branch region of the intron to prevent splicing of the intron, and a third construct, called pCMV Δ , in which we removed the CMV promoter to minimize transcription of the *DsRed* gene while still allowing intron splicing (Figure 1a and Extended Data Figures 1 and 2a, b)>>. On page 2, line 66 to 70 we write: << In all three constructs, we induced a DSB in the *DsRed* gene on either side of the intron, or on both sides using two single guide RNAs (sgRNAs) with Cas9 endonuclease using the same sgRNAs cutting in the same sites on these three constructs. In all constructs, the sgRNA A binds near the junction of the 5' exon, Exon1, and the intron, and the sgRNA B binds near the junction of the intron and the 3' exon, Exon2 (Figure 1a)>>. We have further edited the text on page 3 from line 76 to 80 to emphasize the accuracy and the high level of control of our experimental procedures to ensure reliable and reproducible results: <<The Sense, Branch Δ , and pCMV Δ plasmid constructs were assayed together employing the same experimental procedures and conditions. Specifically, individually, the Sense, Branch Δ , and pCMV Δ constructs were transfected four independent times each into cells of the same culture of HEK-293T wild-type cells, as well as HEK-293T knock-out cells having mutations in the catalytic subunit of RNase H2 (RNase H2A KO) (Extended Data Figure 4)>>. Moreover, we note that our results have been corroborated by extensive reproducibility of the findings. The results obtained with the Sense constructs the wild-type HEK-293T cells have been reproduced in the RNASEH2A KO cells, in the wild-type cells using the Antisense constructs, also in the experiment with the overexpression of RNASEH1, and furthermore using the yeast chromosomal system. In addition, each individual construct has been assayed four independent times in all our DSB-repair experiments. And, the newly performed experiments in the HEK293 cells, in which the constructs cannot replicate, further corroborate the validity of our findings.

In addition, as RNase H could disrupt the DNA-RNA hybrid including R-loop, it is possible that RNASEH1 KO or RNASEH2A KO may affect RNA stability and by extension gene expression in a global manner. While the interactions of the transcript RNA with the broken DNA ends in the absence of RNase H2 may directly mediate DSB repair more efficiently, it does not exclude the possibility that the global effect on RNA stability indirectly enhance DSB repair. RNASEH1 KO or RNASEH2A KO might even affect transfection and gene expression of plasmids including reporter plasmids and the expression plasmids for

Cas9 and sgRNA in human cells, causing a change in DSB induction and DSB repair. In the end, this effect on DSB induction and DSB repair may interfere analysis of RNA-mediated DSB repair in this manuscript. However, this possibility has not been excluded in the revised manuscript.

-We emphasize that our findings show that transcript RNA has a direct role in DSB repair by end joining in cells (human and yeast) expressing normal, wild-type RNase H genes. We also used human RNASEH2A KO cells, but such KO has small impact on the capacity of RNA to be involved in DSB repair by NHEJ and MMEJ. We also note that we have not worked with RNASEH1 KO in human cells, but we performed the overexpression of RNASEH1, see section entitled 'Overexpression of RNase H1 reduces the efficiency of RNA-mediated DSB repair' on page 8 to 9. The key finding is that RNA impacts DSB repair in wild-type cells. To better clarify this point, we edited the text of the Abstract on page 1 from line 24 to 25: <<Here, we show that both sense and antisense-transcript RNA facilitate DSB repair in a sequence-specific manner in wild-type human and yeast cells>>, and in the Discussion on page 15 from line 471 to 472: <<This study provides evidence that a transcript RNA mediates DNA DSB repair via end joining in a sequence-specific manner in human and yeast cells expressing wild-type RNase H genes>>, and on page 15 from line 492 to 496: <<We propose two new mechanisms of RNA-mediated DSB repair: RNA-mediated NHEJ (R-NHEJ) and RNA-mediated MMEJ (R-MMEJ), in which the RNA, due to its complementarity to the DNA sequence from which it is transcribed, bridges the DSB ends in a way that facilitates NHEJ or MMEJ, respectively in cells expressing wild-type RNase H genes (Figure 8)>>. The capacity of RNA to impact NHEJ and MMEJ in wild-type RNase H cells is different from the mechanism we published before about RNA-templated DSB repair, which occurs in the absence of both RNase H1 and H2 (Keskin et al., *Nature* 2014) because RNA-templated DSB repair requires also synthesis on the template RNA (Meers et al., *Mol Cell* 2020). Moreover, the results obtained in the human wild-type RNase H cells are also reproducible in the human RNASEH2A KO cells. Same in the yeast cells, RNA impacts end joining mechanisms in wild-type RNase H1 and H2 cells, and even stronger results are obtained in the *rmh1 rmh201*-double mutant cells.

Reviewer #3 (Remarks to the Author):

The revised version of the manuscript by Jeon et al. provided new data to establish the role of RNA in both NHEJ and MMEJ in replicating cells. In the previous version, the authors have shown that both coding and non-coding transcript RNA facilitates DSB repair in a sequence-specific manner in human cells. In this revised version, the authors have included two new experimental findings:

- (i) Overexpression of the human ribonuclease (RNase) H1 in the human cells reduces efficiency of RNA-mediated DSB repair and
- (ii) Extended the model of transcript RNA-mediated DSB repair to another higher eukaryote, yeast *Saccharomyces cerevisiae*, to show that an antisense transcript RNA guides double-strand gap repair in its DNA by NHEJ.

They also have provided further clarifications about the experimental design, based on the reviewer's comments, and modified the text accordingly. Overall, the manuscript has improved considerably. However, a few more clarifications (detailed below) would be helpful before this study is suitable for publication in *Nature Communications*.

1. The authors have provided clarifications about the possibility of sequential cleavage to produce a single DSB vs. synchronized cleavage at two sites to produce a gapped DSB via Cas9-sgRNA. However, it is evident from the data that such efficiency of cutting, in both the forms, are of very low efficiency. The authors admit that they could not distinguish between uncut plasmid vs. error-free NHEJ repair and together

they contain more than 90% of the sequence reads. This remains a concern for the efficiency of this plasmid-based assay system to mimic DSB repair under physiological conditions in mammalian cells.

-We thank the Reviewer for the positive comments and for underscoring the clarifications provided and the considerable improvement of the manuscript. Concerning the cut efficiency of Cas9 in the splicing and non-splicing constructs, we tested this in vitro and found very similar and high cut efficiency. For the experiments in the cells, we use the same Cas9 and the same sgRNA DNA preps for transfection together with the splicing and non-splicing constructs. The splicing and non-splicing constructs have the same structure. For example the Sense and the Branch Δ have the same sequence except for the absence of the branch site in the intron of the Branch Δ construct, and the cutting sites recognized by the sgRNAs with Cas9 are identical in these constructs. Testing cleavage efficiency in cells (human or yeast) is challenging because transcript RNA is directly involved in the repair process, and we cannot specifically eliminate its presence. Inhibiting general transcription would also interfere with Cas9 expression and disrupt the expression of DNA repair genes. To circumvent this concern, we have i) used the in vitro system, which is the only system without transcription and in which we do not see significant difference in the cleavage between the splicing and non-splicing constructs; ii) performed multiple repeats of our experiments and in different settings to emphasize the reproducibility of our results using the HEK293T and the HEK293 and cell lines, using the sense and antisense systems in human cells, and using the yeast system that corroborates the validity of our findings. iii) The new experiments generating 1 or 2 DSBs within the first exon of the *DsRed* gene show similar DSB-repair frequency between the splicing and non-splicing construct. On page 18 from line 594 to 597, we write in the Discussion: <<The fact that a DSB or a double-strand gap within Exon1 minimized the effect of splicing in the RNA-mediated end-joining repair analyzed in this study underscores that a transcript RNA can impact DSB repair via end joining differently depending on the location of the breaks in DNA>>.

We do not believe that the cleavage efficiency is low in the cells. In the 1-DSB system, the DSB is likely repaired precisely, preserving the original sequence, which appears as if the sequence was never cut. Conversely, in the 2-DSB system, sequences that have lost the intron occur at a relatively high frequency compared to those that retain the intron. This observation suggests that cleavage efficiency is not inherently higher in the 2-DSB system. Instead, it is more likely that the repair process in the 1-DSB system more frequently restores the original sequence, whereas the 2-DSB system leads to more varied repair outcomes.

We agree with the Reviewer that the plasmid system has some differences with the chromosomal system. We agree that there could also be events of recombination facilitated by the plasmid active replication, which may restore the original unbroken sequence. In fact, we may underestimate the capacity of RNA to impact NHEJ and MMEJ particularly following a single DSB using the plasmid system. The new results obtained in the HEK293 cells, in which the plasmid constructs do not undergo DNA replication show that RNA-mediated end joining is independent from DNA synthesis. See Results section entitled 'The sequence of a transcript RNA guides DSB and double-strand gap repair in the corresponding DNA by NHEJ and MMEJ independently from DNA synthesis (page 10 to 12 from line 330 to 383) and mainly displayed in Figure 6. Page 12, line 380 to 383: << In sum, the results from the HEK293 cell constructs indicate that RNA transcripts can enhance the repair of a double-strand gap via NHEJ or MMEJ pathways. These repair mechanisms are sequence-dependent and occur at the RNA corresponding DNA gene site. Importantly, they function independently of the gene region's DNA replication status>>.

Additionally, the plasmid system offers several advantages, enabling us to uncover RNA's direct role in DSB repair, the main finding of our study. It is a highly controlled and modular system. Working directly with human or yeast chromosomal DNA would have been practically impossible to achieve this discovery initially. All plasmid constructs are built from the same backbone (pDsRed plasmid, Addgene, #54493), thus they have identical backbone sequence, and the specific differences are clearly described and presented in Extended Data Figure 1. The Sense and the Branch Δ only differ for the deletion of the branch sequence

in the intron. The Antisense and 5'-Splicing Δ differ for 6 bp of 5'-splice site. These small differences between the splicing and non-splicing constructs of the Sense and Antisense systems gave markedly different results in NHEJ and MMEJ. Yet, the Sense and the Antisense systems gave very similar results with each other despite their different structure and different sgRNAs used. There was also strong consistency of the results between constructs transfected into the HEK293T and those transfected into the HEK293 cells. And the yeast constructs (which utilizes the *his3* gene in place of the *DsRed* gene and different sgRNAs) also gave similar results to the Sense and Antisense plasmid systems in human cells, highlighting a role of RNA in DSB repair by NHEJ and MMEJ. Our results show high reproducibility in all the independent samples for the Sense and the Antisense constructs, and also with the yeast constructs. We also agree that with all we have learned from the plasmid systems in human cells, and the chromosomal system in yeast cells, we are ready to move to a system integrated in human chromosomal DNA to study RNA-mediated NHEJ and MMEJ. However, uncovering and characterizing the mechanisms of direct RNA-mediated NHEJ and MMEJ is a step-by-step process. As explained above in the response to Reviewer #2, integration of constructs into human DNA represents a new project that goes beyond the scope of the current study.

To further clarify this point, we have also modified the text of the Results on page 3 to 4 from line 98 to 103: <<While it was not possible to distinguish the sequence of error-free repair by NHEJ from that of the uncut constructs and from constructs with error free recombination between a cut and an uncut plasmid, we searched for in/dels near the DSB site as the signature for NHEJ (see Methods). The in/del signature was practically absent in the No-DSB controls, as well as in the control sequences 30 bp downstream from the DSB sites (Extended Data Figure 5a, b). NHEJ was prevalent compared to MMEJ in all constructs (Extended Data Figure 6a, b). However, NHEJ was particularly dominant over MMEJ in the construct without splicing, Branch Δ , compared to the constructs with splicing, Sense and pCMV Δ (Extended Data Figure 6a, b)>>, and in the Discussion on page 17 to 18 from line 564 to 572: <>. We underscore the common background structure and sequence of the constructs by editing the text on page 2 from line 62 to 66: << From the Sense, we made a second construct, called Branch Δ , in which we deleted the branch region of the intron to prevent splicing of the intron, and a third construct, called pCMV Δ , in which we removed the CMV promoter to minimize transcription of the *DsRed* gene while still allowing intron splicing (Figure 1a and Extended Data Figures 1 and 2a, b)>> and that we use the same sgRNAs for the splicing and non-splicing constructs, which are transfected into the cells in the same experiment using the same cell culture and the same procedure, at the same time. We have edited the text on page 3 from line 76 to 83 to emphasize the accuracy and the high level of control of our experimental procedures to ensure reliable and reproducible results: <<The Sense, Branch Δ , and pCMV Δ plasmid constructs were assayed together in the same experiment employing the same experimental procedures and conditions. Specifically, individually, the Sense, Branch Δ , and pCMV Δ constructs were transfected four independent times each into cells of the same culture of HEK-293T wild-type cells, as well as HEK-293T knock-out cells having mutations in the catalytic subunit of RNase H2 (RNase H2A KO) (Extended Data Figure 4). In each *DsRed* construct, we transfected the same plasmid expressing Cas9 and the same plasmid producing sgRNA A or sgRNA B to generate 1 DSB, or both plasmids for sgRNA A and B to generate a double-strand gap (2 DSBs) (Figure 1a, b)>>. Moreover, we note that our results have been corroborated by extensive reproducibility of the findings. The results

obtained with the Sense constructs in the wild-type HEK-293T cells have been reproduced in the RNASEH2A KO cells, in the RNase H wild-type HEK293 cells, in the wild-type HEK293T cells using the Antisense constructs, also in the experiment with the overexpression of RNASEH1, and furthermore using the yeast chromosomal system. In addition, each individual construct has been assayed four independent times in all our DSB-repair experiments.

Furthermore, we have performed an experiment with a new set of sgRNAs (E and J) which generate DSBs only within the first exon of the *DsRed* gene of the splicing and non-splicing constructs (Sense, Branch Δ , and pCMV Δ). These new sgRNAs cleave the exon with frequency that is similar to that observed for the sgRNAs A, B, C.C', and D, see (Extended Data Figure 3). Differently from a DSB or a double-strand gap at the intron-exon junction, both the splicing and non-splicing constructs repair the damage with similar frequency of NHEJ or MMEJ as expected because the transcript RNA shares the same complementarity with the broken DNA in both the splicing and the non-splicing constructs. These results provide a strong support to our model of RNA-mediated NHEJ and MMEJ. These new experiments and findings are described in the Results in two new sections entitled 'A DSB in the exon sequence is repaired with similar end-joining frequency in the splicing and non-splicing constructs' on page 6 from line 167 to 181, and 'A double-strand gap within an exon sequence is repaired with similar end-joining frequency in the spliced and non-spliced constructs' on page 8 from line 234 to 261. The new findings are shown in Figure 1a, 2a, 3d,e, and 4a,c-e, Extended Data Figure 1, 3b, 6c,d,f 9a,d,f, and 11a,c,f, and are discussed in the Discussion on page 16 from line 508 to 518: << When a DSB or a double-strand gap are generated within an exon sequence, then there is no difference or much less difference in DSB-repair frequency by end joining between the splicing and the non-splicing constructs. In fact, the transcript RNAs generated from the splicing and the non-splicing constructs all retain the exon sequence around the DSB sites. Thus, the transcript RNAs affects similarly the end-joining repair in these constructs allowing the RNA to preferentially maintain the original DNA sequence before breakage. The slightly higher frequency of Exon1 segment pop-out observed in the splicing constructs compared to the non-splicing construct can be explained by a reduced level of complementarity with the DNA sequence downstream of the DSB by sgRNA J of the spliced RNA compared to the non-spliced RNA. Moreover, it is possible that the spliced RNA via its interaction downstream of the intron may enlarge the break at the DSB by sgRNA J facilitating segment loss>>, and on page 18 from line 594 to 597: << The fact that a DSB or a double-strand gap within Exon1 minimized the effect of splicing in the RNA-mediated end-joining repair analyzed in this study underscores that a transcript RNA can impact DSB repair via end joining differently depending on the location of the breaks in DNA>>.

2. The authors have clarified that they reproduced the experiments four times and every time maintaining the same culture and transfection conditions using the same set of plasmid substrates /sgRNA /cas9 constructs. Also, they have provided evidence of similar NHEJ and MMEJ signatures among the various plasmids, indicating similar DSB repair mechanism is operative in each case. However, concerns remain about the transfection efficiency as multiple plasmids are being transfected together and there is no way to judge transfection efficiency of the individual plasmids. Some clarification will help understanding the data.

-We appreciate the Reviewer comment. The plasmid expressing Cas9 and those expressing the sgRNAs are co-transfected from the same tube together with the splicing or non-splicing construct into different wells of the same cell culture in the same transfection experiment. Therefore, within the same experiment there is no variation of Cas9, and the sgRNA/s used between the splicing and non-splicing constructs. The splicing and non-splicing constructs are very similar to each other in terms of structure and sequence and their concentration was accurately measured. Thus, when we transfect cells with 0.4 ug of Sense construct in one well and 0.4 ug of Branch Δ construct in another well, we expect that the number of molecules

transfected are practically the same for these two constructs in each well. In addition, the same Sense, Branch Δ , and pCMV Δ constructs used for experiments with sgRNAs A and B, which cut at the exon-intron junctions, were used in experiments with a new set of sgRNAs E and J, which generate DSBs into the first exon. Using E and J sgRNAs for 1 DSB or 2 DSBs, the results showed minimal difference in NHEJ and MMEJ frequency between the splicing and non-splicing constructs supporting a role of RNA in DSB repair that affects DSB repair via NHEJ and MMEJ in a way that is dependent on the position of the DSB and the complementarity of the RNA for the DNA-DSB ends. Again results in yeast DNA support the data obtained using the plasmids system in the human cells. To clarify this point, we modified the text of the Methods on page 4, from line 176 to 180: << For the transfection experiments, the sense and antisense plasmids, as well as the Cas9, sgRNA plasmids, and the plasmids used in the experiments for RNASEH1 overexpression were all isolated by Midiprep (GeneJET Plasmid Midiprep Kit, Thermo Scientific) and their concentration was determined using the Nanodrop instrument>>. The experiments using the sgRNAs E and J are described in two new sections entitled ‘A DSB in the exon sequence is repaired with similar end-joining frequency in the splicing and non-splicing constructs’ on page 6 from line 167 to 181, and ‘A double-strand gap within an exon sequence is repaired with similar end-joining frequency in the spliced and non-spliced constructs’ on page 8 from line 234 to 261. The new findings are shown in Figure 1a, 2a, 3d,e, and 4a,c-e, Extended Data Figure 1, 3b, 6c,d,f 9a,d,f, and 11a,c,f, and are discussed in the Discussion on page 16 from line 508 to 518: << When a DSB or a double-strand gap are generated within an exon sequence...>>.

3. The authors negated the effect of ongoing replication by comparing the efficiency of NHEJ/MMEJ repair within various constructs in the replicating system, which they intentionally utilized to mimic the physiological situation. Though the authors have mentioned their future goal in studying DSB repair using the same constructs in a non-replicating system, still we suggest that comparing the repair (in at least one representative plasmid) in a replicating vs. non-replicating system will help in alleviating any concern in this regard. Such study will be particularly important in non-replicating cells, as NHEJ is the predominant repair pathway and thus, the role of RNA in such repair can be analyzed.

-We thank the Reviewer for the comment and suggestion. We have performed a new set of experiments in the HEK-293 cells, in which the plasmid constructs cannot replicate, using the Sense and the Branch Δ constructs with sgRNAs A, or A+B, as done in the HEK-293T cells. The results are very much in line with those obtained in the HEK293T cells, in which instead the plasmid constructs undergo DNA replication. The new findings show that RNA impacts the DSB repair outcomes in human cells independently from DNA synthesis. The results of this new set of experiments and analyses are presented in the Results in the section entitled ‘The sequence of a transcript RNA guides DSB and double-strand gap repair in the corresponding DNA by NHEJ and MMEJ, independently from DNA synthesis’ (page 10 to 12 from line 330 to 383) and mainly displayed in Figure 6. Page 12, line 380 to 383: << In sum, the results from the HEK293 cell constructs indicate that RNA transcripts can enhance the repair of a double-strand gap via NHEJ or MMEJ pathways. These repair mechanisms are sequence-dependent and occur at the RNA corresponding DNA gene site. Importantly, they function independently of the gene region's DNA replication status>>.

4. The authors speculate that “by the time Cas9 has been transcribed, translated, and imported into the nucleus to cut the target site(s) together with the sgRNA(s), it is expected that there is already transcript from the DsRed gene on the different constructs”. Therefore, whether Cas9 plasmid will cut one or both sites, full length RNA transcript will be present to stabilize the DSB ends and perform RNA-mediated repair. However, there is no experimental evidence to show the presence of full-length transcript or at least the truncated transcript covering the region between sgRNA A and sgRNA B cut sides. Such evidence will help specifically to address the concern.

-To address concerns regarding the availability of transcript at the time of double-strand break (DSB) induction by Cas9, we edited the text on page 3 from line 81 to 85 in the main text: <<Together with each *DsRed* construct, we transfected the same plasmid expressing Cas9 and the same plasmid producing sgRNA A or sgRNA B to generate 1 DSB, or both plasmids for sgRNA A and B to generate a double-strand gap (2 DSBs) (Figure 1a, b). By the time Cas9 has been transcribed, translated, and imported into the nucleus to cut the target site(s) together with the sgRNA(s), it is reasonable to expect that there is already transcript from the *DsRed* gene on the different constructs>>, and we provide a strong support to this proposition with the results in the yeast cells, in which the splicing and non-splicing constructs are integrated in the genome and are expressed from the constitutive pTEF promoter, while the Cas9 transcript, expressed from the galactose-inducible promoter, is turned on by addition of galactose to the medium. RNA-seq data from RNA extracts from yeast cells before addition of galactose to the culture show the presence of both the spliced and not-spliced RNA before galactose is added (Extended Data Figure 15a). These results provide clear proof that the transcript RNAs (spliced and not-spliced) of the splicing and non-splicing constructs are present before Cas9 is even transcribed, page 12 to 13, line 397 to 400: << RNA-seq data obtained from yeast cells prior to the addition of galactose reveals a significant presence of spliced transcripts (Extended Data Figure 15a), indicating that the RNA transcripts are already present before the induction of Cas9 transcription>>.

In sum, both the Sense and Branch Δ constructs, transcribed from the CMV promoter, are co-transfected with the Cas9 plasmid into human HEK293T or HEK293 cells. Given that these constructs share the same promoter as Cas9, concurrent transcription of these constructs alongside Cas9 is a reasonable expectation. This assumption is corroborated by our findings in yeast cells, where the constructs are expressed from the constitutive pTEF promoter, while Cas9 is under the control of a galactose-inducible promoter. RNA-seq data obtained from yeast cells prior to the addition of galactose revealed a significant presence of spliced transcripts (refer to Extended Data Figure 15a), indicating that the RNA transcripts necessary for facilitating RNA-mediated repair are already present before the induction of Cas9 transcription. This evidence strongly supports our model that RNA transcripts are available to stabilize DSB ends and mediate RNA-dependent repair processes immediately upon DSB induction.

5. The authors reiterated the consistency of the cell culture and transfection conditions and similar pattern of results among WT vs. RNase H KO cells (both representing HEK293T) to address the concern of any Cas9 mediated in/dels being misrepresented as NHEJ repair. However, they did not provide any experimental evidence (may be representative with one plasmid construct) that results are similar in HEK293T cells and in any other cell line, where L1 transpositions are less to begin with.

-We have conducted new experiments in the HEK-293 cell line. The results obtained in the HEK293 cells are very much in line with those obtained in the HEK293T cells. The results of this new set of experiments and analyses are presented in the Results in the section entitled ‘The sequence of a transcript RNA guides DSB and double-strand gap repair in the corresponding DNA by NHEJ and MMEJ independently from DNA synthesis’ (page 10 to 12 from line 330 to 383) and mainly displayed in Figure 6. Moreover, in the yeast cells, experiments were also conducted in the *spt3* mutant background, which suppresses the activity of the yeast retrotransposon Ty1 and Ty2, which are the most active ones. See text on page 13 from line 423 to 425:<< To avoid cDNA interfering with double-strand gap repair via NHEJ in yeast chromosomal DNA of the *rmh1 rmh201*-null cells, we deleted the *SPT3* gene that is required for reverse transcription and formation of cDNA in yeast⁵>>.

6. This manuscript is solely based on high-quality NGS data that the authors have further analyzed and extrapolated in this revised version. It remains a strength of this manuscript. However, the biochemical

characterization of the pathways remain a weak section of the manuscript. At least one experiment involving depletion of Pol θ to assess its effect on the RNA-mediated MMEJ will be helpful in further strengthening their model.

-We appreciate the fact that the Reviewer recognizes that the NGS data are a strength of our study. We also used mutants of RNASEH2A, we overexpress RNASEH1 and used yeast mutants of *RNH201* and *RNH1*. We conducted experiments supporting a role of RNA in DSB repair by NHEJ and MMEJ, with the overexpression of human RNASEH1 in the assays in human cells, and double deletion of *RNH201* and *RNH1* in yeast cells. Moreover, the new experiments we included in the revised manuscript with the sgRNAs that generate breaks by Cas9 within the first exon emphasize that the position of the DSB is important to influence the impact of RNA in DSB repair via NHEJ and MMEJ. In addition, in this study we introduce (for the first time) the use of the graph theory to generate variation distance graphs using the sequence reads generated at DSB repair sites. This is a new approach to study DSB repair, and its value has been underscored in the Discussion of the manuscript on page 16 to 17 from line 519 to 535: <<The variation-distance graphs used to analyze the sequencing reads...>>. We feel that further biochemical characterizations of the pathways are beyond the scope of this work and will be conducted in successive studies. Nonetheless, we did delete the *KU70* gene in the yeast system and studied DSB repair by RNA in these mutant cells.

The results of this new set of experiments are presented on page 14 to 15 from line 439 to 466: <<Finally, to examine whether the RNA-driven double-strand gap repair via NHEJ occurred through the Ku70/Ku80-NHEJ factor, we deleted the *KU70* gene in the yeast strains containing the *his3*-Antisense or the Branch Δ constructs generating strains YL085,6,7,8 which also express the sgRNAs C and D and Cas9 under the galactose promoter (Supplementary Table 5). Wild-type and *ku70*-null strains were incubated in the presence of galactose to activate the Cas9 nuclease and generate a double-strand gap. The galactose-incubation time was reduced from 48h to 18h to minimize variability across the different strains (see Methods). The genomic DNAs were then extracted from the wild-type and *ku70*-null strains and prepared for NGS to determine the frequency of the different repair outcomes following generation of the double-strand gap in the *his3* gene of the Sense and Branch Δ constructs. The results of the data analysis are shown in Figure 7d and No-DSB controls, in which galactose was not added preventing Cas9 expression are shown in Extended Data Figure 18c. We compared the frequencies of intron deletion obtained for the Antisense construct with those obtained for the Branch Δ construct. Like the results with 48h incubation in galactose, we found that the construct with splicing, Antisense, had a higher frequency of intron pop-out than the construct without splicing, Branch Δ (over a factor of 5) also when the cells were incubated in galactose for 18h (percentages in bold in Figure 7d). Not only the NHEJ frequency but also the frequency of MMEJ was significantly higher for the Antisense construct compared to the Branch Δ construct, likely due to the short (18h) incubation in galactose that reduces sample variability (Figure 7d). Notably, deletion of *KU70* eliminated the frequency difference for intron-pop out between the splicing and the non-splicing constructs (Figure 7d). These results suggest that the role of the spliced RNA in promoting intron deletion occurs mainly through the Ku70-NHEJ pathway. Interestingly, while yeast cell survival was profoundly reduced in the *ku70*-null cells, both those with the Antisense and those with the Branch Δ construct, as expected when cells were kept on galactose, survival calculated immediately after the 18h-galactose incubation (see Methods) was higher and showed higher percentage for the Branch Δ construct than the Antisense construct possibly indicating a greater capacity of the RNA of the Branch Δ construct to aid in gap repair. In the *ku70*-null mutants, the survival was unchanged for the Antisense construct, while for the Branch Δ construct, the survival was reduced by a factor of 3 (Figure 7d). The results may suggest that RNA may play a role in maintaining genome integrity, although further experiments are needed to fully characterize this role>>.

We also modified the text on page 18 to 19, from line 594 to 604 at the end of the Discussion: <<The fact that a DSB or a double-strand gap within Exon1 minimized the effect of splicing in the RNA-mediated end-joining repair analyzed in this study underscores that a transcript RNA can impact DSB repair via end joining differently depending on the location of the breaks in DNA. Interestingly, the predominant difference in double-strand gap repair outcomes between splicing and non-splicing constructs in yeast depended on Ku70. These results suggest that RNA-mediated end joining proceeds via a Ku70-dependent mechanism. MMEJ outcomes were generally less prominent in yeast compared to human cells. Unexpectedly, in the *ku70* mutants the MMEJ frequency was even lower than in wild-type cell. Future work will not only focus on investigating genetic factors, such as the effects of different NHEJ mutants in yeast and human cells and the role of the reverse transcriptase activity of Pol δ^{24} in R-MMEJ in human cells, but will also explore how the position of DSBs relative to exon and intron sequences affects RNA-mediated DSB repair in cells>>.

Reviewer #4 (Remarks to the Author):

The study by Jeon et al. provides evidence based on plasmid assays in human cells that RNA transcripts can participate in non-homologous end-joining (NHEJ) reactions at Cas9-induced DNA double-strand breaks (DSBs). There was unanimous criticism by ref. 1 and the other referees that the results with plasmid assays say little about RNA-mediated NHEJ ever occurring in the human genome, and certainly that these results do not support the notion of a significant role of such repair events for genome stability. Yeast experiments have been added with inducible Cas9-DSB/transcription constructs, like those used on plasmids in human cells, integrated into chromosomal DNA. The results are shown in Fig. 6 and Extended Data Fig. 15.

-We thank the Reviewer for the critique. We note Reviewer #2 here above suggests the use of plasmids carrying a “different intron with a Branch site for the same DsRed exons to generate the new Sense construct” and that Reviewer #3 does recognize the results on plasmids in the human cells and suggests testing in a different cell type, particularly in a system in which the plasmid cannot be replicated (see above). To address the critiques, we have conducted new experiments in the HEK-293 cells, in which the plasmids cannot be replicated. The results of this new set of experiments and analyses are presented in the Results in the section entitled ‘The sequence of a transcript RNA guides DSB and double-strand gap repair in the corresponding DNA by NHEJ and MMEJ, independently from DNA synthesis’ (page 10 to 12 from line 330 to 383) and mainly displayed in Figure 6. Page 12, line 380 to 383: << In sum, the results from the HEK293 cell constructs indicate that RNA transcripts can enhance the repair of a double-strand gap via NHEJ or MMEJ pathways. These repair mechanisms are sequence-dependent and occur at the RNA corresponding DNA gene site. Importantly, they function independently of the gene region's DNA replication status>>.

We do not state that our results demonstrate a role of RNA in genome stability; in the Summary, we write that << The results demonstrate an unexpected role of transcript RNA in directing the way DSBs are repaired in DNA, suggesting that RNA may directly modulate genome stability and evolution>>. From the results of the new experiments generating DSBs in the exon, we write on page 8, line 259 to 261: <<Overall, these results for the repair of the double-strand gap generated by two DSBs within Exon1 support a direct role of transcript RNA in DSB repair and a role of RNA in retaining the integrity of its complementary DNA sequence upon DNA breakage >>. Moreover, in the Discussion we write on page 18 line 587 to 589: <<This suggests that non-coding RNA, like lncRNAs, even when transcribed at low levels, could serve a more general function than previously anticipated in maintaining genomic integrity in both replicating and non-replicating cells>>, and at the end of the Discussion on page 19, from line 619 to 620: <<The findings provide new avenues for understanding mechanisms of genome integrity, genome modification, and

evolution>>. The experiments in budding yeast with the integrated constructs, provide results that are in line with the those obtained in human cells with the plasmid constructs. As described above in response to Reviewer #2, in the revised version of the manuscript, we underscore that the main value of our findings is revealing a new function of transcript RNA in cells, i.e., a direct function of RNA in end joining mechanisms (NHEJ and MMEJ). To better emphasize the value of our findings, we have edited the manuscript in a few sites starting from the title <<RNA-mediated double-strand break repair by end-joining mechanisms>>. The corresponding edits have will be marked in red in the text.

I agree with the criticism regarding the experimental set-up and concerns that the use of a plasmid-based repair assay in human cells, with repair events inferred from sequencing data only, is prone to artefacts and says little about the biological significance of the apparent influence of RNA transcripts on proximal DSB repair NHEJ.

The addition of experiments in yeast where DSBs are introduced on a chromosome go some way to mitigate some of the concerns. Satisfyingly, the trends observed in yeast can be interpreted as consistent with an influence of RNA on repair. On the other hand, the yeast data shown in Fig. 6 and Ext. Data Fig. 17 appears to show huge variation of the data from experiment to experiment, and this weakens the new evidence; however, the trends observed between experimental strains appear to remain consistent, despite considerable variation in the data between biological replicates. The opportunity to exploit the yeast system to prove the participation of NHEJ factors in the repair events in questions and mitigate the referees' other concerns has not been taken, but this could probably still be done quite easily. Similarly, yeast might offer a way to address in vivo Cas9 cut efficiency, which is another concern raised and not addresses thus far.

-We appreciate the Reviewer's comments and the fact that the Reviewer recognizes the value of the experiments in yeast cells and their consistency with those performed in the human cells. We disagree that the DNA repair events cannot be inferred from the sequencing data because NHEJ and MMEJ have very specific sequence signatures of DSB repair. However, our conclusions are not only based on sequencing data. We also overexpressed RNASEH1 and used yeast mutants of *RNH201* and *RNH1*. The experiments in yeast have more variation than those in human cells because the strains even those of the same genotype and carrying the same constructs are separated from each other by several generations, while in the plasmid experiments in the human cells, always cells of the same culture are used in each experiment. Furthermore, in the revised manuscript we performed a series of new experiments that are detailed here below.

i) To address the plasmid concern we performed a set of new experiments with introduction of DSB(s) only in the first exon of the splicing and non-splicing constructs, which showed, as expected, much more uniform DSB repair outcomes between the splicing and non-splicing constructs. These results provide a strong support to our model of RNA-mediated NHEJ and MMEJ. These new experiments and results are described in the Results in 2 new sections entitled 'A DSB in the exon sequence is repaired with similar end-joining frequency in the splicing and non-splicing constructs' on page 6 from line 167 to 181, and 'A double-strand gap within an exon sequence is repaired with similar end-joining frequency in the spliced and non-spliced constructs' on page 8 from line 234 to 261. The new findings are shown in Figure 1a, 2a, 3d,e, and 4a,c-e, Extended Data Figure 1, 3b, 6c,d,f 9a,d,f, and 11a,c,f, and are discussed in the Discussion on page 16 from line 508 to 518: << When a DSB or a double-strand gap are generated within an exon sequence, then there is no difference or much less difference in DSB-repair frequency by end joining between the splicing and the non-splicing constructs. In fact, the transcript RNAs generated from the splicing and the non-splicing constructs all retain the exon sequence around the DSB sites. Thus, the transcript RNAs affects similarly the end-joining repair in these constructs allowing the RNA to preferentially maintain the original DNA sequence before breakage>>.

ii) As also suggested by Reviewer #3, we introduced the splicing and non-splicing constructs in the HEK293 cells, in which the plasmids cannot replicate, and we reproduced the bias-DSB repair outcomes between

the splicing and non-splicing constructs independently from DNA synthesis. The new findings show that RNA impacts the DSB repair outcomes in human cells independently from DNA synthesis. The results of this new set of experiments and analyses are presented in the Results in the section entitled ‘The sequence of a transcript RNA guides DSB and double-strand gap repair in the corresponding DNA by NHEJ and MMEJ, independently from DNA synthesis’ (page 10 to 12 from line 330 to 383) and mainly displayed in Figure 6. Page 12, line 380 to 383: << In sum, the results from the HEK293 cell constructs indicate that RNA transcripts can enhance the repair of a double-strand gap via NHEJ or MMEJ pathways. These repair mechanisms are sequence-dependent and occur at the RNA corresponding DNA gene site. Importantly, they function independently of the gene region's DNA replication status>>.

iii) For the experiments in the HEK293 and a parallel set of experiments in the HEK293T cells, the plasmid constructs were extracted from the human cells 3 days following transfection and the results were consistent but stronger than those previously obtained 6 days after transfection corroborating our findings. See page 11, line 341 to 349: << Because the plasmids do not replicate in the HEK293 cells, to prevent significant dilution of the Sense and the Branch Δ constructs, the plasmid DNA of the Sense and the Branch Δ constructs were extracted after three instead of six days following transfection (see Methods). The extracted DNAs were prepared for NGS to study the sequence of the *DsRed* gene around the DSB and the DNA double-strand gap and determine the of NHEJ and MMEJ. No-DSB controls, without the Cas9 plasmid, had more than 99% of sequences retaining the original intron both for the Sense and the Branch Δ constructs in the presence of sgRNA A or both sgRNA A and B plasmids (Extended Data Figure 6e.g). The results showed strong similarity between repair outcomes obtained in the HEK293 and in the HEK293T cells (Figure 6)>>.

iv) The data variation observed in the yeast experiments was significantly mitigated by incubating the cells in galactose medium for a shorter time (18 h instead of 48 h); thus, strengthening the results. See page 14, line 450 to 455: <<Like the results with 48h incubation in galactose, we found that the construct with splicing, Antisense, had a higher frequency of intron pop-out than the construct without splicing, Branch Δ (over a factor of 5) also when the cells were incubated in galactose for 18h (percentages in bold in Figure 7d). Not only the NHEJ frequency but also the frequency of MMEJ was significantly higher for the Antisense construct compared to the Branch Δ construct, likely due to the short (18h) incubation in galactose that reduces sample variability (Figure 7d). >>.

v) We deleted the NHEJ-*KU70* gene in the yeast strains of the splicing and non-splicing system and found that in the *ku70* mutant the transcript RNA does not affect the DSB repair outcomes between the splicing and non-splicing constructs demonstrating that transcript RNA impacts DSB repair via end joining through the Ku70 pathway of NHEJ. These results are presented at the bottom of the section entitled ‘An antisense transcript RNA guides double-strand gap repair in its corresponding DNA by Ku70-dependent NHEJ in yeast cells’ and are shown in Figure 7d, see page 14, line 439 to 458: << Finally, to examine whether the RNA-driven double-strand gap repair via NHEJ occurred through the Ku70/Ku80-NHEJ factor, we deleted the *KU70* gene in the yeast strains containing the *his3*-Antisense or the Branch Δ constructs generating strains These results suggest that the role of the spliced RNA in promoting intron deletion occurs mainly through the Ku70-NHEJ pathway>>.

We feel that further biochemical characterizations of the pathways are beyond the scope of this work and will be conducted in successive studies. We modified the text on page 18 to 19, from line 594 to 604 at the end of the Discussion: <<The fact that a DSB or a double-strand gap within Exon1 minimized the effect of splicing in the RNA-mediated end-joining repair analyzed in this study underscores that a transcript RNA can impact DSB repair via end joining differently depending on the location of the breaks in DNA. Interestingly, the predominant difference in double-strand gap repair outcomes between splicing and non-splicing constructs in yeast depended on Ku70. These results suggest that RNA-mediated end joining proceeds via a Ku70-dependent mechanism. MMEJ outcomes were generally less prominent in yeast

compared to human cells. Unexpectedly, in the *ku70* mutants the MMEJ frequency was even lower than in wild-type cell. Future work will not only focus on investigating genetic factors, such as the effects of different NHEJ mutants in yeast and human cells and the role of the reverse transcriptase activity of Pol θ in R-MMEJ in human cells, but will also explore how the position of DSBs relative to exon and intron sequences affects RNA-mediated DSB repair in cells>>.

Concerning the cut efficiency of Cas9 in the splicing and non-splicing constructs, we tested this in vitro and found very similar and high cut efficiency. For the experiments in the cells, we use the same Cas9 and the same sgRNA DNA preps for transfection together with the splicing and non-splicing constructs. The splicing and non-splicing constructs have the same structure. For example the Sense and the Branch Δ have the same sequence except for the absence of the branch site in the intron of the Branch Δ construct, and the cutting sites recognized by the sgRNAs with Cas9 are identical in these constructs. Testing cleavage efficiency in cells (human or yeast) is challenging because transcript RNA is directly involved in the repair process, and we cannot specifically eliminate its presence. Inhibiting general transcription would also interfere with Cas9 expression and disrupt the expression of DNA repair genes. To circumvent this concern, we have i) used the in vitro system, which is the only system without transcription and in which we do not see significant difference in the cleavage between the splicing and non-splicing constructs; ii) performed multiple repeats of our experiments and in different settings to emphasize the reproducibility of our results using the HEK293T and the HEK293 and cell lines, using the sense and antisense systems in human cells, and using the yeast system that corroborates the validity of our findings. iii) The new experiments generating 1 or 2 DSBs within the first exon of the *DsRed* gene show similar DSB-repair frequency between the splicing and non-splicing construct. On page 18 from line 594 to 597, we write in the Discussion: <<The fact that a DSB or a double-strand gap within Exon1 minimized the effect of splicing in the RNA-mediated end-joining repair analyzed in this study underscores that a transcript RNA can impact DSB repair via end joining differently depending on the location of the breaks in DNA>>.

We do not believe that the cleavage efficiency is low in the cells. In the 1-DSB system, the DSB is likely repaired precisely, preserving the original sequence, which appears as if the sequence was never cut. Conversely, in the 2-DSB system, sequences that have lost the intron occur at a relatively high frequency compared to those that retain the intron. This observation suggests that cleavage efficiency is not inherently higher in the 2-DSB system. Instead, it is more likely that the repair process in the 1-DSB system more frequently restores the original sequence, whereas the 2-DSB system leads to more varied repair outcomes.

To further clarify this point, we have also modified the text of the Results on page 3 to 4 from line 98 to 105: <<While it was not possible to distinguish the sequence of error-free repair by NHEJ from that of the uncut constructs and from constructs with error free recombination between a cut and an uncut plasmid, we searched for in/dels near the DSB site as the signature for NHEJ (see Methods). The in/del signature was practically absent in the No-DSB controls, as well as in the control sequences 30 bp downstream from the DSB sites (Extended Data Figure 5a, b). NHEJ was prevalent compared to MMEJ in all constructs (Extended Data Figure 6a, b). However, NHEJ was particularly dominant over MMEJ in the construct without splicing, Branch Δ , compared to the constructs with splicing, Sense and pCMV Δ (Extended Data Figure 6a, b)>>, and in the Discussion on page 17 to 18 from line 564 to 572: <>. We underscore the common background structure and sequence of the constructs by editing the text on page 2 from line 62 to 66: << From the Sense, we made a second construct, called Branch Δ , in which we deleted the branch region of the intron to prevent splicing of the intron, and a third construct, called pCMV Δ , in which we removed the CMV promoter to minimize transcription of the *DsRed* gene while still allowing intron splicing (Figure 1a and Extended Data Figures 1 and 2a, b)>> and that we use the same sgRNAs for the splicing and non-splicing constructs, which are transfected into the cells in the same experiment using the same cell culture and the same procedure, at the same time. We have edited the text on page 3 from line 76 to 83 to emphasize the accuracy and the high level of control of our experimental procedures to ensure reliable and reproducible results: <<The Sense, Branch Δ , and pCMV Δ plasmid constructs were assayed together in the same experiment employing the same experimental procedures and conditions. Specifically, individually, the Sense, Branch Δ , and pCMV Δ constructs were transfected four independent times each into cells of the same culture of HEK-293T wild-type cells, as well as HEK-293T knock-out cells having mutations in the catalytic subunit of RNase H2 (RNase H2A KO) (Extended Data Figure 4). In each *DsRed* construct, we transfected the same plasmid expressing Cas9 and the same plasmid producing sgRNA A or sgRNA B to generate 1 DSB, or both plasmids for sgRNA A and B to generate a double-strand gap (2 DSBs) (Figure 1a, b)>>. Moreover, we note that our results have been corroborated by extensive reproducibility of the findings. The results obtained with the Sense constructs in the wild-type HEK-293T cells have been reproduced in the RNASEH2A KO cells, in the RNase H wild-type HEK293 cells, in the wild-type HEK293T cells using the Antisense constructs, also in the experiment with the overexpression of RNASEH1, and furthermore using the yeast chromosomal system. In addition, each individual construct has been assayed four independent times in all our DSB-repair experiments.

Furthermore, we have performed an experiment with a new set of sgRNAs (E and J) which generate DSBs only within the first exon of the *DsRed* gene of the splicing and non-splicing constructs (Sense, Branch Δ , and pCMV Δ). These new sgRNAs cleave the exon with frequency that is similar to that observed for the sgRNAs A, B, C.C', and D, see (Extended Data Figure 3). Differently from a DSB or a double-strand gap at the intron-exon junction, both the splicing and non-splicing constructs repair the damage with similar frequency of NHEJ or MMEJ as expected because the transcript RNA shares the same complementarity with the broken DNA in both the splicing and the non-splicing constructs. These results provide a strong support to our model of RNA-mediated NHEJ and MMEJ. These new experiments and findings are described in the Results in two new sections entitled 'A DSB in the exon sequence is repaired with similar end-joining frequency in the splicing and non-splicing constructs' on page 6 from line 167 to 181, and 'A double-strand gap within an exon sequence is repaired with similar end-joining frequency in the spliced and non-spliced constructs' on page 8 from line 234 to 261. The new findings are shown in Figure 1a, 2a, 3d,e, and 4a,c-e, Extended Data Figure 1, 3b, 6c,d,f 9a,d,f, and 11a,c,f, and are discussed in the Discussion on page 16 from line 508 to 518: << When a DSB or a double-strand gap are generated within an exon sequence, then there is no difference or much less difference in DSB-repair frequency by end joining between the splicing and the non-splicing constructs. In fact, the transcript RNAs generated from the splicing and the non-splicing constructs all retain the exon sequence around the DSB sites. Thus, the transcript RNAs affects similarly the end-joining repair in these constructs allowing the RNA to preferentially maintain the original DNA sequence before breakage. The slightly higher frequency of Exon1 segment pop-out observed in the splicing constructs compared to the non-splicing construct can be explained by a reduced level of complementarity with the DNA sequence downstream of the DSB by sgRNA J of the spliced RNA compared to the non-spliced RNA. Moreover, it is possible that the spliced RNA via its interaction downstream of the intron may enlarge the break at the DSB by sgRNA J facilitating segment loss>>, and on page 18 from line 594 to 597: << The fact that a DSB or a double-strand gap within Exon1 minimized the effect of splicing in the RNA-mediated end-joining repair analyzed in this study underscores that a transcript RNA can impact DSB repair via end joining differently depending on the location of the breaks in DNA>>.

In the revised MS, some referee questions remain unresolved, and I think the yeast system offers room for improvement. However, given the referees largely raise the same main criticisms, I would suggest that if referees 2 and 3 are enthusiastic about the added yeast experiments and satisfied that it mitigates their concerns, there isn't anything particular in referee 1's comments that should block further considerations of the MS.

-We appreciate the Reviewer comments. As written above, we have substantially expanded our work in human as well as in yeast cells with three new sets of experiments: generating DSBs in the first exon of the splicing and non-splicing constructs in the HEK293T cells, introducing the splicing and non-splicing constructs in the HEK-293 cells, in which the plasmids cannot replicate, and deleting the *KU70* gene in the yeast strains with the integrated splicing and non-splicing constructs. All these experiments strengthened our conclusion for a direct role of RNA in DSB repair via NHEJ and MMEJ.

Below is a point-by-point assessment of referee 1's concerns, pointing out improvements that should be made.

Point 1: Comment only.

Point 2: Requires experiments addressing chromosomal repair to support potential in vivo significance of RNA-mediated NHEJ.

This has been addressed by using the yeast model with chromosomal transcription/Cas9-cleavage construct. While a chromosomal construct in human cells might have been preferable, the results shown in Fig. 6 and Extended Data Fig. 15 show trends for the apparent yeast repair products consistent with an influence of the "splicing status" of the RNA transcribed locally. This in turn supports the notion of participation of RNA in at least some types of DSB repair reactions under the conditions employed. It is unclear why the data presented in Ext Data Fig 17 and Fig. 6c, generated by the same experimental set-up, shows such different gap repair by NHEJ frequencies. While the results do not go against the overall trends observed, this does suggest a high degree of experimental variation, potentially relating to a few other concerns raised by the referees around fluctuations in Cas9 cleavage efficiency in vivo and asynchronous cleavage where two DSBs are expected to occur simultaneously. In the yeast system routine cut efficiency could have been measured by PCR or Southern blotting (if necessary, in a repair deficient strain) to rule out these concerns. Nonetheless, while the yeast data set is not extensive, the trends observed are consistent with the conclusions drawn from the experiments in human cells.

-We appreciate the Reviewer comment supporting the significance of our experiments in the yeast cells. The data presented in Figure 7c (of the revised manuscript) are from wild-type cells, those presented in Extended Data Figure 18 are from wild-type RNase H cells that carry the *spt3* mutation to prevent effect of the cDNA in DSB repair. This was and is indicated in the text as well as in the legend of the of Extended Data Figure 18. For better clarity, we have edited the legend of the Extended Data Figure 18 to indicate that the cells, while having wild-type RNase H genes, carry the *spt3* allele. The experimental variation then is not as the Reviewer interpreted, but due to the different genotype of the cells in the experiments presented in Figure 7c and Extended Data Figure 18a. As discussed above, measuring cut efficiency in cells is not meaningful because RNA is directly involved in DSB repair. By the time we assess DSB cut efficiency, RNA may have already contributed to DSB repair, making it impossible to separate the DSB event from RNA-driven repair. This could only be done if we could completely inhibit transcription, which is not feasible in cells. Again, the in vitro cleavage experiments in part compensate for this, and further, the robust reproducibility of the results in the different systems (sense, antisense in human cells, and the yeast system) corroborate the validity of our conclusions.

Furthermore, as discussed here above, the data variation observed in the yeast experiments was significantly mitigated by incubating the cells in galactose medium for a shorter time (18 h instead of 48 h); thus, strengthening the results. See page 14, line 450 to 455: <<Like the results with 48h incubation in galactose, we found that the construct with splicing, Antisense, had a higher frequency of intron pop-out than the construct without splicing, Branch Δ (over a factor of 5) also when the cells were incubated in galactose for 18h (percentages in bold in Figure 7d). Not only the NHEJ frequency but also the frequency of MMEJ was significantly higher for the Antisense construct compared to the Branch Δ construct, likely due to the short (18h) incubation in galactose that reduces sample variability (Figure 7d). >>. These new experiments also further corroborate the validity of our results and conclusions.

Point 3: Unfortunately, end joining is measured only by sequencing. There was no attempt to validate repair according to pathway by genetic means (or using an inhibitor).

This is a valid criticism (repair outcomes inferred from DNA sequencing only) that is important to address. It might have been feasible in the revision to address this in the yeast system where a simple KO of a NHEJ repair factor could have been tested for its effect on the observed repair products (potentially wiping out most repair but allowing uncut cells to survive; even simple loss of viability would offer a readout of NHEJ factor involvement).

-As discussed above, we deleted the NHEJ-*KU70* gene in the yeast strains of the splicing and non-splicing system and found that in the *ku70* mutant the transcript RNA does not affect the DSB repair outcomes between the splicing and non-splicing constructs demonstrating that transcript RNA impacts DSB repair via end joining through the Ku70 pathway of NHEJ. These results are presented at the bottom of the section entitled 'An antisense transcript RNA guides double-strand gap repair in its corresponding DNA by Ku70-dependent NHEJ in yeast cells' and are shown in Figure 7d, see page 14, line 439 to 458: << Finally, to examine whether the RNA-driven double-strand gap repair via NHEJ occurred through the Ku70/Ku80-NHEJ factor, we deleted the *KU70* gene in the yeast strains containing the *his3*-Antisense or the Branch Δ constructs generating strains These results suggest that the role of the spliced RNA in promoting intron deletion occurs mainly through the Ku70-NHEJ pathway>>.

Point 4: They must be able to exclude a possible influence on cutting by transcription or r-loops. >This is an issue raised by all referees and is very valid. The uncertainty around in vivo cut efficiency (see also point 2) weakens the conclusions. Different scenarios with transcription and the type of transcript made (with or without intron) apparently affecting repair, but instead affecting Cas9 cleavage and/or dwell time on the cut (with its own potential influence on the nature and/or efficiency of repair) are possible. While this is not a trivial experimental problem, the yeast system might provide a way to check this by PCR or Southern blotting, if necessary, in NHEJ repair mutants. If this is not possible, it could be argued that whether RNAs truly influence only downstream, i.e., actual repair events, is not proven beyond reasonable doubt (and this would need acknowledgement of this limitation in the MS). Again, a point raised by all referees as a limitation of the experimental set-up, so there will be a verdict on this point from referees 2 and 3.

-We do appreciate the Reviewer comment and we acknowledge that while the process of intron splicing occurs at the RNA level and not on the DNA, we cannot completely rule out that somehow the splicing machinery can interfere with Cas9 cleavage. However, if for example the Cas9 cleavage would be more efficient in the Sense construct compared to the Branch Δ construct, we would expect that not just exon-exon MMEJ but also exon-intron MMEJ should be more efficient in the Sense than in the Branch Δ after 1 DSB, and in the 2 DSB systems, not just intron pop-out by NHEJ and MMEJ but also intron flipping should

be more frequent in the Sense, but our results show that this is not the case. In addition, the Sense and pCMV Δ , which have major difference in the level of transcription, while not identical (specific differences have been highlighted in the text), do show quite similar results to each other and much different from those of the Branch Δ construct. These results would support a more prominent role of RNA in influencing repair than in influencing cutting. We have modified the text of the Discussion on page 18 to 19, from line 590 to 599: <<While we cannot conclusively prove that the splicing process and/or the presence of the transcript directly influence Cas9 activity, the evidence suggests that RNA plays a significant role in mediating repair outcomes post-cleavage, rather than primarily affecting the cleavage efficiency itself. This observation aligns with the similar repair profiles seen in constructs with high and low transcription level, and with the reproducibility of the results across different cell lines, genotypes, and species. The fact that a DSB or a double-strand gap within Exon1 minimized the effect of splicing in the RNA-mediated end-joining repair analyzed in this study underscores that a transcript RNA can impact DSB repair via end joining differently depending on the location of the breaks in DNA. Interestingly, the predominant difference in double-strand gap repair outcomes between splicing and non-splicing constructs in yeast depended on Ku70. These results suggest that RNA-mediated end joining proceeds via a Ku70-dependent mechanism>>.

Points 5 and 6 > resolved by additional clarifications.

-We thank the Reviewer.

Minor point 7 > I agree with the authors that the presentation of variation-distance graphs is useful.

-We thank the Reviewer for grasping the value of the analyses of the DSB repair sequences using the variation-distance graphs. We have emphasized the value of these graphs in the Discussion on page 16 to 17, from line 519 to 535: <<The variation-distance graphs used to analyze the sequencing reads derived from R-NHEJ ...>>.

Minor point 8: The authors do not complement RNaseH2A
Adequately mitigated by RNase H1 over-expression experiments.

-We thank the Reviewer for appreciating the value of the RNASEH1 overexpression experiments.

Minor point 9: why assess products 6 days later, since repair should be finished far earlier?
Satisfactory explanation (sufficient time for multiple rounds of Cas9 cleavage where accurate NHEJ has occurred to allow for enrichment of in/dels to facilitate distinction from uncut) given by the authors.

-We appreciate the Reviewer comment and the stimulus to examine a shorter time to assess the DSB repair products. As the noted by the Reviewer, we assessed the DSB repair products after 6 days to allow more time for the repair, see Methods page 4, line 181 to 183: << Cells were incubated for 6 days, to allow for more constructs to be cut and repaired, at 37 °C in a 5% CO₂ humidified incubator after the transfection and used for the following experiments>>. Nonetheless, as indicated above, in the new set of experiments in the HEK293 and a parallel set of experiments in the HEK293T cells, the plasmid constructs were extracted from the human cells 3 days following transfection. The results were not only consistent but even stronger than those previously obtained 6 days after transfection, corroborating our findings. See Methods on page 4, line 183 to 186: << In transfections experiments using HEK293 cells and HEK293T cells as control, the human cell were transfected with the same amount of plasmid and incubated for 3 days at 37 °C in a 5%

CO2 humidified incubator. After 3 days, each HEK293 sample and HEK293T control sample were collected from three wells>>, and page 11, line 341 to 349: <<Because the plasmids do not replicate in the HEK293 cells, to prevent significant dilution of the Sense and the Branch Δ constructs, the plasmid DNA of the Sense and the Branch Δ constructs were extracted after three instead of six days following transfection (see Methods). The extracted DNAs were prepared for NGS to study the sequence of the *DsRed* gene around the DSB and the DNA double-strand gap and determine the of NHEJ and MMEJ. No-DSB controls, without the Cas9 plasmid, had more than 99% of sequences retaining the original intron both for the Sense and the Branch Δ constructs in the presence of sgRNA A or both sgRNA A and B plasmids (Extended Data Figure 6e,g). The results showed strong similarity between repair outcomes obtained in the HEK293 and in the HEK293T cells (Figure 6)>>.

REVIEWERS' COMMENTS

Reviewer #2 (Remarks to the Author):

With new data provided and revision of the manuscript, the authors have properly addressed most of the concerns. This revised manuscript is now more solid. While lack of validation at the chromosomal level in human cells remains a limitation, this revised manuscript may provide some insights into RNA-mediated DSB repair.

Reviewer #3 (Remarks to the Author):

This is the second revision of the manuscript by Jeon et al. In the earlier versions of the manuscript (original and revised), the authors have shown that both coding and non-coding transcript RNA facilitates DSB repair in a sequence-specific manner in yeast and human cells. In this version, the authors have included a few new experimental findings involving both mammalian and in yeast cells to establish the role of RNA in both NHEJ and MMEJ:

1. The authors have used an Exon-based DSB system in human cells where they have induced DSBs in the exon and provided evidence that the repair frequency is similar with spliced and non-spliced transcript, further indicating that sequence directed RNA-mediated repair is operative.

2. They have verified some critical data in HEK293 cells and have utilized this cellular model with the anticipation that the transfected plasmids couldn't replicate in HEK293 cells due to the absence of the T-antigen. This data provided evidence that the RNA-mediated repair was independent of the DNA synthesis. However, an appropriate reference would be helpful to the common readers.

3. Furthermore, they have provided evidence for the involvement of NHEJ pathway by deleting KU70 gene in yeast.

Apart from this additional evidence, the authors have mostly provided long explanations/clarifications for a large part of the concerns. Overall, this manuscript has improved from its previous version. However, still several concerns remain, mostly regarding the lack of providing human cellular and biochemical data as detailed below:

Comment 1: Abstract: "While RNA can be a template for HR, the direct role and extent of transcript RNA in DSB-repair via end joining, as well as its overall impact on repair outcomes, remain unknown." While searching the literature, this reviewer finds three reports (PMIDs: 27703167; 32205441; 36758800) that have indeed shown that NHEJ-mediated repair of DSBs in mammalian cells can utilize nascent RNA as template. A recent report also showed that DNA polymerase θ , a polymerase involved in MMEJ repair, promotes RNA-templated repair in mammalian cells (PMID: 34117057). Most of these publications have already been cited by the authors and thus, the statement could be modified accordingly in the proper context.

Comment 2: The authors earlier speculated that "by the time Cas9 has been transcribed, translated, and imported into the nucleus to cut the target site(s) together with the sgRNA(s), it is expected that there is already transcript from the DsRed gene on the different constructs". The authors have provided a conclusive evidence from the genome integrated constructs in the yeast system where both spliced and non-spliced RNAs are present as evident from the RNA-seq data before the Cas9 is induced by the addition of galactose to the medium. However, the authors have simply modified the text without any experimental evidence in mammalian cells. In mammalian cells, all the plasmid transfections (DSB harboring DsRed construct, Cas9 and sgRNAs) are done together and there is no inducible system for controlling the expression of the sgRNAs, and thus, it is a more complicated

system and difficult to follow and interpret the results. A more detailed discussion will clarify the issue.

Comment 3: To address the concern regarding the lack of biochemical characterization of the repair pathways, the authors used a KU70 deleted yeast strain to perform the experiments. The intron-pop out frequency was similar in spliced vs non-spliced constructs indicating KU70-mediated NHEJ was responsible for such spliced RNA mediated intron deletion. They also have observed higher percentage of repair and therefore, survival for the branch-deletion construct that was significantly (3-fold) reduced in KU70 null-strains. This remains a good control to show R-NHEJ pathway is active in yeast. However, the MMEJ repair pathway was not well investigated. The authors could deplete one MMEJ component in mammalian cells to establish the pathway. Thus, overall, the data in yeast was much stronger compared to the results involving mammalian cells.

Reviewer #4 (Remarks to the Author):

Storici and co-workers have provided a substantial experimental revision of the MS. The new data strengthens the conclusion that RNA transcripts influence DNA double-strand break repair outcomes along NHEJ outcomes (this is now more clearly demonstrated with a NHEJ repair mutant in Fig. 7) in yeast and human.

A remaining caveat regarding in vivo Cas9 cut efficiency is adequately acknowledged in the revised MS.

The concerns of this reviewer have been sufficiently addressed in this substantive revision. The study constitutes an original contribution to the DNA double strand break repair field by describing novel mechanistic insight. I support publication of the revised MS.

Point-by-point response to the reviewers' comments

July, 8th 2024

REVIEWERS' COMMENTS

Reviewer #2 (Remarks to the Author):

With new data provided and revision of the manuscript, the authors have properly addressed most of the concerns. This revised manuscript is now more solid. While lack of validation at the chromosomal level in human cells remains a limitation, this revised manuscript may provide some insights into RNA-mediated DSB repair.

-We thank the Reviewer for the positive comments on our study.

Reviewer #3 (Remarks to the Author):

This is the second revision of the manuscript by Jeon et al. In the earlier versions of the manuscript (original and revised), the authors have shown that both coding and non-coding transcript RNA facilitates DSB repair in a sequence-specific manner in yeast and human cells. In this version, the authors have included a few new experimental findings involving both mammalian and in yeast cells to establish the role of RNA in both NHEJ and MMEJ:

1. The authors have used an Exon-based DSB system in human cells where they have induced DSBs in the exon and provided evidence that the repair frequency is similar with spliced and non-spliced transcript, further indicating that sequence directed RNA-mediated repair is operative.

-We thank the Reviewer for the supportive comments.

2. They have verified some critical data in HEK293 cells and have utilized this cellular model with the anticipation that the transfected plasmids couldn't replicate in HEK293 cells due to the absence of the T-antigen. This data provided evidence that the RNA-mediated repair was independent of the DNA synthesis. However, an appropriate reference would be helpful to the common readers.

-We have added the reference by Tan et al., *Frontiers* 2021 doi: 10.3389/fbioe.2021.796991 At the beginning of the Result section entitled 'The sequence of a transcript RNA guides DSB and double-strand gap repair in the corresponding DNA by NHEJ and MMEJ independently from DNA synthesis' on page 11 on line 357-358: << The splicing and non-splicing constructs carry the SV40 origin of replication that is activated once transfected into the HEK293T cells because the cells express the T antigen¹⁹>>.

3. Furthermore, they have provided evidence for the involvement of NHEJ pathway by deleting KU70 gene in yeast.

Apart from this additional evidence, the authors have mostly provided long explanations/clarifications

for a large part of the concerns. Overall, this manuscript has improved from its previous version. However, still several concerns remain, mostly regarding the lack of providing human cellular and biochemical data as detailed below:

Comment 1: Abstract: “While RNA can be a template for HR, the direct role and extent of transcript RNA in DSB-repair via end joining, as well as its overall impact on repair outcomes, remain unknown.” While searching the literature, this reviewer finds three reports (PMIDs: 27703167; 32205441; 36758800) that have indeed shown that NHEJ-mediated repair of DSBs in mammalian cells can utilize nascent RNA as template. A recent report also showed that DNA polymerase θ , a polymerase involved in MMEJ repair, promotes RNA-templated repair in mammalian cells (PMID: 34117057). Most of these publications have already been cited by the authors and thus, the statement could be modified accordingly in the proper context.

-As indicated by the Reviewer, we did specifically recognize that NHEJ-related proteins have been found to form a multiprotein complex with RNA polymerase II and to be associated with transcribed genes after inducing a DSB in these DNA loci, suggesting that RNA may help error-free NHEJ in human cells. Also considering the editorial note to reduce the size of the abstract and remove references from the abstract, we modified the abstract accordingly: <<While most eukaryotic DNA is transcribed into RNA, providing complementary genetic information, much remains unknown about the direct impact of RNA on DSB-repair outcomes and its role in DSB-repair via end joining. Here, we show that ...>>. Moreover, to further address the Reviewer comment, we have edited the Introduction and modified the text on page 2 and 3, from line 57 to 68: <<Moreover, it is still unclear whether and how RNA plays a direct role in DNA repair. However, recent studies over the last decade have provided emerging evidence for RNA's more direct involvement in DSB repair. In budding yeast, an endogenous RNA transcript can be used ... Beyond HR, NHEJ-related proteins have been found to form a multiprotein complex with RNA polymerase II and to be associated with transcribed genes after inducing a DSB in these DNA loci, suggesting that RNA may help error-free NHEJ in human cells^{13,14}>>.

Our references #13 and 14 in the Introduction correspond to:

PMID 27703167: Chakraborty et al., and Hazra ‘Deficiency in classical nonhomologous end-joining-mediated repair of transcribed genes is linked to SCA3 pathogenesis’ PNAS 2020

PMID 36758800: Chakraborty et al., and Hazra ‘Human DNA polymerase η promotes RNA-templated error-free repair of DNA double-strand breaks’ JBC 2023

Our reference # 26 corresponds to PMID 34117057: Chandramouly et al., and Pomerantz ‘Pol θ reverse transcribes RNA and promotes RNA-templated DNA repair’ Sci Adv 2021, which we cite in the Discussion.

Comment 2: The authors earlier speculated that “by the time Cas9 has been transcribed, translated, and imported into the nucleus to cut the target site(s) together with the sgRNA(s), it is expected that there is already transcript from the DsRed gene on the different constructs”. The authors have provided a conclusive evidence from the genome integrated constructs in the yeast system where both spliced and non-spliced RNAs are present as evident from the RNA-seq data before the Cas9 is induced by the addition of galactose to the medium. However, the authors have simply modified the text without any experimental evidence in mammalian cells. In mammalian cells, all the plasmid transfections (DSB harboring DsRed construct, Cas9 and sgRNAs) are done together and there is no inducible system for controlling the expression of the sgRNAs, and thus, it is a more complicated system and difficult to follow and interpret the results. A more detailed discussion will clarify the issue.

- To better clarify this point, we modified the text of the Discussion on page 16, line 513-523 as follows: << The plasmid system for human cells allows easy engineering of splicing and non-splicing constructs, providing an exportable system for experiments in various cell lines like HEK293T wild-type, RNaseH2A KO cells, and HEK293 cells. Because constructs with the *DsRed* gene transcript, Cas9, and sgRNA are co-transfected in human cells, we assume that by the time Cas9 and sgRNA(s) reach the nucleus to cut the target site(s), *DsRed* gene transcripts are likely already present. This assumption is strongly supported by our yeast cell results, where splicing and non-splicing constructs are integrated into the genome and expressed from the constitutive pTEF promoter. The Cas9 transcript, expressed from a galactose-inducible promoter, is activated by adding galactose to the medium. Indeed, the findings from the plasmid constructs in human cells are consistent with the results from yeast chromosomal constructs, suggesting a conserved role for transcript RNA in DSB repair mechanisms across eukaryotic cells. Future directions include integrating the splicing and non-splicing constructs into the human genome>>>.

Comment 3: To address the concern regarding the lack of biochemical characterization of the repair pathways, the authors used a KU70 deleted yeast strain to perform the experiments. The intron-pop out frequency was similar in spliced vs non-spliced constructs indicating KU70-mediated NHEJ was responsible for such spliced RNA mediated intron deletion. They also have observed higher percentage of repair and therefore, survival for the branch-deletion construct that was significantly (3-fold) reduced in KU70 null-strains. This remains a good control to show R-NHEJ pathway is active in yeast. However, the MMEJ repair pathway was not well investigated. The authors could deplete one MMEJ component in mammalian cells to establish the pathway. Thus, overall, the data in yeast was much stronger compared to the results involving mammalian cells.

-We agree with the Reviewer that biochemical characterization, including roles of specific MMEJ factors is an important direction of our study. Indeed in the Discussion, on page 20, line 635-641 we write: << Future work will not only focus on investigating genetic factors, such as the effects of different NHEJ mutants in yeast and human cells and the role of the reverse transcriptase activity of Pol η ²⁶ in R-MMEJ in human cells, but will also explore how the position of DSBs relative to exon and intron sequences affects RNA-mediated DSB repair in cells. For example, the choice of using spliced RNA vs. non-spliced RNA (i.e., pre-mRNA) in R-NHEJ and R-MMEJ for the splicing constructs may mainly depend on the position of the DSB relative to the exon and intron sequences, and in part also on the distance of the microhomologies from the 3'-splice site for R-MMEJ>>>.

Reviewer #4 (Remarks to the Author):

Storici and co-workers have provided a substantial experimental revision of the MS. The new data strengthens the conclusion that RNA transcripts influence DNA double-strand break repair outcomes along NHEJ outcomes (this is now more clearly demonstrated with a NHEJ repair mutant in Fig. 7) in yeast and human.

A remaining caveat regarding in vivo Cas9 cut efficiency is adequately acknowledged in the revised MS.

The concerns of this reviewer have been sufficiently addressed in this substantive revision. The study constitutes an original contribution to the DNA double strand break repair field by describing novel mechanistic insight. I support publication of the revised MS.

-We thank the Reviewer for their supportive comments on our study.